# PD-L1-directed PlGF/VEGF blockade synergizes with chemotherapy by targeting CD141+ cancer-associated fibroblasts in pancreatic cancer

Duk Ki Kim[1,2,3,4,8], Juhee Jeong[3,8], Dong Sun Lee[2], Do Young Hyeon[5], Geon Woo Park[3], Suwan Jeon[3], Kyung Bun Lee[6], Jin-Young Jang ●[7], Daehee Hwang[5], Ho Min Kim ●[1,2,9] ✉ & Keehoon Jung ●[3,4,9] ✉

Pancreatic ductal adenocarcinoma (PDAC) has a poor 5-year overall survival rate. Patients with PDAC display limited benefits after undergoing chemotherapy or immunotherapy modalities. Herein, we reveal that chemotherapy upregulates placental growth factor (PlGF), which directly activates cancer-associated fibroblasts (CAFs) to induce fibrosis-associated collagen deposition in PDAC. Patients with poor prognosis have high PlGF/VEGF expression and an increased number of PlGF/VEGF receptor-expressing CAFs, associated with enhanced collagen deposition. We also develop a multi-paratopic VEGF decoy receptor (Ate-Grab) by fusing the single-chain Fv of atezolizumab (anti-PD-L1) to VEGF-Grab to target PD-L1-expressing CAFs. Ate-Grab exerts anti-tumor and anti-fibrotic effects in PDAC models via the PD-L1-directed PlGF/VEGF blockade. Furthermore, Ate-Grab synergizes with gemcitabine by relieving desmoplasia. Single-cell RNA sequencing identifies that a CD141+ CAF population is reduced upon Ate-Grab and gemcitabine combination treatment. Overall, our results elucidate the mechanism underlying chemotherapy-induced fibrosis in PDAC and highlight a combinatorial therapeutic strategy for desmoplastic cancers.

Pancreatic ductal adenocarcinoma (PDAC) accounts for over 80% of pancreatic cancers and has a 5-year overall survival rate of <8%[1]. The standard treatment for advanced-stage PDAC is chemotherapy, including gemcitabine, FOLFIRINOX, and nab-paclitaxel, which scarcely improves patient survival[2,3]. While immune checkpoint inhibitors targeting cytotoxic T lymphocyte-associated protein 4 (CTLA-4) and programmed cell death-1 (PD-1)/programmed cell death-ligand 1 (PD-L1) have displayed efficacy in a variety of tumors, including advanced melanoma, lung cancer, sarcoma, and renal cell carcinoma[4–7], PDAC patients do not significantly benefit from immunotherapy[8,9]. One of the major challenges compromising therapeutic outcomes in PDAC is desmoplasia[10,11].

[1]Graduate School of Medical Science and Engineering, Korea Advanced Institute of Science and Technology (KAIST), Daejeon 34141, Republic of Korea. [2]Center for Biomolecular and Cellular Structure, Institute for Basic Science (IBS), Daejeon 34126, Republic of Korea. [3]Department of Anatomy and Cell Biology, Department of Biomedical Sciences, Seoul National University College of Medicine, Seoul 03080, Republic of Korea. [4]Institute of Allergy and Clinical Immunology, Seoul National University Medical Research Center, Seoul 03080, Republic of Korea. [5]School of Biological Sciences, Seoul National University, Seoul 08826, Republic of Korea. [6]Department of Pathology, Seoul National University College of Medicine, Seoul 03080, Republic of Korea. [7]Department of Surgery and Cancer Research Institute, Seoul National University College of Medicine, Seoul 03080, Republic of Korea. [8]These authors contributed equally: Duk Ki Kim, Juhee Jeong. [9]These authors jointly supervised this work: Ho Min Kim, Keehoon Jung. ✉e-mail: hm_kim@kaist.ac.kr; keehoon.jung@snu.ac.kr

Cancer-associated fibroblasts (CAFs) are one of the abundant cell types in the desmoplastic stroma, which is the major source of extracellular matrix (ECM) within the tumor microenvironment (TME)[12]. Emerging evidence indicates that the dense collagen matrix confers resistance to anti-PDAC therapies[13,14]. Moreover, activated CAFs secrete paracrine ligands that promote tumor growth through angiogenesis and immunosuppression[15]. Therefore, various therapeutic targets have been identified for suppressing CAF activation, such as the angiotensin inhibitor losartan, which reduces tumor fibrosis by suppressing TGF-β[13]. The TGF-β inhibitor, galunisertib, improves overall survival in patients with unresectable pancreatic cancer when administered together with gemcitabine[16]. Another phase I trial of galunisertib in combination with duravalumab displayed an acceptable tolerability and safety profile[17]. Small-molecule inhibitors targeting CAF/fibrosis-related molecules, including FGFR, CXCR4, ROCK, and FAK, have also been developed[18–20]. Despite extensive efforts, none of these agents have received FDA approval for the treatment of PDAC due to limited efficacy. Such frustrating outcomes may be due to the intrinsic heterogeneity of the CAF population within the TME.

Although single-cell analyses have revealed several subtypes of CAFs, including inflammatory CAFs (iCAFs), myofibroblastic CAFs (myCAFs), antigen-presenting CAFs (apCAFs), and mesenchymal stem cell-derived CAFs[21–24], the CAF subtype playing a major role in PDAC desmoplasia remains unclear. Recently, a blockade of the placental growth factor (PlGF) was shown to enrich quiescent CAFs and reduce desmoplasia, improving survival in intrahepatic cholangiocarcinoma mouse models[25], albeit the uncertain relevance to PDAC. PlGF is a member of the vascular endothelial growth factor (VEGF) family[26]. In humans, four PlGF isoforms (PlGF-1–4) have been identified, of which

only PlGF-2 is present in mice[26]. Unlike VEGF, which binds to both VEGF receptor 1 (VEGFR1) and 2, PlGF binds only to the former and to the co-receptors neuropilin-1 and -2 (NRP-1 and NRP-2)[26]. PlGF promotes tumor angiogenesis[27], enhancing cancer cell metastasis[28], invasion, and survival[29] and stimulates murine liver fibrosis[30]. However, the role of PlGF in PDAC is unclear, resulting in a lack of efficient PlGF/VEGF-targeting cancer therapeutics.

In this study, we show that chemotherapy upregulates PlGF, which directly activates CAFs to produce ECM in PDAC. We confirm the relevance of these findings by analyzing specimens from patients with PDAC. Furthermore, we develop a multi-paratopic VEGF decoy receptor, dubbed "Ate-Grab," to sequester PlGF/VEGF within the TME. The antitumor and anti-fibrotic effects of Ate-Grab are explored, in addition to its mechanism of action. These findings provide a mechanistic basis for developing efficient therapeutic strategies to treat desmoplastic cancers.

## Results

### Chemotherapy induces tumor fibrosis in murine orthotopic PDAC

We assessed the efficacy of gemcitabine in an orthotopic PDAC mouse model established using Pan02 cells (Supplementary Fig. 1a). Gemcitabine treatment did not significantly reduce in vivo tumor growth and tumor weight, compared to the control (Fig. 1a, b). Since tumor fibrosis is often associated with chemotherapy resistance in PDAC[14,31,32], we analyzed collagen fibers (major ECM components) and their source, CAFs, within tumors. As visualized via second-harmonic generation (SHG) using the two-photon laser, we observed a significant increase of tumor fibrosis in gemcitabine-treated Pan02 tumors compared to

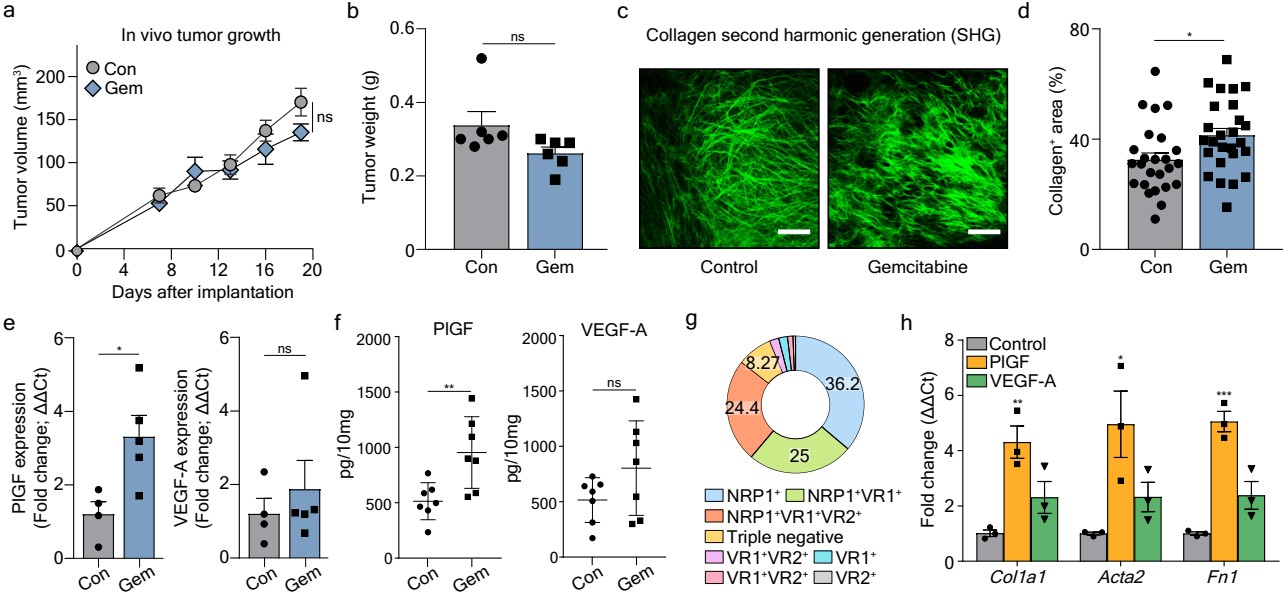

**Fig. 1 | Gemcitabine induces tumor fibrosis through increased PlGF/VEGF-A expression, which activates cancer-associated fibroblasts (CAFs).** On day 10 when tumor volume reached 50–100 mm³, tumor-bearing mice were intraperitoneally treated with either PBS (control, n = 6) or gemcitabine (50 mg/kg every 3 days for five times, n = 6). **a** Tumor volumes of control (Con) and gemcitabine (Gem) treated mice. ns, no significance. Two-way ANOVA with Sidak's multiple comparisons test was performed. **b** Tumor weights of control (Con) and gemcitabine (Gem) treated mice (n = 6/group) compared at the end of the experimental period. No statistical difference was observed. **c** Representative images showing second-harmonic generation (SHG) signals, which indicate the distribution of collagen. SHG signals were obtained at 840 nm and pseudo-colored with green. Scale bar, 100 μm. **d** Average percentages of collagen⁺ area out of the total area in tumors treated with PBS (control) or gemcitabine (n = 25/group). For quantification, four to five SHG images of 512 μm in width and height were obtained per tumor. P = 0.013.

**e** Relative *Plgf* and *Vegfa* mRNA expression levels of Pan02 tumor tissue homogenates measured by qRT-PCR. Data from four biological repeats for control group and five biological repeats for gemcitabine group were analyzed. Plgf: P = 0.022, Vegfa: P = 0.509. ns, no significance. **f** Relative *Plgf* and *Vegfa* protein levels of Pan02 tumor tissue homogenates measured by ELISA. Data from seven biological repeats (n = 7) were analyzed. PlGF: P = 0.008, VEGF-A: P = 0.133. ns, no significance. **g** Expression of PlGF/VEGF-related receptors (NRP1; VR1, VEGFR1; VR2, VEGFR2) in CAFs. **h** RT-PCR detection of myCAF markers, *Col1a1*, *Acta2*, and *Fn1* transcripts, in CAFs in response to PlGF, or VEGF-A. After sequential serum deprivation for sorted CAFs, the indicated proteins (PlGF, VEGFA) and atezolizumab or Ate-Grab were added and co-incubated for 6 h. Data from three biological repeats (n = 3) were analyzed. Col1a1: P = 0.005, Acta2: P = 0.030, Fn1: P < 0.001. All Data are presented as the mean ± SEM. *P < 0.05, **P < 0.01, ***P < 0.001 vs. control; two-tailed Student's t-test (**b, d, e, f, h**). Source data are provided as a Source Data file.

controls (Fig. 1c, d). Flow cytometry demonstrated indicated increased PDGFRα⁺ CAFs after gemcitabine treatment, which was consistent with the increased collagen deposition (Supplementary Fig. 1b–d). These results suggested that gemcitabine treatment increases CAF populations in the PDAC microenvironment and facilitates tumor fibrosis.

## Chemotherapy-induced PlGF/VEGF directly activates CAFs to produce ECM

Gemcitabine upregulates the expression of PlGF[27,33], which activates fibroblasts and promotes fibrosis in several pathological conditions[25,30,34]. To explore the significance of PlGF in pancreatic cancer, we measured its mRNA and protein levels in our orthotopic PDAC model and observed an increase after gemcitabine treatment, while the level of VEGF-A increased slightly (Fig. 1e, f). To determine whether the upregulation of PlGF was associated with CAF-mediated tumor fibrosis, we analyzed the expression of VEGF family member receptors on PDGFRα⁺ CAFs. Flow cytometry revealed that the majority of CAFs (~92.7%) in the Pan02 TME expressed PlGF co-receptor NRP1, while a significant portion of CAFs also expressed either VEGFR1 (~52.2%) or VEGFR2 (28.5%) (Fig. 1g; Supplementary Fig. 1e). The percentage of CAFs that did not express NRP1, VEGFR1, or VEGFR2 stayed below 8.27%, which implied that VEGF family members, particularly PlGF, are likely to interact with and stimulate CAFs (Fig. 1g).

Next, we analyzed the effects of VEGF-A and PlGF on CAFs with regard to fibrosis. Using previously reported CAF subsets[21], we evaluated how the marker expressions are affected by PlGF or VEGFA. Primary PDGFRα⁺ CAFs isolated from murine orthotopic Pan02 tumors (Supplementary Fig. 1b) were treated with recombinant mPlGF or mVEGF-A$_{164}$, and several marker genes of CAF subsets were determined. Recombinant mPlGF treatment induced myCAF marker expression; a ~4-fold increase in *Col1a1* and a ~5-fold increase in *Acta2* and *Fn1* expression compared to controls (Fig. 1h). Treatment with recombinant mVEGF-A$_{164}$ induced *Col1a1* expression, but not significantly (Fig. 1h). While PlGF upregulated the expression of several myCAF marker genes and induced a tendency to decrease in apCAF markers, no significant change in the expression of iCAF markers was observed (Supplementary Fig. 1f). Therefore, PlGF/VEGF-A can directly activate CAFs to produce ECM components, such as collagen, potentially promoting desmoplasia in PDAC.

## Patients with poor prognosis show high levels of collagen deposition, PlGF/VEGF, and CAFs

To validate our pre-clinical findings, we analyzed surgically resected primary tumor tissues from PDAC patients (Supplementary Table 2). Through proteogenomic analyses of human PDAC tumors, we previously identified six PDAC subtypes with different prognoses[35]. We performed Masson's trichrome staining to observe tumor fibrosis and immunohistochemistry (IHC) staining to measure the expressions of PLGF, VEGFA, and their receptors (NRP1, VEGFR1, VEGFR2) using tissues derived from 10 PDAC patients with poor prognosis and 10 PDAC patients with good prognosis (Supplementary Fig. 2a). Tumor fibrosis was negatively correlated with patient prognosis ($r = -0.6499$, $p = 0.002$) as was PLGF and VEGFA expression (Fig. 2a–d; Supplementary Fig. 2b, c). Furthermore, tumor fibrosis demonstrated a significant correlation with PlGF and VEGF expression ($r = 0.6586$, $p = 0.002$; $r = 0.5955$, $p = 0.006$ respectively) (Fig. 2e, f).

Double-staining IHC analysis revealed that CAFs expressed PlGF/VEGF receptors (Fig. 2g), and NRP1⁺ CAF, VR1⁺ CAF, and VR2⁺ CAF were strongly negatively correlated with patient prognosis ($r = -0.7199$, $p < 0.001$; $r = -0.6898$, $p < 0.001$; $r = -0.7131$, $p < 0.001$, respectively) (Fig. 2h–j and Supplementary Fig. 2d, e), while NRP1⁺ CAF and VR1⁺ CAF positively correlated with tumor fibrosis measured by Masson's trichrome⁺ area ($r = 0.594$, $p = 0.006$; $r = 0.5353$, $p = 0.015$ respectively) (Fig. 2k and Supplementary Fig. 2f, g). Consistent with preclinical data

obtained from orthotopic PDAC mice, expressions of PlGF/VEGF-A, VR1⁺CAF, and NRP1⁺CAF represented a statistically significant correlation with tumor fibrosis and PDAC patient prognosis.

## PD-L1 is expressed on CAFs in human and murine PDAC

Based on the results above, we postulated that upregulated PlGF/VEGF directly activates CAFs through receptors NRP1/VEGFR1/VEGFR2, promoting desmoplasia via collagen production and compromising chemotherapy efficacy. Thus, we sought to target PlGF/VEGF-A within the TME to inhibit CAF activation and subsequent tumor fibrosis.

PD-L1 is an established target for cancer immunotherapy[36]. Studies have demonstrated that CAFs, unlike normal fibroblasts, express PD-L1 on their surface[37,38]. IHC staining using PDAC patient-derived tumor tissues revealed that PD-L1-expressing CAFs were negatively correlated with patient prognosis and positively correlated with tumor fibrosis ($r = -0.6672$, $p = 0.001$; $r = 0.7805$, $p < 0.001$, respectively) (Fig. 2l–n; Supplementary Fig. 2h, i). Additionally, we analyzed the protein expression of PD-L1 in murine orthotopic Pan02 tumors using flow cytometry (Supplementary Fig. 2j and k), and confirmed that PD-L1-expressing CAFs are abundant in the TME, compared to other PD-L1-expressing cell types (Fig. 2o). Therefore, PD-L1 expression could enable CAF targeting within the TME.

## Ate-Grab, a multi-paratopic VEGF decoy receptor for PD-L1-directed PlGF/VEGF blockade

We previously developed VEGF-Grab as a soluble decoy receptor consisting of the glycosylated D2-D3 domain of VEGFR1 and the Fc domain of human IgG1, with improved binding affinity to VEGF-A and PlGF[39]. Moreover, VEGF-Grab fused with a single-chain variable fragment (scFv) could be designed to target specific molecules[40]. Herein, we developed "Ate-Grab," which consists of the scFv and the [G4S]$_3$ linker of the anti-PD-L1 antibody, atezolizumab, fused to the N terminus of VEGF-Grab, to target PlGF/VEGF within the PD-L1-enriched TME (Fig. 3a). Recombinant Ate-Grab protein was produced using the Expi™ 293F transient expression system (Supplementary Fig. 3a). Purified Ate-Grab was analyzed through SDS-polyacrylamide gel electrophoresis (PAGE) under reducing and non-reducing conditions (Fig. 3b). Biochemical analyses indicated that Fc-mediated dimerization was well maintained and the molecular weights of purified Ate-Grab were higher than those originally calculated, which was due to glycosylation, as previously reported[39,40]. We then evaluated the binding affinities of Ate-Grab for VEGF-A, PlGF, and PD-L1 via ELISA (Supplementary Fig. 3b) The inner paratope of Ate-Grab bound to VEGF-A and PlGF with high affinity, comparable to that of VEGF-Grab (Fig. 3c). Moreover, there was no difference in PD-L1 binding affinity between Ate-Grab and atezolizumab (Fig. 3c). We performed cell-based assays to evaluate the Ate-Grab blocking capacity and found that Ate-Grab inhibited VEGFR2 downstream signaling to a similar extent as VEGF-Grab (Fig. 3d–f). Tube formation assays (Fig. 3g, h) and migration assays (Supplementary Fig. 3c, d) indicated that Ate-Grab retained the original capacity of VEGF-Grab for endothelial cell (EC) inhibition. Furthermore, Ate-Grab exhibited a similar PD-L1 axis-blocking activity to atezolizumab, as determined via bio-blockade assays (Fig. 3i). Immunofluorescence staining revealed that Ate-Grab (not VEGF-Grab) bound to PD-L1-expressing Pan02 cells to an extent comparable to atezolizumab (Supplementary Fig. 3e). Based on previously published CAF subset markers by Elyada et al.[21], we examined in vitro marker gene expression of myCAF and apCAF in our Pan02 tumor-infiltrated CAFs in response to atezolizumab, VEGF-Grab, or Ate-Grab, and observed that Ate-Grab effectively inhibits PlGF-induced myCAF marker upregulation (*Col1a1*, *Acta2*) compared to atezolizumab, while no significant change was observed with apCAF marker expression (*Cd74, H2-Ab1*) between the two groups (Fig. 3j). Therefore, we confirmed that Ate-Grab exhibited potent anti-PlGF/VEGF activity and blocked PD-L1 signaling.

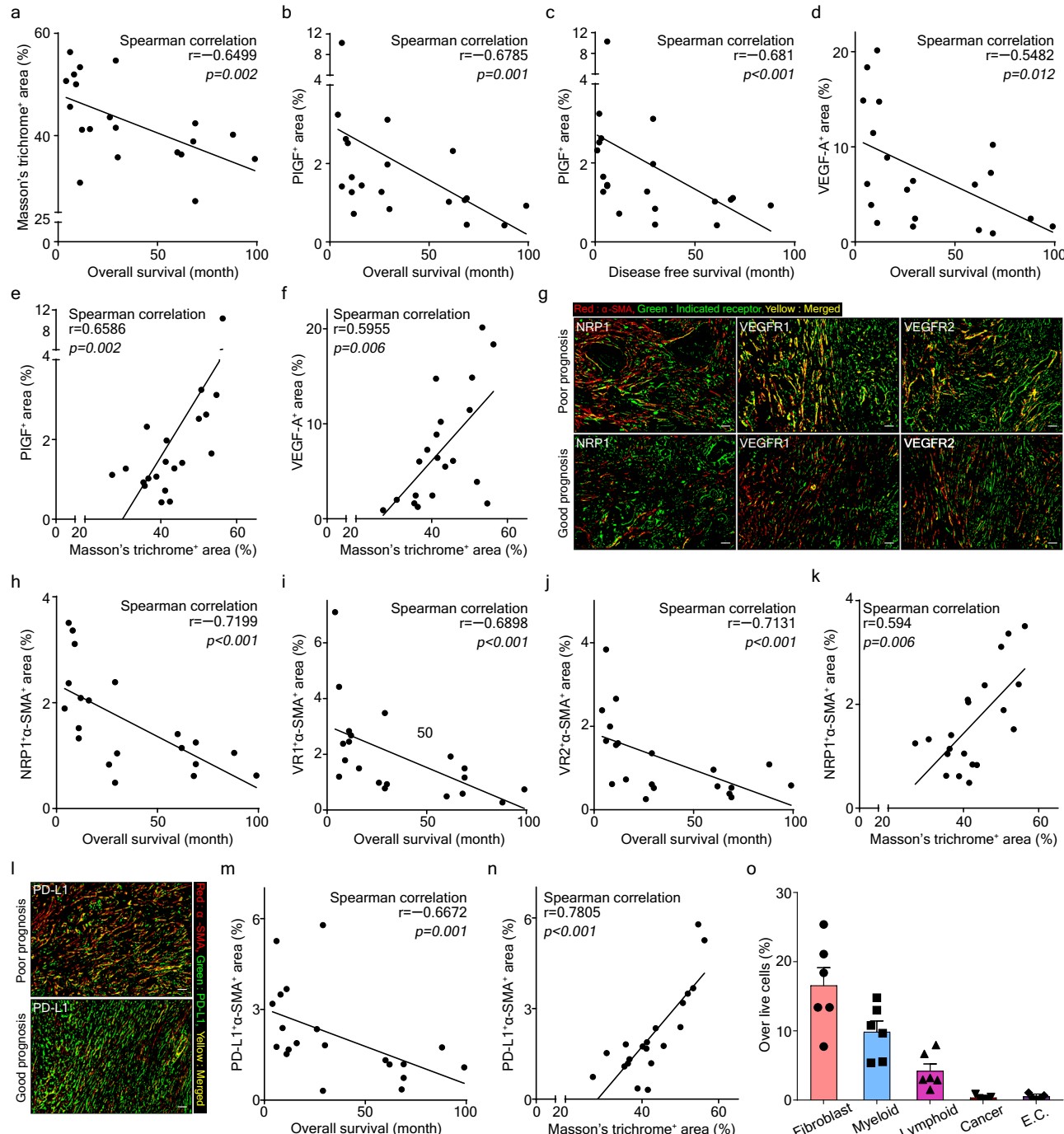

**Fig. 2 | PDAC patients with poor prognosis have elevated levels of intratumoral collagen, PlGF/VEGF-A, and CAFs expressing NRP1/VEGFR1/VEGFR2/PD-L1.** Correlation analyses between **a** tumor fibrosis (measured by Masson's trichrome[+] area) and overall survival of PDAC patients, between PlGF[+] area% (detected by IHC staining) and **b** overall survival or **c** disease-free survival of PDAC patients, between **d** VEGFA[+] area% (detected by IHC staining) and overall survival of PDAC patients, and between tumor fibrosis and **e** PlGF[+] area% or **f** VEGFA[+] area%. Data from 20 PDAC patient samples were analyzed (n = 20). Statistical significance was accessed by two-tailed Student's t test. **g** Double IHC staining for α-SMA with NRP1, VEGFR1, or VEGFR2 in tumors from PDAC patients with poor prognosis and good prognosis. Images were produced via Image J. Red color and DAB color were separated via color deconvolution of IHC images and merged using color merge functions. The red color represents α-SMA in CAFs, the green color represents each indicated receptor, and the yellow color (merged area) indicates co-expression of the two proteins. Four random spots additionally examined showed similar results. Scale

bar, 100 μm. **h–k** Correlation analyses between overall survival of PDAC patients and **h** NRP1[+]α-SMA[+] area%, **i** VR1 (VEGFR1)[+]α-SMA[+] area%, or **j** VR2 (VEGFR2)[+]α-SMA[+] area%, and between **k** tumor fibrosis and NRP1[+]α-SMA[+] area%. Data from 20 PDAC patient samples were analyzed (n = 20). Statistical significance was accessed by two-tailed Student's t test. **l** IHC staining for α-SMA and PD-L1. Red, α-SMA in CAFs; green, PD-L1; yellow, co-expression of α-SMA and PD-L1. Four random spots additionally examined showed similar results. Correlation analyses between PD-L1[+]α-SMA[+] area% and **m** overall survival of PDAC patients or **n** tumor fibrosis. Data from 20 PDAC patient samples were analyzed (n = 20). Statistical significance was accessed by two-tailed Student's t test. **o** Flow cytometry analysis of the ratio of PD-L1[+] cells (indicated) over Pan02 tumor-infiltrating live cells. Data from six Pan02 tumor-bearing mouse-derived tumor samples were analyzed (n = 6). Data are presented as the mean ± SEM. Cancer, cancer cell; E.C., endothelial cell. Source data are provided as a Source Data file.

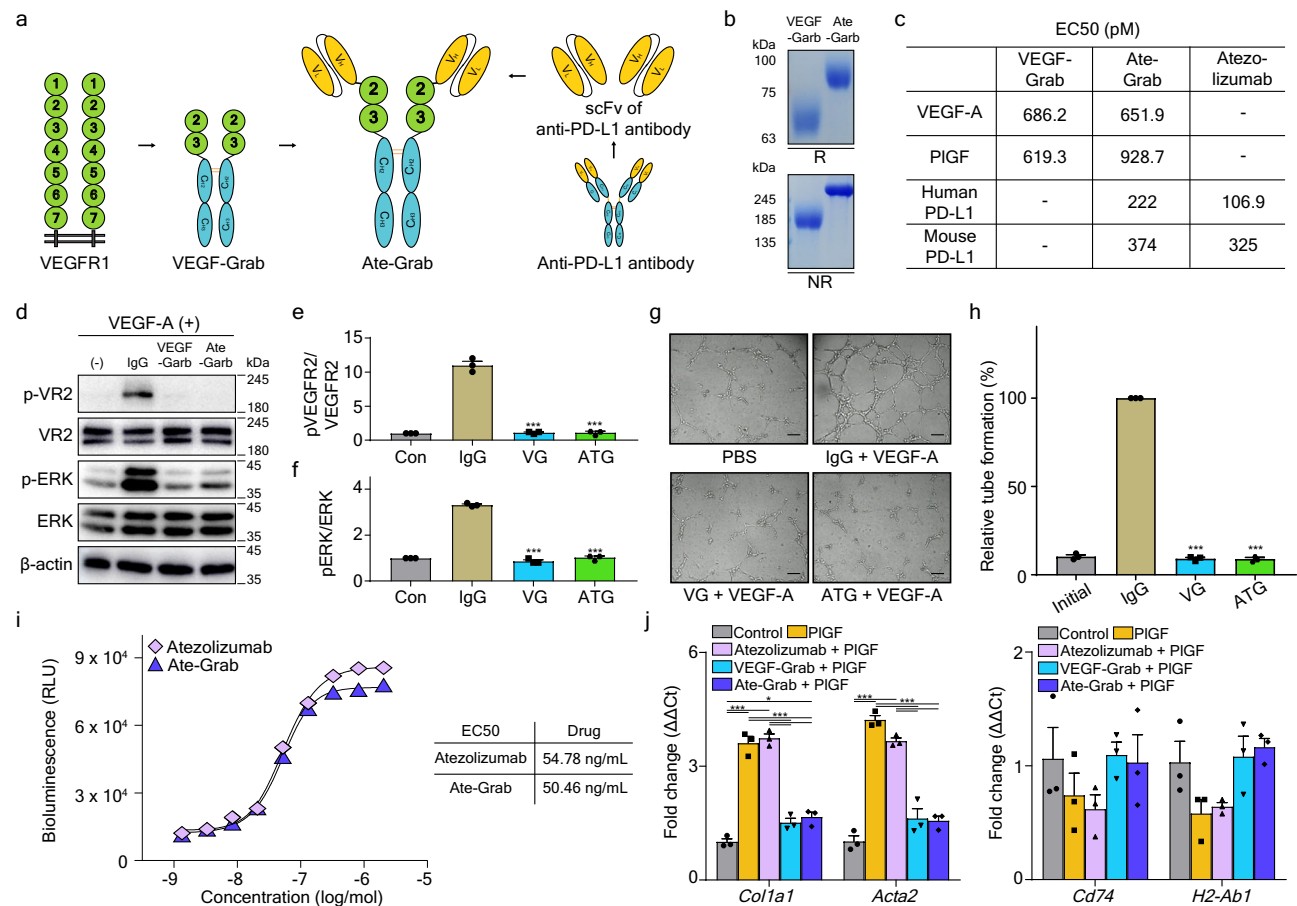

**Fig. 3 | Ate-Grab is a multi-paratopic decoy receptor that can target both PlGF/VEGF-A and the PD-1/PD-L1 axis. a** Schematic diagram of the Ate-Grab design. The single-chain variable fragment of atezolizumab with a [G4S]3 linker was fused to the N-terminus of VEGF-Grab. **b** Representative image of SDS-PAGE analysis on VEGF-Grab and Ate-Grab. Both reducing and non-reducing conditions were used for 10% acrylamide gel electrophoresis, and products were visualized through Coomassie blue staining. R, Reducing condition; NR, Non-reducing condition. **c** Target binding affinities of Ate-Grab and template drugs assessed via ELISA. **d** Western blot analysis for phospho-VR2, VR2, phospho-ERK, ERK, and β-actin after treatment with IgG, VEGF-Grab (VG), Ate-Grab (ATG), or without any treatment (control) in HUVECs incubated with VEGF-A. Total and phosphorylated levels of VEGFR2 (VR2) and ERK were detected. The experiment was repeated for three times independently with similar results. Relative quantities of **e** phospho-VEGFR2/VEGFR2 and **f** phospho-ERK/ERK across control, IgG, VG, and ATG groups. Data from three biological

repeats were analyzed ($n = 3$/group). **g** Representative images from wound-healing assay depicts tube formation of HUVECs treated with each indicated protein combination (25 nM) in the presence of VEGF-A (1 nM). Scale bar, 500 μm. **h** Relative quantities of tube-formed area from the tube formation assay. Data from three biological repeats were analyzed ($n = 3$/group). **i** PD-1/PD-L1 blockade bioassay for Ate-Grab and atezolizumab. **j** Relative expression levels of myCAF (*Col1a1*, *Acta2*) and apCAF (*Cd74*, *H2-Ab1*) markers in CAFs from Pan02 tumors. CAFs were isolated from Pan02 tumors via fluorescence-activated cell sorting (FACS) and treated with PlGF or the indicated drug combinations with PlGF for 6 h after the sequential serum deprivation. Data from three biological repeats were analyzed ($n = 3$/group). Data are presented as the mean ± SEM. *$P < 0.05$ (*Col1a1*-Control vs Ate-Grab+PlGF in **j**: $P = 0.029$), ***$P < 0.001$ versus control (**e**, **f**, **h**, **j**). Statistical significance was accessed by One-way ANOVA with Tukey's multiple comparisons (**e**, **f**, **h**, **j**). Source data are provided as a Source Data file.

## Ate-Grab exerts antitumor and anti-fibrotic effects in an orthotopic murine PDAC model

To assess the in vivo specificity of Ate-Grab, we analyzed the distribution of Ate-Grab in mice. IVIS whole-body imaging revealed that Ate-Grab localized to tumor tissues more efficiently than VEGF-Grab, which lacks the anti-PD-L1 scFv (Fig. 4a). To determine the effect of Ate-Grab on tumor growth, we compared VEGF-Grab, atezolizumab, and Ate-Grab in the orthotopic Pan02 model (Supplementary Fig. 4a). Ate-Grab inhibited in vivo tumor growth by ~30% compared to the control, while atezolizumab and VEGF-Grab did not significantly reduce tumor growth (Fig. 4b). The end-point weights of isolated tumors were in line with in vivo tumor growth, as the Ate-Grab group exhibited a ~28% decrease compared to the control group (Fig. 4c).

We also evaluated tumor fibrosis by measuring collagen deposition via SHG two-photon microscopy. Ate-Grab treatment significantly reduced collagen deposition in the tumors as opposed to

atezolizumab and VEGF-Grab (Fig. 4d, e). We investigated the mechanisms underlying the reduced collagen deposition in the Ate-Grab group. Since CAFs are a major source of collagen in the TME[41,42], we analyzed CAFs in Pan02 tumors via flow cytometry (Fig. 4f). The Ate-Grab-treated group was the only one to significantly reduce CAFs compared to the control group (Fig. 4f).

As tumor fibrosis is closely related to vessel normalization[10,11], we evaluated Pan02 tumor vasculature through immunofluorescence analysis of CD31 as an endothelial cell marker and NG2 or PDGFRβ as a pericyte marker. The Ate-Grab-treated group, which exhibited decreased tumor fibrosis (Fig. 4d, e), had greater NG2+ or PDGFRβ+ pericyte coverage (Fig. 4g, h and Supplementary Fig. 4b, c). Furthermore, a perfusion assay using 2000 kDa dextran revealed that only Ate-Grab significantly recovered vessel perfusion (Fig. 4i, j). These results suggest that Ate-Grab treatment inhibits CAF activation by targeting VEGF/PlGF and relieves the vessel compression caused by tumor desmoplasia, promoting vessel normalization.

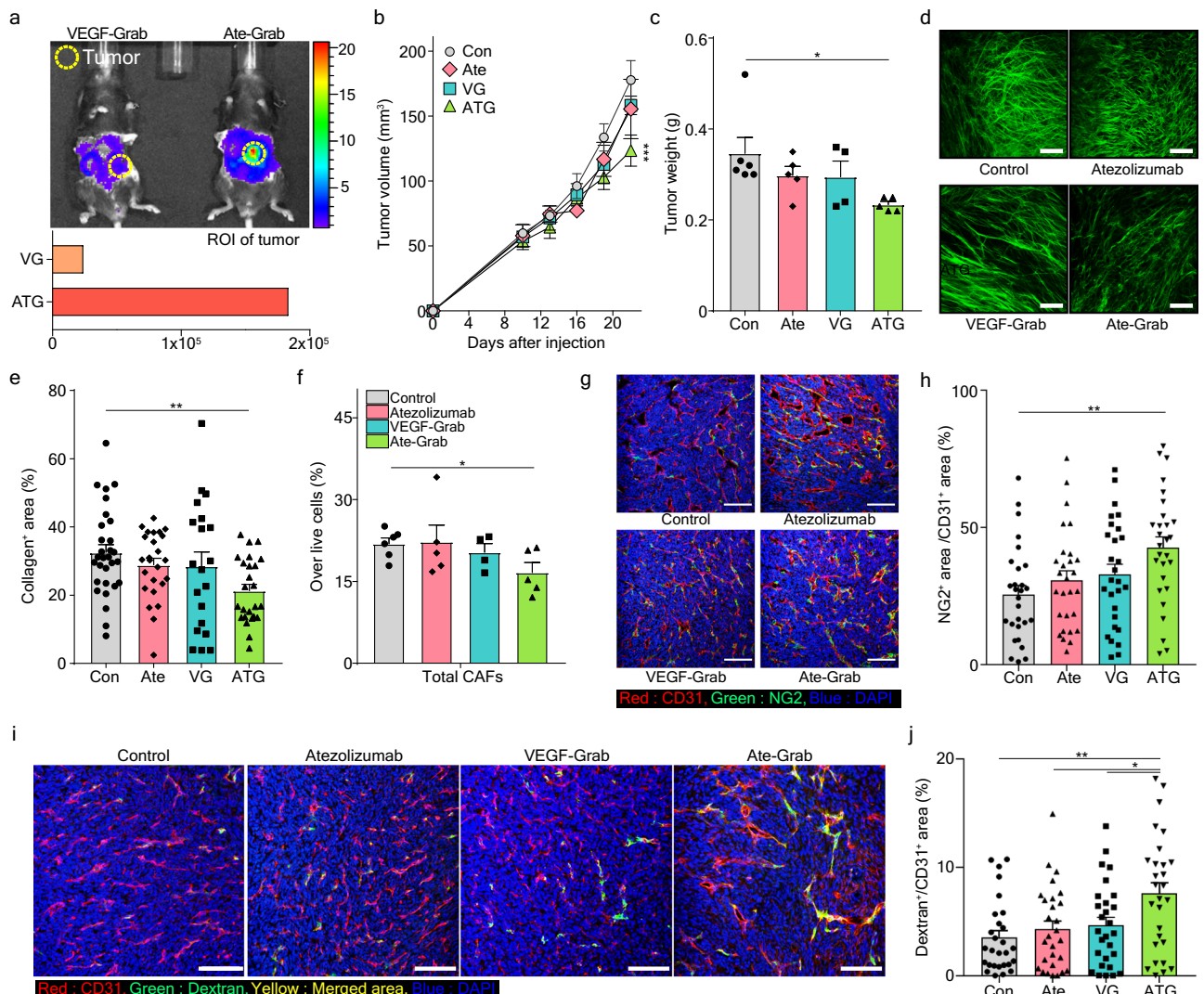

**Fig. 4 | PD-L1-directed tumor targeting of Ate-Grab exhibits antitumor and anti-fibrotic effects via regulating CAFs. a** In vivo distribution of Cy5.5-conjugated VEGF-Grab (VG) and Ate-Grab (ATG) in an orthotopic Pan02 tumor model. Fluorescence with overlapped mouse image (up) and quantification of ROI in the tumor (down) are shown. Yellow circle: Tumor mass. **b–h** When tumor volume reached 50–100 mm³, tumor-bearing mice were intraperitoneally treated with either PBS (control; $n = 6$), atezolizumab ($n = 5$), VEGF-Grab (VG; $n = 4$) or Ate-Grab (ATG; $n = 5$) (10 mg/kg, every 3 days). **b** Tumor volumes periodically monitored via abdominal ultrasonography. ***$P < 0.001$ versus control. Two-way ANOVA with Tukey's multiple comparisons. **c** Tumor weights measured at the end of the experimental period. *$P < 0.05$ ($P = 0.018$). Two-tailed Student's $t$ test. **d** Representative images showing SHG signals of each treatment-administered Pan02 tumor. **e** Average percentages of collagen⁺ area out of the total area of tumors treated with the indicated drugs, as quantified using ImageJ. Data from randomly selected fields of view were analyzed (Con, $n = 25$; Ate, $n = 28$; VG, $n = 21$; ATG, $n = 29$). **$P < 0.01$ ($P = 0.009$) versus control. One-way ANOVA with Tukey's multiple comparisons. **f** Flow cytometry analysis showing percentages of total CAFs in each treatment-administered Pan02 tumor. PBS (control; $n = 6$), atezolizumab ($n = 5$), VEGF-Grab (VG; $n = 4$) or Ate-Grab (ATG; $n = 5$), *$P < 0.05$ ($P = 0.029$). Two-tailed Student's $t$ test. **g** Representative immunofluorescence (IF) images staining for NG2 (green) and CD31 (red) in Pan02 tumors treated with different agents. Yellow indicates the co-expression of NG2 and CD31. Scale bar, 100 μm. **h** Quantifications of yellow regions (NG2⁺CD31⁺) in IF data were analyzed by ImageJ. Data from 28 randomly selected fields of view per group were analyzed ($n = 28$). **$P < 0.01$ ($P = 0.005$), One-way ANOVA with Tukey's multiple comparisons. **i** Representative immunofluorescence (IF) images staining for CD31 (red) and Dextran (green) in Pan02 tumors treated with different agents. Yellow indicates the co-expression of CD31 and Dextran. Scale bar, 100 μm. **j** Quantifications of Dextran⁺/CD31⁺ in IF data were analyzed by ImageJ. Data from 28 randomly selected fields of view per group were analyzed. *$P < 0.05$ (VG vs ATG: $P = 0.048$, Ate vs ATG: $P = 0.021$), **$P < 0.01$ (Con vs ATG: $P = 0.002$), One-way ANOVA with Tukey's multiple comparisons. All error bars in this figure are presented as the mean ± SEM. Source data are provided as a Source Data file.

## Ate-Grab synergizes with gemcitabine by relieving tumor fibrosis

Given the above, we examined its efficacy in combination with gemcitabine in our PDAC mouse model. While gemcitabine alone did not significantly reduce tumor growth but induced tumor fibrosis, co-treatment with Ate-Grab induced tumor suppression, dramatically suppressing tumor fibrosis (Fig. 5a–c, Supplementary Fig. 4d). Additionally, the total number of PDGFRα⁺ CAFs was significantly decreased in the combined treatment group (Fig. 5d). We

examined whether gemcitabine or Ate-Grab treatment affected signaling pathways related to CAF activation. Gemcitabine increased the expression of p-STAT3, and co-treatment with Ate-Grab significantly reversed p-STAT3 expression in CAFs (Supplementary Fig. 4e, f). Ate-Grab also showed a tendency to reduce p-SMAD2 and p-SMAD3 expression, albeit not significant (Supplementary Fig. 4g).

Activated CAFs reportedly suppress tumor growth and metastasis[43]. Hence, we performed H&E staining of lungs and livers

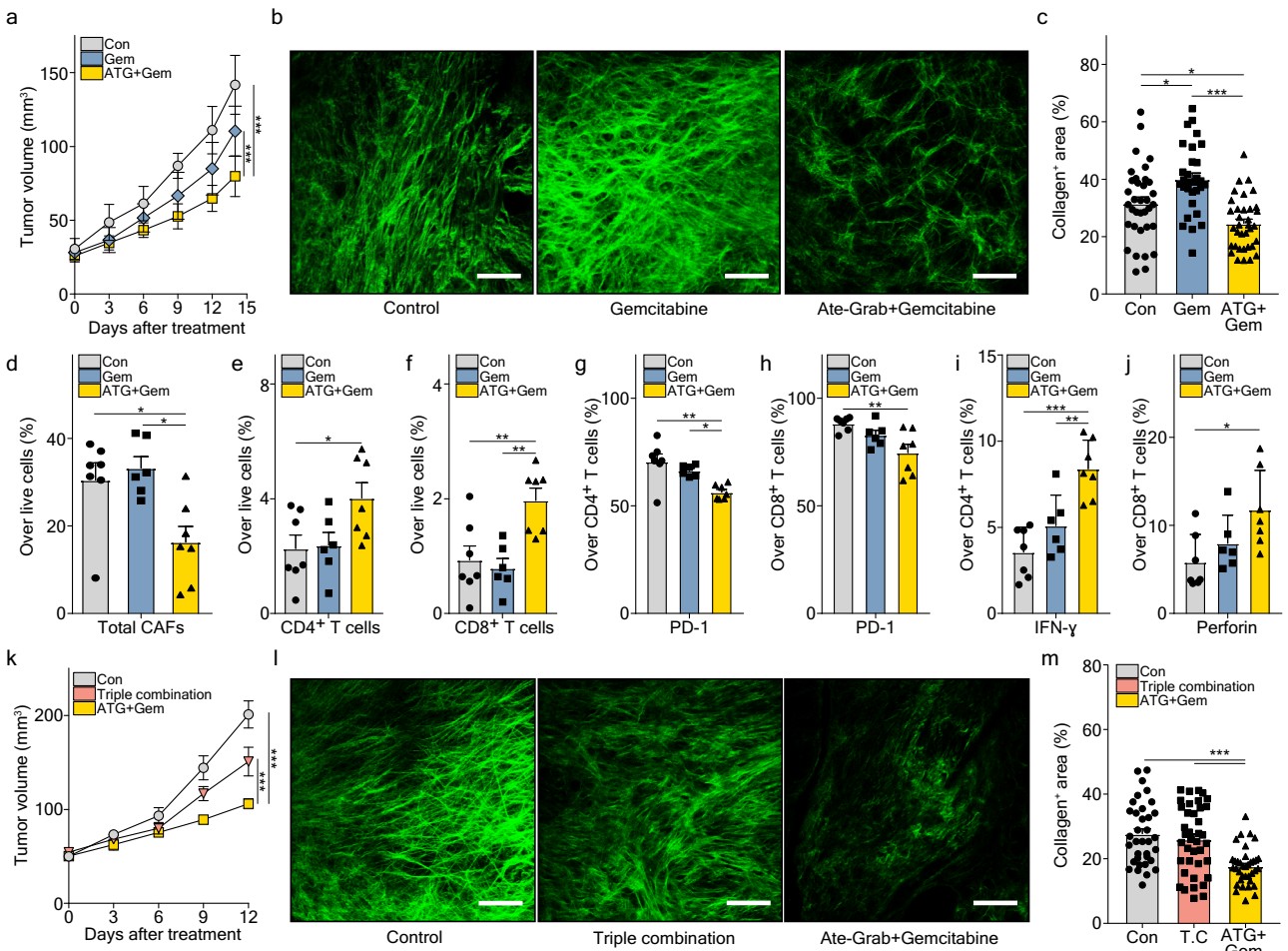

**Fig. 5 | Ate-Grab acts in synergy with the antitumor effects of gemcitabine, unlike the triple combination of atezolizumab, VEGF-Grab, and gemcitabine.** **a–j** When tumor volume reached 50–100 mm³, tumor-bearing mice were intraperitoneally treated with either PBS (Con; $n = 7$), gemcitabine (Gem; $n = 6$), and Ate-Grab+gemcitabine (ATG + Gem; $n = 7$) (50 mg/kg for gemcitabine, 10 mg/kg for Ate-Grab, every 3 days). **a** Tumor volumes of drug (indicated)-treated mice. ***$P < 0.001$; two-way ANOVA with Tukey's multiple comparisons. **b** Representative SHG images of Pan02 tumors, on day 24 after tumor inoculation. Green signals indicate collagen distribution. Scale bars, 100 μm. **c** Average percentages of collagen⁺ area in tumors of each group. Data from randomly selected fields of view were analyzed (Con, $n = 35$; Gem, $n = 30$; ATG + Gem, $n = 35$). *$P < 0.05$ (Con vs Gem: $P = 0.011$, Con vs ATG + Gem: $P = 0.030$), ***$P < 0.001$. one-way ANOVA with Tukey's multiple comparisons. Average percentages of **d** total CAFs (Con vs ATG + Gem: $P = 0.024$, Gem vs ATG + Gem: $P = 0.010$), **e** CD4⁺ T cells ($P = 0.048$) and **f** CD8⁺ T cells (Con vs ATG + Gem: $P = 0.007$, Gem vs ATG + Gem: $P = 0.04$) over total live cells in Pan02 tumors from each treatment group, measured by flow cytometry. *$P < 0.05$,

**$P < 0.01$, One-way ANOVA with Tukey's multiple comparisons. Average percentages of PD-L1 expression on **g** CD4⁺ T cells (Con vs ATG + Gem: $P = 0.001$, Gem vs ATG + Gem: $P = 0.023$) and **h** CD8⁺ T cells (Con vs ATG + Gem: $P = 0.006$), **i** IFN-γ expression in CD4⁺ T cells (Gem vs ATG + Gem: $P = 0.005$), and **j** Perforin expression in CD8⁺ T cells (Con vs ATG + Gem: P = 0.020). To measure cytokine expression in T cells, cells were stimulated with PMA/Ionomycin for 6 h before cytokine staining. *$P < 0.05$, **$P < 0.01$, ***$P < 0.001$. One-way ANOVA with Tukey's multiple comparisons. **k** Tumor volumes of control (Con; $n = 7$), Atezolizumab+VEGF-Grab+gemcitabine (Triple combination; $n = 9$) treated mice and Ate-Grab+gemcitabine (ATG + Gem; $n = 8$) treated mice, measured by ultrasonography. ***$P < 0.001$; two-way ANOVA with Tukey's multiple comparisons. **l** Representative SHG images of Pan02 tumors. Scale bars, 100 μm. **m** Average percentages of collagen⁺ area. Data from randomly selected fields of view were analyzed (Con, $n = 35$; T.C, $n = 39$; ATG + Gem, $n = 36$). ***$P < 0.001$. One-way ANOVA with Tukey's multiple comparisons. T.C, Triple combination. All error bars in this figure are presented as the mean ± SEM. Source data are provided as a Source Data file.

derived from Pan02 tumor-bearing mice; however, no metastases were observed in the specimens (Supplementary Fig. 4h).

As excessive desmoplasia impedes T cell infiltration, we analyzed lymphocyte populations within the TME (Supplementary Fig. 4i). Co-treatment of Ate-Grab with gemcitabine significantly increased the number of intratumoral CD4⁺ and CD8⁺ T cells compared to gemcitabine alone (Fig. 5e, f). There was no significant difference in B and NK cell abundance between treatment groups (Supplementary Fig. 4j). Beyond the difference in overall T cell infiltration, Ate-Grab co-treatment substantially reduced PD-1 expression, a T cell exhaustion marker, on CD4⁺ and CD8⁺ T cells, compared to the control or gemcitabine mono-treated group (Fig. 5g, h). Moreover, we observed that there were more IFN-γ- or perforin-secreting CD4⁺ and CD8⁺ T cells in the Ate-Grab combination group (Fig. 5i, j). Therefore, our data indicate

that Ate-Grab synergizes with gemcitabine by relieving desmoplasia and subsequently boosting antitumor immunity.

To determine whether the synergistic effect of Ate-Grab and gemcitabine was simply caused by the concurrent PD-L1 (atezolizumab) and VEGF/PlGF (VEGF-Grab) blockade, we compared antitumor effects of Ate-Grab+gemcitabine with atezolizumab+VEGF-Grab +gemcitabine (triple combination). We highlighted that Ate-Grab +gemcitabine showed a superior antitumor effect, while triple combination had negligible effects on tumor growth compared to the control group (Fig. 5k and Supplementary Fig. 5a, b). Similarly, only Ate-Grab+gemcitabine significantly inhibited tumor fibrosis and CAF numbers (Fig. 5l, m; Supplementary Fig. 5c). Furthermore, Ate-Grab +gemcitabine strongly induced a higher T cell infiltration and lower PD-1 expression (Supplementary Fig. 5d, e), accompanied by an

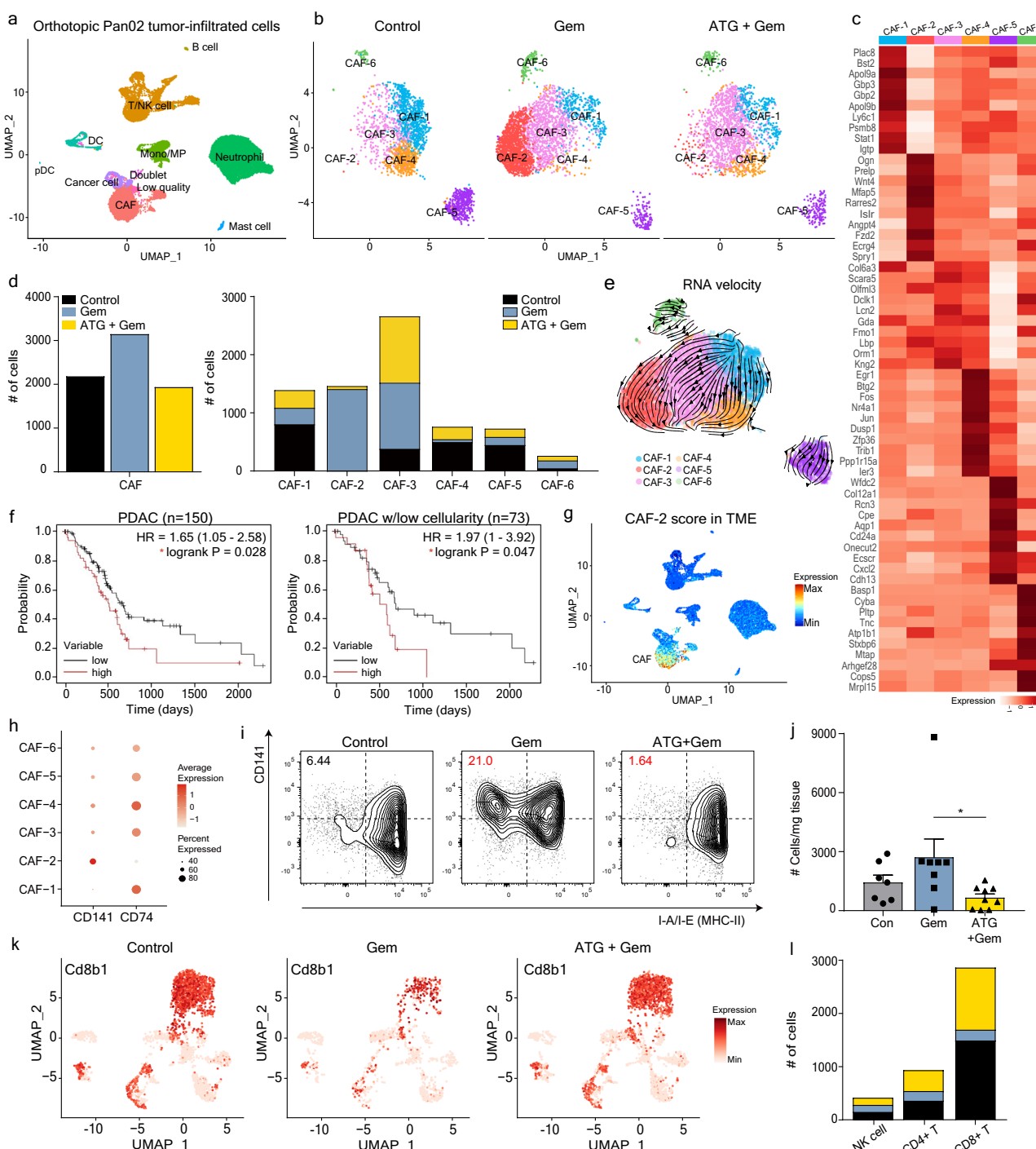

increase of antitumor cytokine expression of CD8+ T cells (Supplementary Fig. 5f), but triple combination did not. Overall, these data confirmed that the structural features of Ate-Grab enable the PD-L1-directed blockade of PlGF/VEGF within the TME in combination with gemcitabine.

## The CD141+ CAF population as an anti-fibrotic therapeutic target

To investigate the changes occurring in the TME cellular composition upon treatment, we compared the single-cell transcriptomes of untreated (Control), gemcitabine-treated (Gem), and Ate-Grab+gemcitabine-treated (ATG + Gem) PanO2 tumors. Live cells from five untreated tumors, five gemcitabine-treated tumors, and four Ate-Grab + gemcitabine-treated tumors were separately pooled and subjected to single-cell RNA sequencing (scRNA-seq) analysis (Supplementary

Fig. 6a). We sorted 35,697 cells based on a previously reported quality control scheme and identified nine distinct cell clusters (except for a low-quality cluster and doublet cluster) from these cells using a graph-based clustering method (Fig. 6a)[44]. The nine clusters were named based on the predominant expression of cellular markers including those related to CAFs, T/NK cells, B cells, monocytes/macrophages, neutrophils, DCs, plasmacytoid DCs, mast cells and cancer cells (Supplementary Fig. 6b). Of note, PlGF co-receptors, *Nrp1* and *Nrp2*, were specifically upregulated in CAFs within the PanO2 TME (Supplementary Fig. 6c, d).

Furthermore, we performed sub-clustering of CAFs to systematically investigate the effect of Ate-Grab in heterogeneous CAF populations. Six CAF sub-clusters (CAF-1−6) with distinct features were identified (Fig. 6b, c; Supplementary Fig. 6e). Surprisingly,

**Fig. 6 | Comprehensive analysis of single-cell RNA sequencing data revealed the CD141⁺ CAF population as a target of Ate-Grab in the orthotopic Pan02 tumor model. a** Uniform manifold approximation and projection (UMAP) plot for orthotopic Pan02 tumor-infiltrating cells (35,697 cells). Nine cell types (except for doublets and low-quality cells) were assigned based on the expression of marker genes. Pan02 tumors derived from 4 to 5 mice per group were allocated to each single-cell RNA seq result. **b** Unbiased clustering of CAF (cancer-associated fibroblasts) subsets revealed that combination therapy with Ate-Grab and gemcitabine (co-treatment) mostly depleted CAF-2. Control, untreated; Gem, gemcitabine mono-treated; ATG + Gem, Ate-Grab and gemcitabine co-treated. **c** Top ten differentially expressed genes (DEGs) in six CAF subpopulations. **d** The number of each CAF subpopulation was quantified and compared between the three groups. **e** RNA velocity vector field for CAF differentiation indicated by streamlines. **f** Left: Kaplan-Meier overall survival curve of 150 pure PDAC (pancreatic ductal adenocarcinoma) patients grouped based on the expression of the top 30 CAF-2 DEGs (AUC value > 0.7, pct.1-pct.2 > 0.05). Right: Kaplan-Meier overall survival curve of 73 pure PDAC patients with low tumor cellularity grouped based on the expression of the top 30 CAF-2 DEGs. HR: hazard ratio. **g** Feature plot of the average expression of the top 30 CAF-2 DEGs scored in the mother plot (Pan02 tumor-infiltrated cells) to show CAF specificity of CAF-2 DEGs. TME, tumor microenvironment. **h** Dot plots of *Cd141* and *Cd74* expression in the CAF subpopulations. **i** Flow cytometry validated the phenotype of CAF-2 at the protein level, defined as FSC-SSC/CD45⁻/CD31⁻/EpCAM⁻/PDGFRα⁺/ MHCII⁻/CD141⁺ (left upper quadrant). Gating panel corresponds to FACS data in Fig. 6j and Supplementary Fig. 6e. **j** Bar plot showing each treatment effect on CAF-2 composition in orthotopic Pan02 tumor. Data from seven samples of control group, eight samples of gemcitabine group and nine samples of Ate-Grab+gemcitabine group were analyzed. Data are presented as the mean ± SEM. One-way ANOVA with Tukey's multiple comparisons, *$P < 0.05$ ($P = 0.044$). **k** Feature plots of *Cd8b1* expression in untreated (left), gemcitabine-treated (middle) and Ate-Grab + gemcitabine-treated (right) tumor-infiltrating T/NK populations. **l** Stacked bar plot of the number of lymphocyte subsets in untreated (black), gemcitabine-treated (blue), and Ate-Grab+gemcitabine-treated (yellow) Pan02 tumors. $n = 1$/group. Source data are provided as a Source Data file.

comparative analysis of CAFs between the groups revealed that gemcitabine strongly increased CAF-2 population in the TME and that the addition of Ate-Grab to gemcitabine effectively reversed the increase of CAF-2 (Fig. 6b, d). Interestingly, the CAF-2 population, which was expanded by gemcitabine and suppressed by Ate-Grab, expressed genes related to multiple ECM components (Fig. 6c). Among them, *Ogn* and *Prelp* are involved in the stabilization of collagen structures (Fig. 6c)[45,46], highlighting the fibrosis-promoting roles of CAF-2. In addition, CAF-2 specifically expressed *Runx1* and *Wnt4*, which are important in myofibroblast differentiation (Fig. 6c)[47,48]. Gene set enrichment analysis revealed that the signaling pathways of "TGF-beta regulation of extracellular matrix" and "Wnt signaling pathway", which are activated in pathological fibrosis, were enriched in the CAF-2 subset (Supplementary Fig. 6g), while fibrosis-promoting secretory molecules including *Ogn, Prelp, Omd, Inhba*, and *Ecrg4* were confirmed among the top 30 differentially expressed genes (DEGs) in CAF-2 (Fig. 6c, Supplementary Fig. 6f). To gain insight into cellular transition statuses of CAFs, we performed RNA velocity analysis on our Pan02 CAF subsets and found that CAF-1 differentiates toward CAF-2. These results indicate that Ate-Grab inhibits the cellular transition toward CAF-2 (Fig. 6e).

To further validate our CAF subpopulations, we compared the signatures of our Pan02 CAF subpopulations to those of previously published fibroblast-enriched datasets (Supplementary Fig. 6h–m)[21,49]. In comparison with ref. [21], most Pan02-infiltrated CAF subsets except for CAF-2 highly expressed MHC-II-related genes that were representative signatures of apCAF, while each clustered subset further represented its own unique signature (Supplementary Fig. 6j). CAF-5 shared several myCAF signatures including Spp1, Col12a1, and Ecscr, while CAF-2 expressed some iCAF DEGs, such as *Ogn, Sema3c*, and *Mfap5* (Supplementary Fig. 6j). However, all CAF subsets from the Pan02 tumor hardly showed one-to-one correspondence with iCAF, myCAF, and apCAF[21]. For comparison with ref. [49], we sorted pure CAFs from the data, and re-clustered the sorted CAF cells, so that the DEGs of each CAF subset could be calculated for pure CAF cells (not contaminated with tissue fibroblasts or mesothelial cells) (Supplementary Fig. 6k, l). The comparison revealed that our CAF-2 shared some markers of c0 CAF, including *Gstm1, Ogn*, and *Sema3c*, while several markers of myCAF-like c2 CAF were conserved in CAF-5 (Supplementary Fig. 6m). Notably, CAF-4 represented a substantial similarity with c8 CAF, showing the most c8 CAF signatures (Supplementary Fig. 6m).

To determine the clinical value of CAF-2, we performed Kaplan–Meier overall survival analysis using publicly available PDAC patient data[50] (Fig. 6f; Supplementary Fig. 7a, b). The CAF-specific DEGs of CAF-2, showing higher expression in pancreatic tumors compared to normal tissues (Supplementary Fig. 8a), were negatively correlated with patient prognosis (HR = 1.65), and showed a higher hazard ratio (HR = 1.97) in PDAC patients with lower tumor cellularity (Fig. 6f, g), highlighting CAF-2 as a promising therapeutic target, especially for desmoplastic cancer.

While *Inhba*, revealed as CAF-2 marker gene (Supplementary Fig. 6f), is well known to induce collagen expression in CAFs[51,52], *Inhba* exhibited strong positive correlations with NRP1 and NRP2 in TCGA-PAAD as well as multiple collagen genes[53] (Supplementary Fig. 8b, c). Therefore, scRNA-seq data suggested that tumor fibrosis in pancreatic cancer was promoted via the PlGF-NRP1/NRP2 axis, which induced collagen production through secreted *Inhba* from the CAF-2 population. Moreover, this axis could be depleted by Ate-Grab.

Next, we aimed to identify specific CAF-2 surface markers, distinguishing it from other CAF subsets. Unlike other CAF subsets, which shared apCAF markers defined by MHCII-related features, CAF-2 hardly expressed MHCII-related genes (Fig. 6h). Instead, CAF-2 expressed a high level of CD141 (Fig. 6h). Thus, we defined CAF-2 as CD141⁺CD74(MHC II)⁻ CAFs based on scRNA-seq. Based on this, we successfully confirmed CAF-2 phenotypic marker expression at protein level via flow cytometry (Fig. 6i). Of note, CD141 was positively correlated with NRP1 and NRP2 expression in TCGA-PAAD tumors (Supplementary Fig. 8d). Consistent with scRNA-seq data, flow cytometry revealed that gemcitabine induced CD141⁺ CAF-2 accumulation in tumors, while Ate-Grab+gemcitabine co-treatment reversed accumulation of CAF-2 in tumors (Fig. 6i, j and Supplementary Fig. 8e).

Finally, we explored immune populations indirectly affected by Ate-Grab treatment. In the orthotopic Pan02 TME, PD-L1 was expressed not only in CAFs, but also in immune cells, especially myeloid cells (Supplementary Fig. 2k). Since the scRNA-seq data indicated Cd274 (PD-L1) mRNA expression in some neutrophils, we investigated neutrophil subclusters (Supplementary Fig. 8f). Ate-Grab did not further decrease PD-L1-expressing Neu-2 and Neu-5 subpopulations of neutrophils, which are characterized by type I interferon-sensitized pathways (Supplementary Fig. 8f–h). Meanwhile, it is notable that Ate-Grab treatment highly increased the Neu-4 subpopulation that could be involved in inflammatory responses, such as TNF-alpha signaling via NF-κB (Supplementary Fig. 8f–h). We also analyzed T/NK cell subpopulations and identified 13 different clusters (Supplementary Fig. 8i, j). Ate-Grab treatment increased tumor-infiltrating lymphocytes (TILs), especially CD8⁺ cytotoxic effector memory T cells (T$_{EM}$) (Fig. 6k, l, Supplementary Fig. 8k). Meanwhile, Pan02 tumor-infiltrating CD8⁺ cytotoxic T cells were composed of three different populations with distinct gene expression profiles (Supplementary Fig. 8l). Among these, CD8⁺T-1 (CD137$^{High}$ CD8⁺ T cells) were recently reported to exhibit greater antitumor activity compared to CD137⁻ CD8⁺ TILs in human HCC[54]. In fact, our single-cell analysis of Pan02 tumors revealed CD137$^{High}$ CD8⁺ T cells as the major population with the highest effector cytokine expression among the CD8⁺ cytotoxic T cells increased upon

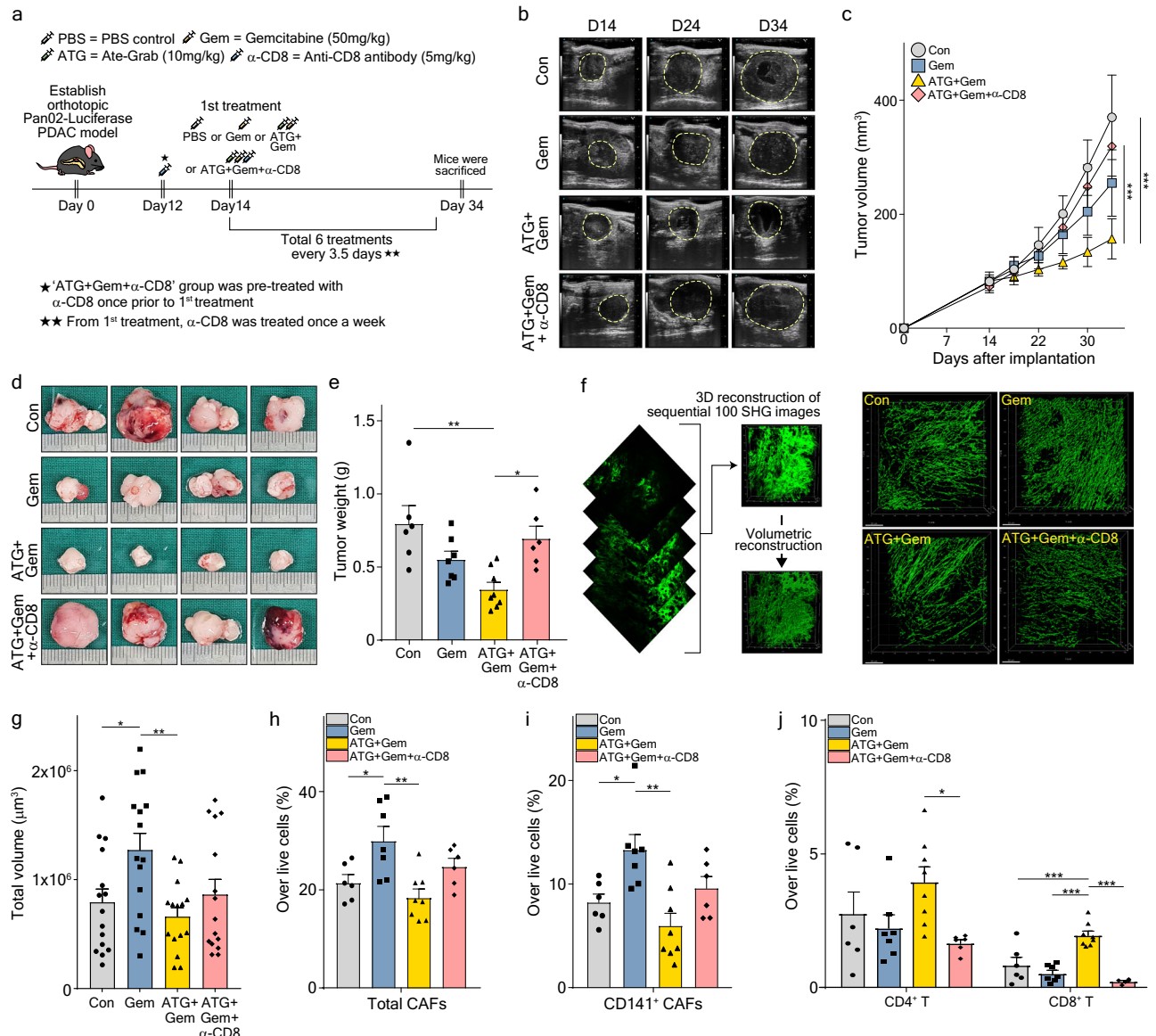

**Fig. 7 | CD8+ T cell mediates antitumor and anti-fibrotic effects of Ate-Grab.**
**a** Experimental treatment scheme. When the tumor volume reached 50–100 mm³, tumor-bearing mice were intraperitoneally treated with either PBS (control), gemcitabine (50 mg/kg, every 3.5 days), gemcitabine with Ate-Grab (10 mg/kg, every 3.5 days), or gemcitabine with Ate-Grab and anti-CD8 antibody (5 mg/kg, once a week). **b** Representative ultrasonographic images for orthotopic Pan02-Luciferase tumor growth in vivo. **c** Tumor volumes compared across indicated treatment groups (Con, n = 6; Gem, n = 7; ATG + Gem, n = 8; ATG + Gem + α-CD8, n = 6). Con, control; Gem, gemcitabine; ATG + Gem, Ate-Grab and gemcitabine co-treatment; ATG + Gem + α-CD8, Ate-Grab, gemcitabine, and α-CD8a antibody co-treatment. ***P < 0.001. two-way ANOVA with Tukey's multiple comparisons. **d** Representative images of KPC tumors from indicated treatment groups. Ruler scale, 1 mm. **e** Tumor weights compared across indicated treatment groups (Con, n = 6; Gem, n = 7; ATG + Gem, n = 8; ATG + Gem + α-CD8, n = 6). Data are presented as the mean ± SEM. *P < 0.05 (ATG + Gem vs ATG + Gem + α-CD8: P = 0.018), **P < 0.01 (Con vs ATG + Gem: P = 0.002), one-way ANOVA with Tukey's multiple comparisons. **f** Schematic diagram of 3D reconstruction of collagen fibers using

IMARIS software. SHG, Second harmonic generation. **g** Total volume of 3D reconstructed collagen fibers in tumors compared across different treatment groups. Data from 15 randomly selected spots per group (three spots per one tumor × five tumors) were analyzed (n = 15/group). Data are presented as the mean ± SEM. *P < 0.05 (Con vs Gem: P = 0.049), **P < 0.01 (Gem vs ATG + Gem: P = 0.007); one-way ANOVA with Tukey's multiple comparisons. Average percentages of **h** total CAFs (Con vs Gem: P = 0.043, Gem vs ATG + Gem: P = 0.002) and **i** CD141+ CAFs (Con vs Gem: P = 0.043, Gem vs ATG + Gem: P = 0.001) in Pan02 tumor microenvironment of each treatment group (Con, n = 6; Gem, n = 7; ATG + Gem, n = 8; ATG + Gem + α-CD8, n = 6), measured by flow cytometry. Data are presented as the mean ± SEM. *P < 0.05, **P < 0.01. One-way ANOVA with Tukey's multiple comparisons. **j** Average percentages of CD4+ and CD8+ T cells in Pan02 tumor microenvironment of each treatment group, measured by flow cytometry (Con, n = 6; Gem, n = 7; ATG + Gem, n = 8; ATG + Gem + α-CD8, n = 6). Data are presented as the mean ± SEM. *P < 0.05 (P = 0.039), ***P < 0.001. One-way ANOVA with Tukey's multiple comparisons. Source data are provided as a Source Data file.

Ate-Grab treatment (Supplementary Fig. 8l). Moreover, Ate-Grab +gemcitabine induced higher expression of *Gzmb*, *Ifng*, and co-stimulatory receptor *Cd28* compared to gemcitabine alone (Supplementary Fig. 8m).

Overall, scRNA-seq analysis indicated that Ate-Grab treatment significantly reduced the number of CD141+ CAF-2 cells, a key subset

that facilitates tumor fibrosis by promoting collagen production, and enhanced T cell infiltration in the TME in vivo.

## Antitumor effect of Ate-Grab is mediated by CD8+ T cells
Based on the increased intratumoral CD8+ T cells and their cytotoxic features observed with Ate-Grab+gemcitabine combination therapy,

we further determined whether the antitumor effect of Ate-Grab is directly mediated by immune-related mechanisms. An anti-CD8 neutralizing antibody was co-administered with Ate-Grab+gemcitabine (Fig. 7a), causing a substantial inhibition of the antitumor effects that were improved in response to the Ate-Grab+gemcitabine regimen (Fig. 7b–e). Specifically, in the Ate-Grab+gemcitabine+anti-CD8 antibody-treated group, tumor volume and weight increased to the levels of the control (Fig. 7b–e). Also, 3D reconstructed images of tumor collagen structure demonstrated that co-treatment of anti-CD8 antibody with Ate-Grab+gemcitabine did not significantly reverse the anti-fibrotic effect exerted by Ate-Grab, while individual collagen fiber volume and length did not change either (Fig. 7f, g; Supplementary Fig. 9a). Flow cytometry analysis also showed that co-treatment of Ate-Grab+gemcitabine with anti-CD8 antibody does not significantly affect total or CD141$^+$ CAF population, while dramatically depleting CD8$^+$ T cells and also reducing the number of CD4$^+$ T cells in vivo (Fig. 7h–j). These results demonstrate that the tumor-inhibiting effect of Ate-Grab is mediated by CD8$^+$ T cells in the TME.

### Efficacy of Ate-Grab in KRAS-mutated murine PDAC

KRAS mutation accounts for more than 90% of all human PDAC cases[55]. As Pan02 tumor is a KRAS-wild type[56], we verified the therapeutic efficacy of Ate-Grab in KRAS-mutated murine PDAC. Orthotopic murine pancreatic cancer models were generated using KPC001 cell line derived from GEMM (Pdx1-Cre, lox-stop-lox-KrasG12D/+, lox-stop-lox-tp53R172H/+)[57], and the mice were divided into three experimental groups based on the different treatment regimens (Supplementary Fig. 9b). Similar to the Pan02 tumor, KPC001 tumor volumes and weights were significantly lower in the Ate-Grab+gemcitabine treatment group than in the gemcitabine and untreated (control) groups (Fig. 8a–d). In addition, we validated that gemcitabine-induced tumor fibrosis, which was effectively suppressed by co-treatment with Ate-Grab (Fig. 8e, f). With the expression of VEGF-related receptors validated in KPC001 tumor-infiltrated CAFs (Fig. 8g, h; Supplementary Fig. 9c), flow cytometry analysis confirmed that gemcitabine treatment increased total CAF% in the TME and co-treatment with Ate-Grab significantly decreased CAFs (Fig. 8i). Notably, further analysis with representative markers of CAF subsets revealed that CD141$^+$ MHCII$^-$ CAFs, defined as CAF-2, were dramatically diminished in response to Ate-Grab co-treatment, while other CAF subsets showed no statistically significant difference between treatments (Fig. 8j; Supplementary Fig. 9c–e). Furthermore, higher infiltration of CD3$^+$ T cells, especially CD8$^+$ T cells, was observed in response to Ate-Grab and gemcitabine co-treatment, while no significant difference was identified in B and NK cell populations (Fig. 8k, l; Supplementary Fig. 9f). Overall, validation analysis with KPC001 orthotopic murine PDAC indicated that Ate-Grab has a comparable therapeutic effect on KRAS-mutated PDAC as that on KRAS-wild type PDAC.

In conclusion, Ate-Grab treatment significantly reduced the number of CD141$^+$ CAF-2 cells, a key subset of CAFs that facilitates tumor fibrosis by promoting collagen production, and enhanced effector T cell infiltration and antitumor activity in the TME (Fig. 9).

## Discussion

In this study, we found that chemotherapy induces the PlGF/VEGF upregulation in PDAC, which directly activated CAFs to produce ECM, leading to desmoplasia. Relieving desmoplasia via Ate-Grab-induced PlGF/VEGF blockade improved chemotherapy efficacy, presumably through the underlying mechanism described hereafter. Due to its anti-PD-L1 scFv, Ate-Grab can be efficiently delivered into PD-L1-rich tumor tissues. The subsequent Ate-Grab-mediated PlGF/VEGF blockade suppresses collagen-secreting CAFs. As the amount of ECM within the TME decreases, blood vessels originally compressed by the dense collagen matrix are "normalized," exhibiting increased pericyte coverage[11]. Vessel

normalization subsequently enables cytotoxic T lymphocyte infiltration and effector cytokine secretion, while the number of exhausted PD-1-positive T cells is decreased. These changes suggest that the immune microenvironment is skewed toward immune stimulation, possibly due to Ate-Grab-induced vessel normalization.

Ate-Grab treatment dramatically reduced the CAF-2 population, identified as CD141$^+$CD74(MHC II)$^-$ CAFs. Among CAF subtypes, CD141$^+$ CAFs resembled myCAFs with regard to gene expression and active collagen production. Notably, the characteristics of CD141$^+$ CAFs are generally different from those previously described for myCAFs. That is, CD141$^+$ CAFs secrete various factors, including INHBA, which can activate other CAF populations to further induce desmoplasia. The CAF-3 population identified in our scRNA-seq analysis also shared some features with myCAFs, including the expression of *Col12a1*. Such ambiguity in CAF subset discrimination may be due to differences between the TMEs analyzed. Nevertheless, the identification of CD141 as a specific surface marker of CAF-2 highlights CD141$^+$ CAFs as potential therapeutic targets.

TGF-β reportedly activates CAFs and promotes tumor fibrosis in pancreatic cancer[58,59]. It also promotes epithelial-mesenchymal transition and exacerbates tumor invasiveness and metastasis[60]. Furthermore, TGF-β suppresses adoptive immune response by altering T cell differentiation toward Th2 rather than Th1, and affects innate immune response by facilitating M2 transition of macrophages[61–63]. These are considered as important reasons why PDAC rarely responds to the current immunotherapies. Although the relationship between the TGF-β signaling and PlGF signaling pathways has not been comprehensively examined, TGF-β reportedly increases PlGF expression, and PlGF activates the TGF-β signaling pathway in different cell types, but not CAFs[64,65]. Herein, we revealed that gemcitabine increased the number of CD141$^+$MHCII$^-$ CAF-2, which exhibits enhanced TGF-β signaling pathway, and Ate-Grab dramatically reduced CAF transition toward CAF-2 by blocking the PlGF effect. Our flow cytometry data illustrated that compared to gemcitabine treatment alone, Ate-Grab co-treatment decreased the expression of pSTAT3 in CAFs, which has reported essential for TGF-β-induced transcription[66–68], and also showed a tendency to decrease pSMAD2 and pSMAD3, which are representative indicators of TGF-β signaling activation[69,70].

Recent studies have demonstrated that PlGF blockade reduces desmoplasia via Akt-NF-kB signaling, enhancing chemotherapy efficacy in intrahepatic cholangiocarcinoma (ICC)[25]. These findings are consistent with our results in PDAC, wherein gemcitabine treatment significantly upregulated PlGF expression and Ate-Grab suppressed tumor desmoplasia. As previously discussed, since PD-L1 is expressed in ~40% of resected human ICC and is associated with poor prognosis, we anticipate that targeting both the PlGF and PD-L1 axes improve ICC treatment outcomes when compared to PlGF blockade alone[25]. Overall, it would be interesting to explore the efficacy of Ate-Grab in other cancer types.

Ate-Grab monotherapy also exerted anti-fibrotic and antitumor effects in the orthotopic PDAC model, implying that there may be certain baseline PlGF/VEGF expression, sufficient to activate CAFs and promote desmoplasia. Given that Ate-Grab alone inhibited collagen deposition and tumor growth, it would be important to evaluate whether T lymphocyte infiltration and activation are enhanced following Ate-Grab monotherapy. If this is the case, combinations of Ate-Grab with currently available blockers of immune checkpoints such as CTLA-4 may have great therapeutic potential in PDAC patients who do not benefit from immunotherapy.

Unlike the therapeutic efficacy of Ate-Grab + gemcitabine, atezolizumab + VEGF-Grab + gemcitabine did not exhibit a significant antitumor effect. This was noteworthy when considering that these combinations target the same pathways. We hypothesized that the main reason for this discrepant efficacy may lie beyond the effects on these signaling pathways. One possibility is that, in contrast to the

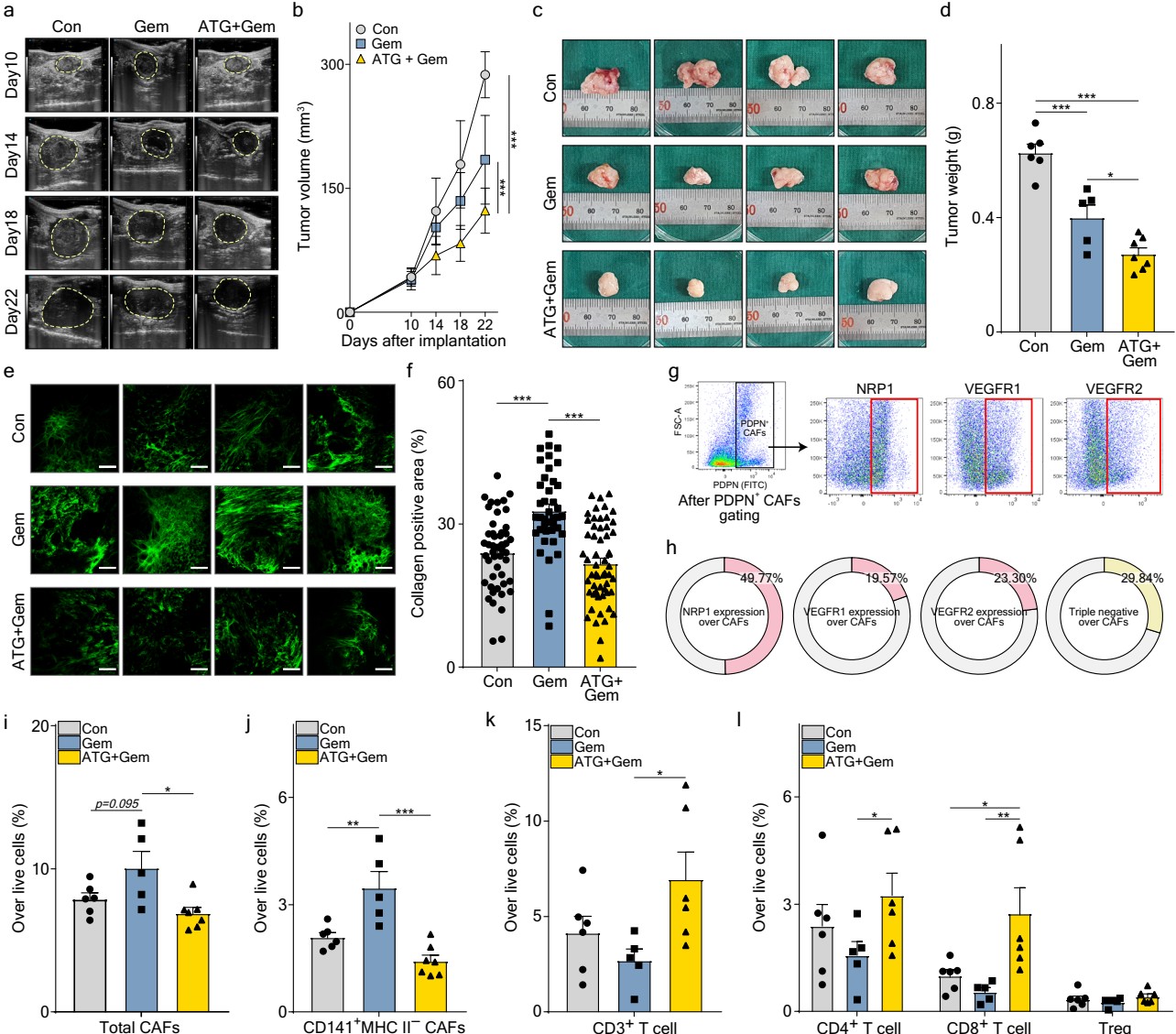

**Fig. 8 | Ate-Grab showed comparable therapeutic efficacy between *KRAS*-mutated murine PDAC and KRAS-wild type murine PDAC. a** Representative ultrasonographic images for orthotopic KPC tumor growth in vivo. Con, control; Gem, gemcitabine; ATG + Gem, Ate-Grab and gemcitabine cotreatment. **b** Tumor volumes compared across indicated treatment groups (Con, $n = 6$; Gem, $n = 5$; ATG + Gem, $n = 7$). ***$P < 0.001$; two-way ANOVA with Tukey's multiple comparisons. **c** Representative images of KPC tumors. Ruler scale, 1 mm. **d** Tumor weights compared across indicated treatment groups (Con, $n = 6$; Gem, $n = 5$; ATG + Gem, $n = 7$). Data are presented as the mean ± SEM. *$P < 0.05$ ($P = 0.030$), ***$P < 0.001$, one-way ANOVA with Tukey's multiple comparisons. **e** Representative SHG images of KPC tumors from each treatment group. Scale bars, 100 μm. **f** Average percentages of collagen⁺ area out of total area in tumors. Data from randomly selected fields of view were analyzed (Con, $n = 48$; Gem, $n = 40$; ATG + Gem, $n = 56$). Data are presented as the mean ± SEM. ***$P < 0.001$, one-way ANOVA with Tukey's multiple comparisons. **g** Flow cytometry gating strategy to evaluate VEGF-related receptor expression in KPC tumor-infiltrated CAFs. Cells gated in red box were considered

positive for the indicated receptors. Gating panel corresponds to FACS data in Fig. 8h. **h** Average percentages of NRP1-, VEGFR1-, or VEGFR2-expressing cells and triple negative cells for these receptors. Average percentages of **i** total CAFs (Gem vs ATG + Gem: $P = 0.011$) and **j** CD141⁺ MHC II⁻ CAFs (Con vs Gem: $P = 0.006$) in the KPC tumor microenvironment. Data from six mice from the control group, five mice from the gemcitabine group, and seven mice from the Ate-Grab+gemcitabine group were analyzed. Data are presented as the mean ± SEM. *$P < 0.05$, **$P < 0.01$, ***$P < 0.001$, one-way ANOVA with Tukey's multiple comparisons. Average percentages of **k** CD3⁺ T cells (Gem vs ATG + Gem: $P = 0.039$), **l** CD4⁺ T cells, CD8⁺ T cells and Tregs in the KPC tumor microenvironment of each treatment group (Con, $n = 6$; Gem, $n = 5$; ATG + Gem, $n = 6$). Data are presented as the mean ± SEM. *$P < 0.05$ (Gem vs ATG + Gem in CD4⁺ T cells: $P = 0.023$, Con vs ATG + Gem in CD8⁺ T cells: $P = 0.013$), **$P < 0.01$ (Gem vs ATG + Gem in CD8⁺ T cells: $P = 0.002$), two-way ANOVA with Tukey's multiple comparisons. Source data are provided as a Source Data file.

enhanced tumor delivery of Ate-Grab owing to its scFv portion, VEGF-Grab may not have been delivered to tumors as efficiently. This can be addressed through the development of another fusion protein between anti-PD-1 and VEGF-Grab, which would still block the PD-1/PD-L1 pathway like Ate-Grab, yet would exhibit different tissue-targeting efficiency owing to the scFv portion. Through this approach, we will be able to assess the importance of tumor specificity versus PD-1/PD-L1 pathway blockade in future studies.

This study has some limitations. For example, the concrete mechanisms of Ate-Grab in vivo were not clarified. Further, although we successfully confirmed that Ate-Grab was efficiently delivered to tumor tissues, the specific cell types directly targeted by Ate-Grab were unclear. Regarding this issue of the specific cell types, we have the following explanations: first, since Ate-Grab has a binding affinity for PD-L1, Ate-Grab can activate T cells and boost their infiltration into the TME. The activated T cells would secrete inflammatory cytokines and

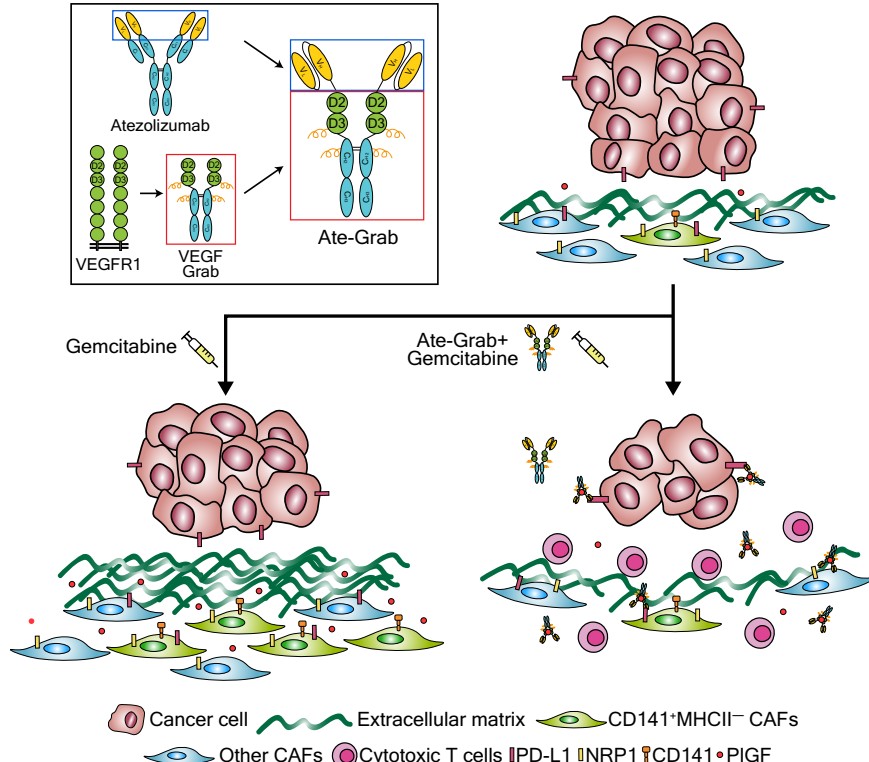

**Fig. 9 | Mode of action of Ate-Grab within the tumor microenvironment (TME).** The functional effects of Ate-Grab in the pancreatic tumor microenvironment are briefly summarized and illustrated. We developed a multi-paratopic-VEGF decoy receptor, dubbed "Ate-Grab," by fusing the single-chain Fv of atezolizumab (an anti-PD-L1 antibody) to the N-terminus of VEGF-Grab. Ate-Grab treatment blocked and sequestered PlGF/VEGF around NRP1+ CAFs (cancer-associated fibroblasts). Simultaneously, Ate-Grab dramatically reduced the number of CD141+ CAFs ("CAF-2"), activating collagen-secreting CAFs. This results in the promotion of cytotoxic T-cell expansion within the TME, thereby suppressing tumor growth compared to the control group.

modulate the TME as observed in our study with murine PDAC by affecting CAFs and myeloid cells, major components of the TME. Second, bi-specific molecules reportedly exhibit inherent functions beyond a simple dual-targeting effect. As the various cell types in the TME express PD-L1, there is a possibility that Ate-Grab blocks angiogenic molecules around PD-L1-expressing cells. To clearly address this question, further studies, including inhibition assays of specific cell types with genetic models, are required.

In summary, we demonstrated that chemotherapy induces PlGF/VEGF upregulation, activating CAFs to promote tumor desmoplasia in PDAC. PlGF/VEGF blockade via Ate-Grab is expected to improve treatment outcomes, especially when combined with standard chemotherapy. Furthermore, the identification of CD141+ CAFs as the major target of Ate-Grab provides a mechanistic basis for the development of efficient therapeutic strategies against desmoplastic cancers. Taken together, our findings may facilitate substantial advances for reprogramming the desmoplastic microenvironment in PDAC.

## Methods
### Study design
All experimental protocols were approved by the Institutional Animal Care and Use Committee (IACUC) of Seoul National University (Authorization No. SNU-200601-1-3) for animal studies, and Institutional Review Board (IRB) of Seoul National University Hospital (Authorization no. SNUH 1705-031-852) for human studies. For human studies, 20 of patients diagnosed with PDAC were included, and informed consent was obtained by all the participants.

The primary objective of this study was to investigate resistance mechanisms of tumors against chemotherapy to overcome its limited efficacy in pancreatic adenocarcinoma treatment. We designed and synthesized a multi-paratopic-VEGF decoy receptor (Ate-Grab), and

successfully evaluated the dual blocking effect of Ate-Grab using multiple cell-based assays. The in vivo effect of the protein drug candidate was verified using a mouse model of orthotopic PDAC, and the underlying mechanisms of the efficacy were explored through various methodologies including single-cell RNA sequencing. To ensure reproducibility, the mice were randomized. The experiments were carried out in a blinded manner using ear tag numbers. Each experiment was performed with multiple male seven to ten-week-old C57BL/6N mice ($n = 4$ to 8 per group). No data outliers were excluded in this study.

### Cell culture
Expi293F™ cells (Thermo, A14527) were maintained in Expi293™ expression medium (Thermo, A1435103) at 37 °C and 8% $CO_2$ with agitation at 125 rpm. HUVECs (Promocell, C-12203) were cultured in EGM-2 (Promocell, C-22110) supplemented with 1% penicillin/streptomycin on 1% gelatin pre-coated plates at 37 °C and 5% $CO_2$. Pan02 cell lines were obtained from NCI-Frederick Cancer DCTD Tumor Repository (#0509770), and were cultured in RPMI-1640 medium (Biowest, L0498) supplemented with 10% FBS, and KPC001 cell lines were cultured in DMEM (Biowest, L0103). Murine PDAC KPC001 cells (KrasG12D p53R172H/+) were kindly provided by Dr. Yves Boucher (Massachusetts General Hospital, Boston)[71].

### Orthotopic pancreatic cancer mouse model
Seven to ten-week-old wild-type male C57BL/6N mice were anesthetized with ketamine (100 mg/kg) and xylazine (10 mg/kg) via intraperitoneal injection. Hair on the left flank was removed, and the pancreas was carefully taken out using a tissue forceps following a 10-mm mid-axillary line incision just below the rib cage. Then, $5 \times 10^5$ cells of Pan02, Pan02-Luciferase, or KPC001 in 20 μL of RPMI-1640 (for Pan02)/ DMEM (for KPC001) and Matrigel (Corning, 354262) mixture

(1:1) were injected into the splenic lobe of the pancreas using an insulin syringe with a 29-gauge needle (BD, 320320).

## Abdominal ultrasound imaging and drug administration

Ultrasound imaging was performed to measure tumor size twice or thrice a week using the Vevo L3100 system (VisualSonic) with an MX550D probe (frequency: 40 MHz). Tumor-bearing mice were anesthetized via intraperitoneal injection of ketamine-xylazine solution. After shaving, the exposed skin was covered with ultrasound gel, and images were acquired upon applying the probe. Tumor mass in the abdominal cavity was defined as low echogenicity from the normal pancreas. The long diameter (LD) and short diameter (SD) were measured. Tumor volume was as: tumor volume = $(LD \times SD^2)/2$. All drugs were administered intraperitoneally every 3 or 3.5 days. According to IACUC, tumor size must not exceed 20 mm (2.0 cm) at the largest diameter in mice, and the maximal tumor size in this study was not exceeded.

## Collagen imaging using two-photon-generated second-harmonic generation (SHG)

Two days after the last administration of treatment, collagen deposition was observed using SHG microscopy at 840 nm on a custom-designed, video-rate laser-scanning microscope. We randomly selected five areas of tumor tissue and acquired a short video of SHG signals. We then averaged the noise over 30 frames using MATLAB to improve contrast and signal-to-noise ratio. The positive signal area was analyzed by a blinded observer using Image J software.

## Single-cell dissociation of tumors for fluorescence-activated cell sorting (FACS) analysis

On day 23 after tumor implantation, tumor tissues were resected, chopped into small pieces, and digested in a 37 °C and 5% $CO_2$ incubator for 20 min with fresh RPMI-1640 (Biowest, L0498) containing collagenase type IV (Worthington Biochemical, LS004189, 1 mg/mL), hyaluronidase (Sigma, H6254, 1 mg/mL), and DNase I (Sigma, DN25, 1.5 mg/mL). Digested tissues were minced, filtered through a 70-μm cell strainer, and used to prepare single-cell suspensions according to the manufacturer's instructions.

## Flow cytometry

Single-cell suspensions ($2 \times 10^6$ cells) from the tumor were incubated with a Live/Dead fixable blue dead cell stain kit (Thermo, L34962) for 30 min at 4 °C. Cells were stained with the following antibodies (1:200 dilutions) for 20 min at 4 °C in the dark: CD274 (BD, cat#745616, Clone: MIH5, BUV395), CD3 (Biolegend, cat#740268, Clone: 17A2, BUV395), CD19 (Biolegend, cat#115534, Clone: 6D5, PerCP-Cy5.5), B220 (Biolegend, cat#103224, Clone: RA3-6B2, APC-Cy7), NK1.1 (Biolegend, cat#108732, Clone: PK136, BV-421), CD45 (eBiosciences, cat#45-0451-82, Clone: 30-F11, PerCP-Cy5.5), CD86 (Biolegend, cat#105036, Clone: GL-1, PE-TR), CD4 (Biolegend, cat#100559, Clone: RM4-5, BV510), CD25 (Biolegend, cat#102049, Clone: PC61, BV711), CD49b (Biolegend, cat#108906, Clone: DX5, FITC), CD8a (Biolegend, cat#100708, Clone: 53-6.7, PE), Foxp3 (eBiosciences, cat#25-5773-82, Clone: FJK-16S, PE-Cy7), PD-1 (eBioscience, cat#17-9985-82, Clone: J43, APC), I-A/I-E (Biolegend, cat#107622, Clone: M5/114.15.2, Alexa 700), Epcam (Biolegend, cat#118225, Clone: G8.8, BV421), CD140a (Biolegend, cat#135906, Clone: APA5, PE), CD304 (Biolegend, cat#145218, Clone: 3E12, PE-TR), TNF-alpha (Biolegend, cat#506349, Clone: MP6-XT22, BV711), IL-2 (Biolegend, cat#503822, Clone: JES6-5H4,PerCP-Cy5.5), Granzyme B (eBioscience, cat#61-8898-82, Clone: NGZB, PE-TR), IL-4 (Biolegend, cat#504118, Clone: 11B11, PE-Cy7), Perforin (Biolegend, cat#154304, Clone: S16009A, APC), IL-17A (Biolegend, cat#506914, Clone: TC11-18H10.1, Alexa700), IFN-gamma (Biolegend, cat#505850, Clone: XMG1.2, APC-Cy7), alphaSMA (Novus Biologicals, cat#NBP2-34522AF647, Clone: 1A4/asm-1, Alexa Fluor 647), CD141 (eBioscience,

cat#25-1411-82, Clone: LS17-9, PE-Cy7). Phospho-SMAD2 (Bioss, Rabbit polyclonal, cat#bs-3420R). Phospho-SMAD3 (Bioss, Rabbit polyclonal, cat#bs-3425R), Phospho-STAT3 (Biolegend, cat#651010, Clone: 13A3-1, BV421). Stained samples were washed with FACS buffer (PBS with 0.5% BSA and 0.1% sodium azide), filtered using a 40-μm cell strainer, and analyzed using BD FACSymphony, LSR Fortessa, and Flowjo v10 software.

## CAF isolation and in vitro culture

On day 24 after tumor implantation, mice were anesthetized via intraperitoneal injection using ketamine-xylazine solution and sacrificed via cardiac puncture. Tumor tissues were then resected. After preparation of single-cell suspensions as per the above-described tissue digestion method, CAFs were isolated through FACS following the gating strategy described in Supplementary Fig. 1a. Sorted CAFs were seeded ($1 \times 10^5$ cells/500 μL) in 24-well plates and cultured in DMEM/F12 medium (Welgene, LM 002-01) supplemented with 20% FBS and 1% penicillin/streptomycin for 16 h at 37 °C and 5% $CO_2$. After 16 h, sequential serum deprivation was performed with 10% FBS for 6 h, 5% FBS for 6 h, and no FBS for 12 h. After sequential serum deprivation, the indicated proteins (PlGF, VEGFA) and atezolizumab or Ate-Grab were added and co-incubated for 6 h.

## qRT-PCR

Total RNA was extracted from cultured CAFs using a GeneAll Hybrid-RTM kit (Geneall, 305-101). The RNA concentration and quality were assessed using a NanoDrop spectrophotometer. High-quality RNA (200 ng to 1 μg) was reverse-transcribed to cDNA using the GoScriptTM reverse transcriptase kit (Promega, A5003). Following primer validation (Supplementary Table 1), SYBRTM Green PCR Master Mix kit was used to carry out amplification reactions on a Real-Time PCR System (Applied Biosystems). Expression was determined via the △△CT method, with *Gadph* used as a housekeeping gene.

## Tissue enzyme-linked immunosorbent assay (ELISA)

Pan02 tumor tissues were resected after five time-treatments of each drug (PBS, Gemcitabine). Resected tumor tissues were homogenized using D1000 benchmark homogenizer with RIPA buffer. VEGF-A and PlGF protein levels were quantified by ELISA kit according to the manufacturer's instructions (VEGF-A: biorbyt, orb437571, PlGF: biorbyt, Orb437465).

## Immunohistochemistry (IHC)

Paraffin-embedded, formalin-fixed tissue blocks were sectioned to 4 μm of thickness. Single IHC was performed for VEGFA and PlGF using the Ventana BenchMark XT Staining system. Briefly, IHC sections were baked and deparaffinized. Antigen retrieval was performed for 24 min at 100 °C using Cc1 solution (Ventana, #05279801001). Endogenous peroxidase was blocked with 3% hydrogen peroxide for 4 min at 37 °C. Primary antibody was applied, and the OptiView universal DAB kit was used for the enzyme-substrate reaction. After immunostaining, the slides were counterstained with hematoxylin, and the hue of hematoxylin was changed to a blue color using a bluing agent. Five random areas of tissue were analyzed for VEGFA and PlGF expression using Image J.

Double IHC was performed for α-SMA with either PD-L1, VEGFR1, VEGFR2, or NRP1. Deparaffinization, antigen retrieval, and peroxidase blockade were done as described above. Primary antibodies against PD-L1, VEGFR1, VEGFR2, or NRP1 were applied for 48, 32, 16, or 16 min, respectively. Polymer amplification was done using an OptiView HRP multimer for 8 min at 37 °C after linker (Optiview, HQ Universal Linker) was applied for 8 min at 37 °C. The OptiView DAB kit was used for the enzyme-substrate reaction. Anti-α-SMA primary antibody was then applied for 16 min at 37 °C. An UltraView universal AP Red detection kit was used for 8 min at 37 °C following the application of a multimer and enhancer. Hematoxylin staining was done as described above. To

assess the double-positive area, five random regions were analyzed via Image J. DAB and AP red color were distinguished through color deconvolution in Image J, and double staining area was measured through the color merge function.

All data are shown as the mean ± SEM, and statistical significance between groups was assessed via the two-tailed Student's t-tests. The following antibodies were used: anti-α-SMA (Polyclonal, Novus Biologicals, cat#NB300-978, 1:200), anti-NRP1 (EPR3113 clone, Abcam, cat#ab81321, 1:200), anti-VEGFR1 (Y103, Abcam, cat#ab32152, 1:250), anti-VEGFR2 (B309.4, Invitrogen, cat# MA5-15157, 1:4000), anti-PD-L1 (E1L3N, Cell Signaling, cat#13684, 1:200), anti-VEGFA (VG-1, Abcam, cat#ab1316, 5 µg/ml), anti-PlGF (polyclonal, Abcam, cat#ab196666, 1:50). IHC images were obtained using a Nikon Eclipse Ts2R microscope (Nikon, Japan).

### Expression and purification of recombinant proteins
Genes encoding the scFv of atezolizumab were cloned into the N-terminal of VEGF-Grab in the pcDNA3.1 vector using the EZ-Fusion™ HT Cloning Kit (Enzynomics, EZ015TS). Each cloned pcDNA 3.1 vector was amplified using DH5-alpha competent cells (Enzynomics, CP010) and purified using a plasmid purification kit (Qiagen, 12145). Purified vectors were transfected into Expi293FTM cells using the ExpiFectamineTM 293 transfection kit (Thermo, A14525). Transfected cells were cultured for 4 days in the presence of enhancers 1 and 2. VEGF-Grab or Ate-Grab in the supernatant were bound to Protein A resin and washed with PBS at 10 times the resin volume. Subsequently, they were neutralized with 1 M Tris-HCL (pH 8.0) and 250 mM NaCl, eluted with 100 mM glycine and 250 mM NaCl, pH 2.7, and purified via size exclusion chromatography.

### Solid-phase binding assays
Ninety-six-well plates (Sigma, Cat number #M9410) were pre-coated with hVEGFA165 (150 ng/mL) (R&D, Cat number #293-VE), PlGF (62.5 ng/mL) (R&D, Cat number #264-PGB/CF), human PD-L1 (1 µg/mL), mouse PD-L1 (1 µg/mL), human PD-1 (1 µg/mL), and mouse PD-1 (1 µg/mL). Various amounts (from 0.1 nmol/L to 10 µmol/L) of VEGF-Grab, Ate-Grab, and atezolizumab were then added to each well via a 1/2 dilution method. Molecules bound to the coated proteins were assessed through ELISA with a horseradish peroxidase (HRP)-conjugated anti-human FC antibody (Novus, NB7449). Absorbance was determined at 450 nm using a Thermo Scientific™ Varioskan™ Flash multimode reader (Product code MIB#5250030). Data were analyzed using GraphPad Prism 7.

### Inhibition of VEGFR2 signaling by VEGF-Grab and Ate-Grab
HUVECs were treated with 25 nM VEGF-grab or Ate-Grab for 15 min, followed by the addition of 1 nM hVEGFA$_{165}$ for 5 min. HUVECs were then washed with 1× PBS and lysed in RIPA buffer supplemented with a phosphatase inhibitor (Thermo Fisher, Cat number #49068450091) and a protease inhibitor cocktail (Sigma, Cat number #11836145001). Then, 75 µg of total protein was loaded on 10% SDS-Page gels and transferred onto PVDF membranes (Thermo, 88518) for western blot analysis.

### Migration assays
HUVECs were cultured on the culture inserts of u-dishes (ibidi, 81156) until 80% confluence. The culture inserts were subsequently removed to generate wound gaps. HUVECs were then incubated in EGM-2 (Promocell, C-22010) without SupplementMix (Promocell, C-39215) containing 5 nM VEGFA and 25 nM of VEGF-Grab or Ate-Grab for 24 h. Migrated cells within the wound were determined.

### Tube formation assay
HUVECs were plated at 10,000 cells/well in 96-well plates coated with growth factor-reduced Matrigel (Corning, 354230) and treated with 25 nM VEGF-Grab or Ate-Grab. After 10 min, 1 nM hVEGFA165 was added, and tube formation was assessed 6 h later.

### PD-1/PD-L1 blockade bioassay
The PD-1/PD-L1 blockade bioassay (Promega, J1250) is a bioluminescent cell-based assay used to measure the PD-1/PD-L1-blocking ability of molecules. The assay involves the co-culture of PD-1 effector cells and PD-L1 aAPC/CHO-K1 cells, wherein the PD-1/PD-L1 interaction inhibits TCR-mediated luminescence. When the interaction is disrupted, TCR activation induces luminescence, which can be detected using a luminometer.

### In vitro localization analysis
Pan02 cells were incubated with 200 nM Cy5.5-conjugated atezolizumab, VEGF-Grab (negative control), or Ate-Grab for 6 h at 37 °C. After incubation, Pan02 cells were washed three times with PBS, fixed in 4% paraformaldehyde for 20 min at room temperature, and counterstained with DAPI (VECTOR, H-1500-10). Images were acquired using a custom-designed video-rate laser-scanning microscope.

### In vivo drug distribution analysis (IVIS imaging)
In vivo fluorescence imaging (IVIS-200, Xenogen) was performed to track antibody distribution. Two weeks after implantation, orthotopic Pan02 tumor-bearing C57BL/6 mice were intraperitoneally administered Cy5.5-conjugated VEGF-Grab or Ate-Grab at 10 mg/kg. Twenty-four hours after injection, mice were anesthetized with isoflurane and imaged under the Cy5.5 excitation/emission channel after abdominal cavity exposure.

### Immunofluorescence
Resected mouse tumor tissues were fixed in 4% paraformaldehyde for 6 h at 4 °C, equilibrated in 30% sucrose overnight at 4 °C, embedded in O.C.T. compound (TissueTek), and 20-µm-thick cryosections were prepared. Sections were blocked with 10% goat serum in PBST (0.4% Triton X-100) for 1 h and immune-stained with the following primary antibodies: anti-CD31 (2H8, Merck, cat#MAB1398Z, 1:200), anti-PDGFR-β (APB5, Invitrogen, cat#14-1402-82, 1:200), and anti-NG2 (Polyclonal, Merck, cat#AB5320, 1:200) (overnight at 4 °C. The slides were then stained with TRITC- (Invitrogen, cat#A18894), AlexaFluor488- (Invitrogen, cat#A-11008), and AlexaFluor647- (Invitrogen, cat#A-21451) conjugated secondary antibodies (Invitrogen) for 1 h at room temperature and mounted with DAPI-containing Vectashield Mounting Medium (Vector Laboratories). The stained samples were visualized under a FV3000 confocal laser-scanning microscope.

### Perfusion assay
After five time-treatments of each drug (PBS, VEGF-Grab, Atezolizumab, and Ate-Grab), Pan02 tumor-bearing mice were sacrificed after intravenous injection of 2000kDa FITC-dextran (Sigma, FD2000S-100MG; 20 mg/kg) into retro-orbital space. Pan02 tumors were resected and fixed in 4% paraformaldehyde for 6 h at 4 °C, equilibrated in 30% sucrose overnight at 4 °C, embedded in O.C.T. compound (TissueTek), and 20-µm-thick cryosections were prepared. Sections were blocked with 10% goat serum in PBST (0.4% Triton X-100) for 1 h and stained with the AlexaFluor647-conjugated secondary antibodies following anti-CD31 antibody.

### Single-cell sorting, library preparation, and RNA sequencing for Pan02 tumor samples
Ten days after Pan02 ($5 \times 10^5$ cells) implantation, tumor-bearing mice were intraperitoneally treated with PBS (Con; $n = 5$), gemcitabine (Gem; $n = 5$) or a combination of Ate-Grab and gemcitabine (ATG + Gem; $n = 4$). Mice were sacrificed for single-cell RNA sequencing on day 24, 2 days after the fifth drug injection. Single-cell suspensions prepared from Pan02 samples were stained with a Live/Dead fixable blue

dead cell stain kit (Thermo, L34962) to sort live cells using a BD Aria III sorter (FACS sorting). Libraries were prepared using the Chromium controller according to the 10X Single Cell 3′ v3 protocol (10x Genomics). Briefly, the cell suspensions were diluted in nuclease-free water to obtain a cell count of 10,000. The cell suspension was mixed with master mix and loaded with Single-cell 3′ Gel Beads and Partitioning Oil into a Single-cell 3′ Chip. Transcripts from single cells were uniquely barcoded and reverse-transcribed within droplets. cDNA molecules were pooled, and the cDNA pool was subjected to end-repair, single "A" base addition, and adapter ligation. The products were then purified and enriched via PCR to create the final cDNA library. The purified libraries were quantified using qPCR according to the qPCR Quantification Protocol Guide (KAPA) and qualified using the Agilent Technologies 4200 TapeStation (Agilent Technologies). The libraries were sequenced on the HiSeq platform (Illumina) with a read length of 28 bp for read 1 (cell barcode and UMI), 8 bp index read (sample barcode), and 91 bp for read 2 (RNA read).

### scRNA-seq data processing

For 10X sequencing data, the Cell Ranger toolkit provided by 10X Genomics was applied to align reads and produce the gene-cell unique molecular identifier (UMI) matrix using the reference genome GRCm38. The number of features (genes) and UMIs were quantified for each cell. Seurat (v3.0.1) was used to create data objects from the cell ranger-processed outputs[72]. Features detected in fewer than three cells were excluded, and cells that expressed below 500 genes, over 7500 genes, or over 15% mitochondrial genes were removed. To integrate the three samples sequenced separately, individual Seurat objects were merged and normalized with the "NormalizeData" function, and technical batch effects were corrected by canonical correction analysis for an initial dimension reduction, using the "FindIntegrationAnchors" function of Seurat v3. To cluster single cells based on their gene expression pattern, an unsupervised graph-based clustering algorithm of Seurat v3 (v3.0.1) was used. The top DEGs were selected via the vst method, and principal component analysis (PCA) was then performed on 2000 DEGs. The "FindClusters" function on 20–50 PCs with a resolution of 0.4–1.5 was used to cluster cells on the UMAP plot. To determine the number of PCs, Seurat:ElbowPlot served as a reference to confirm the amount of variance represented by each PC, and the biological interpretation pf DEGs of each cluster was considered. Clusters of interest (CAF subsets, T cell & NK cell subsets, and Neutrophil subsets) were identified with marker genes and subset by using "SubsetData" with parameter "do.clean" set to true. The clustering procedures with the selection of top variable genes, scaling with UMI regression, PCA, and UMAP were repeated. All codes used for clustering, DEG sorting, and visualization are available at https://www.github.com/satijalab/seurat[72]. To calculate and project the RNA velocity, scVelo for stochastically modeling of transcriptional dynamics of splicing kinetics[73] was used. For each gene, a steady-state-ratio of unspliced and spliced mRNA counts was fitted. RNA velocities were then obtained as residuals from this ratio. For additional details, see the Supplementary Information. All codes used for calculating the RNA velocity and visualization are available at https://github.com/theislab/scvelo[73].

### Survival analysis and tissue-wide expression analysis

Human pancreatic cancer samples from GEO, EGA, and TCGA were used to evaluate the prognostic value of DEGs from CAF subclusters. Kaplan–Meier Plotter (www.kmplot.com) was utilized for survival analysis based on the gene signatures of populations of interest. With Kaplan–Meier Plotter[74], we performed overall survival analysis with 150 pure pancreatic ductal adenocarcinoma patient samples acquired from the TCGA repository (TCGA-PAAD), not restricted to gender, race, stage, grade, and mutation burden. DEGs used for survival analysis were endowed with the same weight, applied with MAS-5

algorithm-based normalization and second scaling normalization, and the mean expression of the genes was calculated. To sort pure PDAC patients, we excluded the following samples from the TCGA-PAAD dataset; did not arise from pancreas/Neuroendocrine/Acinar cell carcinoma/Intraductal papillary mucinous neoplasm/Undifferentiated/Systemic treatment given to the prior/other malignancy/Samples without tumor cellularity information. We also profiled the tissue-wide expression CAF-2 signatures in PAAD using Gepia2[53]. We used log2(TPM + 1) for log-scale, set 1 as the Log2FC cutoff, and 0.01 as the p value cutoff.

### Gene set enrichment analysis

Functional pathways associated with individual clusters were analyzed using the MSigDB Hallmark 2020 and BioPlanet 2019 gene set libraries and *Enrichr*[75]. The results were sorted using based on p value.

### 3-Dimensional quantification of collagen fibers in PanO2 tumor by IMARIS

SHG images were obtained from three different random spots per PanO2 tumor. Each spot consisted of 100 consecutive images in total, taken at 1 μm intervals along the z-axis. The images were used for 3D reconstruction with Imaris 9.3 software (Bitplane Inc.), and surface functions (surface grain size = 1 μm, automatic threshold for region growing) were applied for volume rendering. As a result, we obtained three 3D reconstructed regions, each with a 100 μm-thick layer of collagen per tumor, and further analyzed the images with IMARIS 9.3 software by measuring the volume, area, and number. For one spot, we measured volume and length of an individual fiber, counted the number of fibers (defined as "Total number"), and calculated total volume of the fibers (defined as "Total volume") and average volume and length of each fiber (defined as 'Average volume' and "Average length", respectively).

### Statistical analysis

All data are presented as the mean ± SEM, with the statistical significance between groups assessed via the indicated statistical methods using GraphPad Prism (v.7.0.4). A Tukey test was used to correct for multiple comparisons in one-way ANOVA and two-way ANOVA analysis. P values < 0.05 were considered statistically significant.

### Reporting summary

Further information on research design is available in the Nature Research Reporting Summary linked to this article.

## Data availability

The raw files of single-cell RNA sequencing data in this study have been deposited in the Gene Expression Omnibus database under accession code GSE189753. The TCGA-PAAD publicly available data used in this study are available in the GDC Data Portal under project name TCGA-PAAD. The remaining data are available within the Article, Supplementary Information or Source Data file. Source data are provided with this paper.

## Code availability

All custom-written code is available via GitHub (https://github.com/juhjeong/ATG; https://doi.org/10.5281/zenodo.7104075)[76].

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

## Acknowledgements

This work was funded by the National Research Foundation of Korea [grant 2020R1C1C1015062 to K.J.] and the Institute for Basic Science, Republic of Korea [grant IBS-R030-C1 to H.M.K.]. The authors are greatful to colleagues and Panolos Bioscience for their support and fruitful discussion.

## Author contributions

K.J. and H.M.K. designed and supervised the research; D.K.K., J.J., G.W.P., and S.W.J. performed the experiment; D.S.L. provided reagents; K.B.L. and J.Y.J. provided patient specimens; D.H. and D.Y.H. performed multi-omics analysis; D.K.K, J.J., H.M.K., and K.J. analyzed the data; and D.K.K., J.J., H.M.K., and K.J. wrote the paper.

## Competing interests

The authors declare no competing interests.
