## [Peer Review File · Nature Communications]

PD-L1-directed PIGF/VEGF blockade synergizes with chemotherapy by targeting CD141+ cancer-associated fibroblasts in pancreatic cancerREVIEWER COMMENTS

Reviewer #1 (Remarks to the Author): with expertise in pancreatic cancer, tumor microenvironment, chemotherapy, angiogenesis.

This interesting study investigates the contribution of PLGF-VEGFR1 signaling in CAFs in PDA. The authors provide evidence that PDA treated with chemotherapy increases the expression of PLGF/VEGF, which enhances the desmoplastic reaction in the tumor. In addition, they also suggest that CAFs are the major source of PD-L1 in PanO2 tumors. The authors developed a novel therapeutic agent, a fusion of a single-chain Fv of anti-PD-L1 Ab to a VEGFR1 decoy receptor. This construct termed, Ate-Grab, had anti-tumor and anti-fibrotic effects. It also increased the number of effector T cells in the tumor. In general, the article was well-written and easy to follow. However, there are a few challenges that should be addressed. Overall the study falls short of providing detailed mechanisms of action of Ate-Grab and the model is not robust enough to support all the claims. Below are more detailed comments:

1. This study relies on a single orthotopic mouse model (PanO2 cells). While this is a useful murine model of PDA there some concerns with studies that only rely on PanO2 orthotopic tumors. First, compared to GEMMs, orthotopic models typically display less desmoplasia, which may not accurately recapitulate CAF activity/ heterogeneity. Second, PanO2 is wildtype for Kras, which does not represent the majority of PDA tumors. It is strongly advised to replicate at least some of the work in a GEMM model or at least in a cell line derived from a GEMM.
2. Another concern regarding the model is that the majority of studies are performed with endpoints where tumor size in the control treated animals is quite modest (~0.3 g, Fig 1; 150 mm³, Fig 4 & Fig 5). The effect of therapy in a longer term experiment should be shown, especially if an immune-related mechanism is proposed.
3. The study validates reports that PLGF/VEGF contribute to fibrosis and that chemotherapy can enhance fibrosis in PDA. Thus, the novelty of the study is based on the development of Ate-Grab and the detailed analysis of the effect of Ate-Grab on the microenvironment of PanO2 tumors. Regarding the effect of Ate-Grab on CAFs the authors provide data that Ate-Grab mainly effects myCAF, which have been reported to have tumor-restricting functions. The effect of Ate-Grab should be discussed in this context as studies that have ablated myCAF or reduced myCAF activity have resulted in tumor progression.
4. The claim that CAF is the major source of PD-L1 in the TME is questionable and not consistent with other literatures that suggest cancer cells and myeloid cells are the major source. Therefore, the gating of Fig. 2I needs to be shown, and the expression of PD-L1 in the scRNA seq data also needs to be shown to prove CAF is the major source.
5. Fig. 4f and extended Fig. 5f (myCAF) are not significant, this is counter to the overall conclusions of the authors. Does tumor size impact these results? Also how did the drug affect iCAF and apCAF populations?
6. In the scRNA seq data, the expression of VEGFRs should be shown.
7. A no-chemo treatment control is lacking in the scRNA seq experiments. This clouds interpretation of the data, the caveats that this creates should be discussed.
8. Please compare the signatures of CAF subtypes in this study with previous published CAF scRNA seq data, to see if the CAFs found in this study are the known iCAF, myCAF apCAF or some new subtypes induced by gemcitabine treatment or just specific to the PanO2 orthotopic model.
9. How much of the tumor-inhibiting effects of Ate-Grab is immune related? A CD8 T cell ablation assay in vivo might be useful to discern the importance of the adaptive immune response.

Minor:

1. Fig. 1e, f should be validated through ELISA.

2. Fig. 1g is interesting. The figure legend should specify how CAFs were identified in the flow analysis. Are these CAFs Pdgfra or α SMA positive or was another marker used.

3. Extended Fig. 1d, the current gating may include mesenchymal cancer cells, it needs to be gated for PDGFR α or podoplanin+ cells.

Reviewer #2 (Remarks to the Author): with expertise in pancreatic cancer; system oncology

Here Kim and Jung present some interesting results indicating a role for PIGF/VEGF in regulating tumour desmoplasia and immune evasion and that combined targeting of PIGF/VEGF and PD-L1 in tumour bearing animals improves vessel function and gemcitabine efficiency.

Generally, the manuscript is interesting with intriguing data of potential translational relevance. However, several conclusions are overstated and based on data which are presented with limited clarity. Thus, some of the take-away messages need additional clarifications and experimentation. Further, some conclusions that indicate causative relationships are only supported by correlative observations which should be either re-written or supported by additional data.

Specifically:

Generally, the authors have not provided sufficient information about number of replicates (biological and technical), number of animals in experimental conditions, and, in most cases, individual data points are not shown. This absolutely needs to be corrected as the validity of the results and conclusion cannot be evaluated in sufficient detail without.

Figure 1:

The authors present data demonstrating an increased level of Collagen I/tumour fibrosis following Gemcitabine. Second Harmonics generation typically capture crosslinked fibrillar collagen, and correlates with tissue rigidity. Thus, SHG is not necessarily a broad measure of desmoplasia as a general phenomenon, although I agree it correlates. However, with some textual clarification I am happy with the data included. It would be interesting to determine whether the structure of the Collagen fibres is changed fx length and thickness (this is not a critical suggestion, but mostly out of curiosity).

What is less clear, and should be clarified, is the number of animals included in the study and the number of areas/size of areas included in Fig 1d. Moreover, individual data points should be shown. The concentration and dosing of gemcitabine isn't clear and should be described – this is particular important because some reports in the literature demonstrate gemcitabine sensitivity with the Pan02 cell line in vivo. Thus, is the major effect observed with Ate-Grab under gemcitabine limited conditions (which still may be interesting)

Figure 1e/f demonstrates mRNA expression level of PIGF and VEGF, which isn't the same as protein level and should be complemented with protein measurements (fx ELISA) to solidify this critical piece of data in the manuscript.

Fig 1h shows increased expression of myofibroblastic genes in primary CAFs treated with VEGF or PIGF in vitro. Its unclear how many times the experiment were repeated and whether these were individual independent experiments or technical repeats and how long were cells treated with recombinant proteins for? The authors should include additional target genes in their analysis of better defined myCAF, iCAF and apCAF markers as these are used interchangeably throughout the

manuscript as CAF markers. It is therefore necessary that the authors determine whether PIGF (and VEGF) induce a specific transcriptional subtype or a more general overall expression of CAF markers across all CAF subtypes.

Furthermore, the markers and gating strategy used for CAF isolation cannot exclude epithelial cells that have undergone EMT (low EPCAM) and only captures a minor population of CAFs as PDGFRa and aSMA expression is commonly inversely related (Elyada et al Cancer Discovery 2019, Dominguez et al Cancer Disc 2000 and Hutton et al Cancer Cell 2021). At the very least the authors should verify the absence of tumour cells in their 'CAF' population and also clarify in their text/legend that they are referring to subset of CAFs defined by PDGFRa expression.

Finally, the data suggests that Gemcitabine driving desmoplasia through PIGF signalling in CAFs. This could be due to a change in gene expression or increased abundance of CAFs. This could be tested through IHC staining for CAFs and RT-qPCR on isolated CAFs from the tumours.

Figure 2:

It's unclear whether the patients analysed had been subjected to chemotherapy and whether patients in Sub4 and Sub6 have been subjected to similar treatment regime. Also, the authors compare patient subsets based on best and worst prognosis, however for the purpose of interrogating tumour desmoplasia it would be worthwhile to also compare the subsets with most and least amount of desmoplasia (fx as determined by tumour cellularity).

The representation of the flow cytometry analysis of PD-L1 levels across tumour and stromal cells is dependent on the cell viability, where different cell types are more or less viable. The authors should show the biaxial plots in the supplemental information. It is also not clear which markers were used for cell identification, whether the analysis was done in one experiment or in different experiments and the number of repeats?

Fig3:

For most experiments it's unclear how many biological and technical repeats were included.

Fig3j, as for Fig 1, additional markers of myCAF, iCAF and apCAF should be included and purity of CAFs should be validated. It's unclear why a comparison was made between atezolizumab and Ate-Grab rather than VEGF-Grab and Ate-Grab?

Fig4:

It's unclear at which point tumours were treated from and number of animals per treatment group

The authors claim that Ate-Grab reduce myCAFs, however the gating used in this experiment does not support this conclusion. myCAFs are generally low for PDGFRa (as described above), thus pre-gating for PDGFRa^{pos} CAFs excludes myCAFs and thus the authors have already selected for a CAF subset prior to their analysis. Furthermore, EPCAM will only exclude epithelial and not mesenchymal tumour cells from their analysis, which can cause inference in the analysis. The authors should re-do the experiment where total CAFs are then analysed by markers for myCAF/iCAF and apCAF as well as RNA expression (qPCR perfectly fine) should be used to validate this observation.

The change in vessel formation and function is based solely on PDGFRb overlay with CD31, which doesn't add functional insight. The authors should demonstrate increased perfusion such as by a high molecular dextran.

Fig 5:

As noted above, FACS gating strategy should be clarified, n numbers displayed and individual data points displayed. If similar strategy for CAF analysis were used as in previous figures the authors cannot distinguish between myCAF, iCAF and apCAF. Moreover, the authors cannot exclude mesenchymal tumour cells are included in this analysis. This should be backed up by improved FACS, IHC and gene expression analysis.

The decrease in total CAF numbers is notable (Figure 5d). This is a possible point of concern as increased vascularisation and CAF depletion in other studies (Rhim and Stager Cancer Cell 2014, Ozdemir and Kalluri Cancer Cell 2014) have shown this is correlated with increased metastasis and poor overall survival. The authors should collect and analyse livers for metastatic seeding.

The authors demonstrate decreased tumour volume and hypothesise this is due to increased anti-tumour immunity. However, there is no data to support this is due to decrease in tumour cell abundance and increased killing of tumour cells by immune cells. Additional data to test this hypothesis are needed.

Fig 6:

The scRNAseq leaves a number of critical questions open:

It's curious why so few endothelial cells are picked up when the authors in previous figures demonstrate an 'normalisation' of the vasculature which would be expected to improve therapeutic delivery. At the very least I would anticipate some CD31 endothelial cells would have been identified.

From the methods and text, it appears that the authors use only one gene to identify individual cell types and subsets? This can be misleading and additional genes should be used to verify cell identity to avoid mis-annotation.

As CAFs, endothelial cells (vascular and lymphatic) as well as mesenchymal tumour cells can easily appear similar by scRNAseq, the authors should assign these population with much greater care. Its also curious that only few tumour cells are identified in the scRNA analysis – which is a possible concern as 'CAF's' could originate from epithelial cells and thus be a mis-classification. Alternatively, most tumour cells are dead at the time of analysis and therefore raise a question to the relevance of looking at this particular timepoint?

Its not clear how CAF1-3 were assigned and whether these correspond to previously identified CAF subset (see Elyada et al Cancer Discovery 2019, Dominguez et al Cancer Discovery 2000). Also, the authors should verify that these CAF populations can be identified in other models of PDA such as by re-analysing the afore mentioned papers. This is important to confirm the general observation of the CAF subset identified.

The survival analysis with CAF2 signature is potentially confounded. If individual genes from the CAF2 signature are also expressed in other cell types how can the authors conclude that it is the CAF2 subset that is correlated with patient survival and not changes in other cellular sources? This analysis should also take tumour cellularity into account as the authors may otherwise compare tumour samples with different tumour cellularity. The authors should also annotate the subtype of the tumours to test whether then CAF signature is related to different tumour transcriptional subtypes (eg Collison et al Nat Med 2011, Bailey et al Nature 2016, Moffit et al Nat Genet 2015)

For comparison the authors should include a comparison with CAF1 and 3 transcriptional signatures on patient survival.

The FACS gating in 6f is not clear and should be shown in supplemental data

General edits:

The authors state that "the desmoplastic stroma consists mainly of CAFs..." (line 47) Although CAFs certainly are very abundant, cell quantification by tissue disaggregation is depending on equal liberation of all cell types (which I don't think has been demonstrated to be the case). This also doesn't seem to be the observation in the authors scRNAseq and in other reports (Elyada et al and Dominguez et al). In situ studies, such as IHC, IF etc, are more limited in the number of cells that can be enumerated and cellular composition is likely to change throughout a tumour. Also, I am not aware of studies that have enumerated stromal cells across several tumours using consecutive slides. As such, unless the authors can provide a more convincing reference I think they should change their

statement, perhaps just referencing that CAFs are an abundant cell type in the microenvironment...

L28/29: The authors write "Ate-Grab... target PD-L1 expressing CAFs". There is no evidence in the manuscript that this is a case. While it's a possible explanation this should be part of the discussion and should be presented as one of several possible explanations.

L32/33: The authors write: "the CD141+ CAF population, as responsible for the therapeutic effect of Ate-Grab". Another overstatement in the manuscript which there is no evidence for. The authors can show that the level of CD141+ CAFs are decreased in Ate-Grab + Gem treated tumours in comparison with Gem treated tumours, but there is no evidence to support the presented conclusion. This should be removed or supported by additional evidence.

Reviewer #3 (Remarks to the Author): with expertise in pancreatic cancer; cancer associated fibroblasts

In this manuscript by Kim, Jeong and colleagues, the authors investigate associations among chemotherapy treatment, growth factor signaling, collagen deposition, and response to immune checkpoint blockade in pancreatic cancer. In light of the very poor prognosis for pancreatic cancer patients and the broad ineffectiveness of chemotherapy and ICB within this patient population, mechanistic studies highlighting signaling nodes for therapeutic intervention to foster ICB efficacy in this disease setting is highly significant and represents an important and timely goal for the field. Using pancreatic cancer mouse models and clinical specimens, the authors here test the overarching hypothesis that gemcitabine treatment leads to induction of growth factors including placental growth factor and VEGF-A, which subsequently act on CAFs to enhance collagen deposition. Evidence in support of these connections in the manuscript are rather tenuous; for example, the impact of gemcitabine on collagen I deposition in vivo appears very modest and is of questionable relevance to the therapeutic preclinical studies later in the manuscript. The authors go on to develop and test an antibody called Ate-Grab, building on a prior and somewhat similar molecule but here with the ability to inhibit PD-1/PD-L1 signaling as well as VEGF-A and PIGF signaling. They show efficacy of this antibody combined with gemcitabine in reducing tumor growth in vivo. This is an interesting and novel therapeutic approach, with additional novelty stemming from the focus on PIGF, which has been subject to rather little study in pancreatic cancer. However, effects of VEGF/PIGF inhibition by Ate-Grab in vivo are modest and only shown in a single mouse model which bears questionable relevance to human pancreatic cancer. Addressing the comments below would help to strengthen the authors central claims.

Specific comments:

1. In vivo studies here are limited to transplant experiments using the Pan02 cell line, which is unusual in that it harbors wild-type KRAS while pancreatic tumors are almost always KRAS-mutant. In light of this and the increasingly appreciated connection between oncogenic KRAS signaling and regulation of the immune microenvironment, key experiments should be repeated in an independent model harboring mutant KRAS.
2. The results shown in ED Figure 1a-c are nice, but a change in CAF phenotype should be assessed (i.e., analysis of pSMAD and pSTAT3 within the CAF compartment) beyond just analyzing their abundance.
3. The clinical data presented in Figure 2 are very nice but could be further developed to better support the authors' model. Is there a correlation between collagen abundance and PIGF levels? Is prior gemcitabine treatment status known for these patients, and if so, do this treatment influence collagen and/or PIGF abundance? These analyses would help tie these clinical results to the preclinical data in Figure 1.
4. Figure 3 nicely demonstrates that Ate-Grab inhibits VEGF-A, PIGF, and PD-1/PD-L1 signaling, but what about specificity? Genetic inhibition experiments would be helpful to address this question.
5. CAF phenotypes in response to Ate-Grab should be analyzed to accompany the results presented in Figure 4 (p-STAT3, p-SMAD), which should be doable using tumor tissues already obtained from

prior experiments.

6. In Figure 4g,h and ED Figure 4b,c, PDGFRb is insufficient to define pericytes, and the authors should co-stain for a more specific pericyte marker such as RGS5 or NG2.

7. The conclusion on lines 199-201 is not supported by the data. If the authors wish to make claims about vessel normalization in response to Ate-Grab treatment, functional perfusion experiments are needed.

8. In Figure 4f, how were myCAFs identified/defined?

9. It is important to further assess the CAF-2 population displayed in Figure 6 to confirm that these are CAFs as opposed to PDAC cells with a mesenchymal phenotype. Does this cell population express other, well-established CAF markers like Acta2, Fap, and Pdpn?

10. The report of CD141 expression on CAFs is quite surprising and should be further validated. Figure 6g makes it seem that nearly all PDGFRa+ CAFs express CD141. Can this be demonstrated in tumor tissues by co-staining for CD141 and a CAF marker by IHC?

11. (Minor) Are the results in Figure 1e,f from tumor homogenates? Or from PDAC cells in vitro? This should be stated in the legend and/or results section.

Reviewer #4 (Remarks to the Author): with expertise in scRNAseq, cancer associated fibroblasts

Summary

Kim et al. describe a PIGF/VEGF - PIGF/VEGF receptor axis that induces collagen deposition in pancreatic cancer via cancer-associated fibroblasts. They develop a multi-paratopic VEGF decoy receptor (Ate-Grab) to target PIGF/VEGF within the PD-L1-enriched TME and investigate its impact on the cancer-associated fibroblast landscape in the orthotopic Pan02 tumor model. The authors observe anti-tumor/anti-fibrotic effects and identify a CD141+ CAF population from single-cell RNA sequencing data, which is proposed to be responsible for the therapeutic effect of the Ate-Grab. The manuscript is overall well written and the majority of conclusions are supported by the data. The manuscript is of interest to the CAF research field, but would benefit from additional experiments/analyses/clarifications I am outlining below.

Major

1) It is quite important to understand the baseline expression phenotypes in the single-cell experiment. How does CAF heterogeneity look like in untreated animals with orthotopically implanted tumors?

2) I'm confused by the statement in the discussion: "Of note, the characteristics of CD141+ CAFs reported herein are generally different from those previously described for myCAFs. That is, CD141+ CAFs secrete various factors, including INHBA, which can activate other CAF populations to further induce desmoplasia." Inhba is expressed by myCAFs in human patients and in murine PDAC models (Elyada et al., 2019, CD). In order to properly align the subsets identified in this manuscript with the CAF subtypes described in the literature, I would like to ask the authors to perform a more rigorous comparison between their subtypes and the subtypes identified in other studies. It will be important to understand the CAF heterogeneity in the orthotopic model used by the authors compared to heterogeneity in GEMM models without any treatment, and then relate these findings to the changes observed with Gem alone and ATG + Gem.

3) CAF-1 and CAF-2 look like they exist on a spectrum rather representing two clear distinct subsets of cells. This suggests they differ in their activation programs and can transition between states. The increase in CAF-1 in ATG + Gem seems mostly driven by increased INF-signaling into CAFs, which can induce expression of Gbps, antigen presentation machinery, etc. That makes interpretation slightly more complicated, as the results could be mostly a readout of increased infiltration with cytotoxic T cells that's described by the authors (thus shifting CAFs from CAF2 into CAF1

phenotypes). To understand the relationships between CAF-1 and CAF-2 better, I would like to suggest the authors perform RNA velocity (or) non-velocity-based pseudo time reconstruction analysis to understand program activity associated with transitions between activation states. How does treatment affect the transitions?

4) Given that PD-L1 is not only expressed by CAFs in the TME, as shown in figure 2I, I would suggest the authors at least use their single-cell dataset to investigate a potential direct (or indirect) effect of Ate-Grabon myeloid cells.

5) The authors focus strongly on the PIGF/VEGF - PIGF/VEGF axis when speaking about tumor fibrosis in PDAC. I would like to suggest to discuss this axis in the context of TGF β signaling at least in the discussion section, given the prominent role of TGF β signaling in pancreatic cancer, tumor fibrosis, and cancer immunotherapy response.

6) The authors cite their unpublished work in reference 37 to establish six PDAC subtypes. Given that the cited manuscript has not been formally reviewed, relying on the proposed subtypes (with limited ways to understand the characteristics of those proposed subtypes for the reader of this manuscript) for survival associations seems not the most desirable strategy to me.

In order avoid relying on the subtypes, I suggest that the authors directly analyze the impact of the expression/%area of PIGF/VEGF-A ligands as well as their receptors on patient survival instead of using the subtypes as an intermediate tool.

7) There is no direct evidence for CD141+ CAFs being “responsible for the therapeutic effects of Ate-Grab”, only an association (of these CAFs being reduced upon ATG + Gem). I would like to suggest rewording these statements to reflect the lack of direct evidence (as in: Does depletion of these CAFs have the same therapeutic effect?).

Minor

- It is unclear to me how myCAF β s were defined in figure 4f and 5d (legend does not explain figure for 5d).

- “and batch effects were corrected using the “FindIntegrationAnchors” function.” The authors should define what the batches were and what effect they aimed to correct for. How was the experiment designed?

- “The top DEGs were selected via the vst method, and principal component analysis (PCA) was then performed on 2000 DEGs. We used the “FindClusters” function on 10–30 PCs with a resolution of 0.4–1.5 to cluster cells on the UMAP plot. “How did the authors evaluate which of the results generated should be used for downstream analysis? Why were a certain number of PCs chosen? Or a particular clustering resolution?”

- “Gene set enrichment analysis revealed that epithelial-mesenchymal transition (EMT), which is involved in pathological fibrosis 53, was enriched in the CAF-2 subset (Extended Data Fig. 6f).” Given these are mesenchymal cells, the observation of EMT being upregulated does not seem surprising.

- “Human pancreatic cancer samples from GEO, EGA, and TCGA were used to evaluate the prognostic value of DEGs from CAF subclusters. Kaplan-Meier Plotter was used for survival analysis based on gene signature expression.” Please provide much more detail regarding the data analysis and sources, such as normalization, how signature scores were calculated, if certain patients’ samples were excluded, ...

- In their TCGA survival analysis, the authors decide to use the top40 upregulated genes from their mouse model and perform survival analysis in the TCGA PAAD dataset (how was a score for each

penitent calculated?). I would strongly suggest to exclude the PNET samples in the PAAD TCGA cohort from this analysis. Furthermore, I would like to ask the authors to make a statement about the specificity of their markers. Markers were identified comparing expression within CAFs, but not to other cell types in the TME. So how specific are the genes they have chosen?

- “These results suggested that excessive ECM production by CAFs confers chemotherapy resistance in PDAC.” The authors describe an association, but claim causality. I would recommend to rephrase this, unless this statement can be supported by experimental evidence, such as depletion of (a subset) of CAFs.

Point-by-Point Response

We are grateful to the reviewers for their careful evaluation of our manuscript and for constructive suggestions. In addition to revising the text, we have performed a series of experiments to address the reviewers' concerns. These new results have been incorporated into the revised manuscript. In the text below, the reviewers' comments are in italics and our responses and descriptions of the changes made in the manuscript are in bold blue typeface.

Reviewer #1 (Remarks to the Author): with expertise in pancreatic cancer, tumor microenvironment, chemotherapy, angiogenesis.

This interesting study investigates the contribution of PLGF-VEGFR1 signaling in CAFs in PDA. The authors provide evidence that PDA treated with chemotherapy increases the expression of PLGF/VEGF, which enhances the desmoplastic reaction in the tumor. In addition, they also suggest that CAFs are the major source of PD-L1 in Pan02 tumors. The authors developed a novel therapeutic agent, a fusion of a single-chain Fv of anti-PD-L1 Ab to a VEGFR1 decoy receptor. This construct termed, Ate-Grab, had anti-tumor and anti-fibrotic effects. It also increased the number of effector T cells in the tumor. In general, the article was well-written and easy to follow. However, there are a few challenges that should be addressed. Overall the study falls short of providing detailed mechanisms of action of Ate-Grab and the model is not robust enough to support all the claims. Below are more detailed comments:

We appreciate the reviewer's endorsement of our work. We hope that our revisions have assuaged any remaining concerns.

1. This study relies on a single orthotopic mouse model (Pan02 cells). While this is a useful murine model of PDA there some concerns with studies that only rely on Pan02 orthotopic tumors. First, compared to GEMMs, orthotopic models typically display less desmoplasia, which may not accurately recapitulate CAF activity/heterogeneity. Second, Pan02 is wildtype for Kras, which does not represent the majority of PDA tumors. It is strongly advised to replicate at least some of the work in a GEMM model or at least in a cell line derived from a GEMM.

We appreciate the reviewer's insightful suggestion. To address this concern, we performed additional experiments using the KPC001 cell line (Pdx1-Cre, lox-stop-lox-KrasG12D/+, lox-stop-lox-tp53R172H/+). A murine pancreatic orthotopic model was produced using the KPC001 cells and mice were administered with gemcitabine or Ate-Grab+gemcitabine (Supplementary Fig. 9b). Further, we confirmed both anti-tumor and anti-fibrotic effects of Ate-Grab, comparable to those of the KRAS-wild type Pan02 tumor as shown below.

Figure 8

Figure 8 | (a) Representative ultrasonographic images for measuring orthotopic KPC tumor growth *in vivo*. (b) Tumor volumes measured by ultrasonography and compared across indicated treatment groups (Con, n=6; Gem, n=5; ATG+Gem, n=7). Con, control; Gem, gemcitabine; ATG+Gem, Ate-Grab and gemcitabine cotreatment. ***P<0.001; two-way ANOVA. (c) Representative images of KPC tumors from indicated treatment groups. Ruler scale, 1 mm. (d) Tumor weights measured and compared across indicated treatment groups (Con, n=6; Gem, n=5; ATG+Gem, n=7). Data are presented as the mean ± SEM. *P<0.05, ***P<0.001, one-way ANOVA. (e) Representative SHG images of KPC tumors from each treatment group. Scale bars, 100 μm. (f) Average percentages of collagen+ area out of total area in tumors from each treatment group. Data from randomly selected fields of view were analyzed (Con, n=48; Gem, n=40; ATG+Gem, n=56). Data are presented as the mean ± SEM. ***P<0.001, one-way ANOVA. (g) Flow cytometry gating strategy to evaluate VEGF-related receptor expression in KPC tumor-infiltrated CAFs. Cells gated in red box were considered positive for the indicated receptors. (h) Average percentages of NRP1-, VEGFR1-, or VEGFR2-expressing cells and triple negative cells for these receptors over total KPC tumor-infiltrated CAFs. (i, j) Average percentages of (i) total CAFs and (j) CD141⁺ MHC II⁻ CAFs in the KPC tumor microenvironment of each treatment group, measured by flow cytometry. Data from six mice from the control group, five mice from the gemcitabine group, and seven mice from the Ate-Grab+gemcitabine group were analyzed. (k, l) Average percentages of (k) CD3⁺ T cells, (l) CD4⁺ T cells, CD8⁺ T cells and Tregs in the KPC tumor microenvironment of each treatment group, measured by flow cytometry (Con, n=6; Gem, n=5; ATG+Gem, n=6). Data are presented as the mean ± SEM. *P<0.05, **P<0.01, ***P<0.001, one-way ANOVA.

Supplementary Figure 9 | (b) Experimental treatment scheme. Murine orthotopic pancreatic models were generated by implanting KPC001 cells (5×10^5 cells/mouse) into the pancreas of 7–10-week-old C57BL/6 mice. When the tumor volume reached 50–100 mm³, tumor-bearing mice were intraperitoneally treated with either PBS (control), gemcitabine (50 mg/kg, every 3.5 days), or gemcitabine with Ate-Grab (10 mg/kg, every 3.5 days). **(c)** Flow cytometry gating strategy to evaluate PDPN⁺ CAF in KPC tumor and CAF subsets defined by MHC-II and CD141 expression. **(d)** Average percentages of CD141⁺MHC-II⁻ cells in total CAFs were measured and compared across indicated treatment groups (Con, n=6; Gem, n=5; ATG+Gem, n=7). Data are presented as the mean \pm SEM. **P<0.01; one-way ANOVA. **(e)** Average percentages of CAF subsets in total tumor-infiltrating live cells were measured and compared across indicated treatment groups (Con, n=6; Gem, n=5; ATG+Gem, n=7). Data are presented as the mean \pm SEM; one-way ANOVA. **(f)** Average percentages of B cells and NK cells in total tumor-infiltrating live cells were measured and compared across indicated treatment groups (Con, n=6; Gem, n=5; ATG+Gem, n=6). Data are presented as the mean \pm SEM; one-way ANOVA.

We have further detailed all related results in the Results section as below. (p. 17, highlighted in yellow; Fig. 8a-I; Supplementary Fig. 9b-f)

“KRAS mutation accounts for more than 90% of all human PDAC cases⁵⁶. As Pan02 tumor is a KRAS-wild type⁵⁷, we verified the therapeutic efficacy of Ate-Grab in KRAS-mutated murine PDAC. Orthotopic murine pancreatic cancer models were generated using KPC001 cell line derived from GEMM (Pdx1-Cre, lox-stop-lox-KrasG12D/+, lox-stop-lox-tp53R172H/+)⁵⁸, and the mice were divided into three experimental groups based on the different treatment regimens (Supplementary Fig. 9b). Similar to the Pan02 tumor, KPC001 tumor volumes and weights were significantly lower in the Ate-Grab+gemcitabine treatment group than in the gemcitabine and untreated (control) groups (Fig. 8a-d). In addition, we validated that gemcitabine induced tumor fibrosis, which was effectively suppressed by co-treatment with Ate-Grab (Fig. 8e, f). With the

expression of VEGF-related receptors validated in KPC001 tumor-infiltrated CAFs (Fig. 8g, h; Supplementary Fig. 9c), flow cytometry analysis confirmed that gemcitabine treatment increased total CAF% in the TME and co-treatment with Ate-Grab significantly decreased CAFs (Fig. 8i). Notably, further analysis with representative markers of CAF subsets revealed that CD141⁺ MHCII⁻ CAFs, defined as CAF-2, were dramatically diminished in response to Ate-Grab co-treatment, while other CAF subsets showed no statistically significant difference between treatments (Fig. 8j; Supplementary Fig. 9c-e). Furthermore, higher infiltration of CD3⁺ T cells, especially CD8⁺ T cells, was observed in response to Ate-Grab and gemcitabine co-treatment, while no significant difference was identified in B and NK cell populations (Fig. 8k, l; Supplementary Fig. 9f). Overall, validation analysis with KPC001 orthotopic murine PDAC indicated that Ate-Grab has a comparable therapeutic effect on KRAS-mutated PDAC as that on KRAS-wild type PDAC.”

2. Another concern regarding the model is that the majority of studies are performed with endpoints where tumor size in the control treated animals is quite modest (~0.3 g, Fig 1; 150 mm³, Fig 4 & Fig 5). The effect of therapy in a longer term experiment should be shown, especially if an immune-related mechanism is proposed.

We appreciate the reviewer's valuable suggestion. Simultaneously addressing comment 9, we performed an additional experiment. Compared with the original treatment schedule (started on Day 10 after tumor implantation with five treatments every three days), we adjusted the treatment schedule to track tumor growth for a longer period of time (Fig. 7a). Even after a longer observation, we successfully validated drug efficacy. Regarding the CD8⁺ T cell ablation assay to test whether the drug efficacy is mediated by immunity, we validated that the *in vivo* treatment of anti-CD8 antibody significantly inhibited the anti-tumor effects which was accelerated by Ate-Grab+gemcitabine treatment. In the Ate-Grab+gemcitabine+anti-CD8 antibody-treated group, both volumes and weights of the tumors increased to similar levels as in the control group (Fig. 7b-e), while tumor fibrosis and CAF population % were partially increased, but not significantly (Fig. 7f-g; Supplementary Fig. 9a).

We have now elucidated all of these results in the Results section. (p. 16, highlighted in yellow; Fig. 7a-j, Supplementary Fig. 9a)

“Based on the increased intratumoral CD8⁺ T cells and their cytotoxic features observed with Ate-Grab+gemcitabine combination therapy, we further determined whether the anti-tumor effect of Ate-Grab is directly mediated by immune-related mechanisms. An anti-CD8 neutralizing antibody was co-administered with Ate-Grab+gemcitabine (Fig. 7a), causing a substantial inhibition of the anti-tumor effects that were improved in response to the Ate-Grab+gemcitabine regimen (Fig. 7b-e). Specifically, in the Ate-Grab+gemcitabine+anti-CD8 antibody-treated group, tumor volume and weight increased to the levels of the control (Fig. 7b-e). Also, 3D reconstructed images of tumor collagen structure demonstrated that co-treatment of anti-CD8 antibody with Ate-Grab+gemcitabine did not significantly reverse the anti-fibrotic effect exerted by Ate-Grab, while individual collagen fiber volume and length did not change either (Fig. 7f, g; Supplementary Fig. 9a). Flow cytometry analysis also showed that co-treatment of Ate-Grab+gemcitabine with anti-CD8 antibody does not significantly affect total or CD141⁺ CAF population, while dramatically depleting CD8⁺ T cells and also reducing the number of CD4⁺ T cells *in vivo* (Fig. 7h-j). These results demonstrate that the tumor-inhibiting effect of Ate-Grab is mediated by CD8⁺ T cells in the TME.”

Figure 7

Figure 7 | (a) Experimental treatment scheme. Murine orthotopic pancreatic models were generated by implanting Pan02-Luciferase (5×10^5 cells/mouse) into the pancreas of 7–10-week-old C57BL/6 mice. When the tumor volume reached 50–100 mm³, tumor-bearing mice were intraperitoneally treated with either PBS (control), gemcitabine (50 mg/kg, every 3.5 days), gemcitabine with Ate-Grab (10 mg/kg, every 3.5 days), or gemcitabine with Ate-Grab and anti-CD8 antibody (5 mg/kg, once a week). **(b)** Representative ultrasonographic images for measuring orthotopic Pan02-Luciferase tumor growth *in vivo*. **(c)** Tumor volumes measured by ultrasonography and compared across indicated treatment groups (Con, n=6; Gem, n=7; ATG+Gem, n=8; ATG+Gem+α-CD8, n=6). Con, control; Gem, gemcitabine; ATG+Gem, Ate-Grab and gemcitabine co-treatment; ATG+Gem+α-CD8, Ate-Grab, gemcitabine, and α-CD8a antibody co-treatment. ***P<0.001, two-way ANOVA. **(d)** Representative images of KPC tumors from indicated treatment groups. Ruler scale, 1 mm. **(e)** Tumor weights measured and compared across indicated treatment groups (Con, n=6; Gem, n=7; ATG+Gem, n=8; ATG+Gem+α-CD8, n=6). Data are presented as the mean ± SEM. *P<0.05, **P<0.01, one-way ANOVA. **(f)** Schematic diagram of 3D reconstruction of collagen fibers using IMARIS software (left) and representative images of 3D reconstructed collagen fibers processed by IMARIS (right). **(g)** Total volume of 3D reconstructed collagen fibers in tumors compared across different treatment groups. Data from 15 randomly selected spots per group (three spots per one tumor x five tumors) were analyzed (n=15/group). Data are presented as the mean ± SEM. *P<0.05, **P<0.01; one-way ANOVA. **(h, i)** Average percentages of **(h)** total CAFs and **(i)** CD141⁺ CAFs in Pan02 tumor microenvironment of each treatment group (Con, n=6; Gem, n=7; ATG+Gem, n=8; ATG+Gem+α-CD8, n=6), measured by flow cytometry. Data are presented as the mean ± SEM. *P<0.05, **P<0.01, one-way ANOVA. **(j)** Average percentages of CD4⁺ and CD8⁺ T cells in Pan02 tumor microenvironment of each treatment group, measured by flow cytometry (Con, n=6; Gem, n=7; ATG+Gem, n=8;

ATG+Gem+ α -CD8, n=6). Data are presented as the mean \pm SEM. *P<0.05, **P<0.01, ***P<0.001. one-way ANOVA.

Supplementary Figure 9 | (a) Total number, average volume, and average length of 3D reconstructed collagen fibers in tumors compared across different treatment groups. Data from 15 randomly selected spots per group (three spots per one tumor x five tumors) were analyzed (n=15/group). Data are presented as the mean \pm SEM. **P<0.01; one-way ANOVA.

3. The study validates reports that PLGF/VEGF contribute to fibrosis and that chemotherapy can enhance fibrosis in PDA. Thus, the novelty of the study is based on the development of Ate-Grab and the detailed analysis of the effect of Ate-Grab on the microenvironment of Pan02 tumors. Regarding the effect of Ate-Grab on CAFs the authors provide data that Ate-Grab mainly effects myCAF, which have been reported to have tumor-restricting functions. The effect of Ate-Grab should be discussed in this context as studies that have ablated myCAF or reduced myCAF activity have resulted in tumor progression.

We agree with the reviewer's valuable comment. As the reviewer mentioned, several studies have reported the tumor-restricting functions of myCAF, and the inhibition of cancer metastasis. To evaluate this concern, we measured tumor metastasis in the liver and lungs of tumor-bearing mice through H&E staining across all treatment groups. Tumor metastases in the liver and lung were not observed in all the treatment groups of Pan02 tumor-bearing mice.

Supplementary Figure 4 | (h) Representative H&E images of liver and lung tissues. Scale bar, 100 μ m.

The related results are now illustrated in Supplementary Figure (p. 11. highlighted in yellow; Supplementary Fig. 4h).

“Activated CAFs reportedly suppress tumor growth and metastasis⁴⁴. Hence, we performed H&E staining of lungs and livers derived from Pan02 tumor-bearing mice; however, no metastases were observed in the specimens (Supplementary Fig. 4h).”

4. The claim that CAF is the major source of PD-L1 in the TME is questionable and not consistent with other literatures that suggest cancer cells and myeloid cells are the major source. Therefore, the gating of Fig. 2I needs to be shown, and the expression of PD-L1 in the scRNA seq data also needs to be shown to prove CAF is the major source.

We apologize for omitting gating strategies to define each cell population. CAFs, myeloid cells, lymphoid cells, endothelial cells, and cancer cells are defined as follows, and PD-L1 expression was validated by a comparison with IgG control (Supplementary Fig. 2j, k). As the reviewer suggested, we have now added scRNA-seq data of untreated (no-chemo) Pan02 tumor, and re-clustered the integrated dataset. As shown below in Fig. 1 (for reviewer only), our scRNA-seq data hardly represent Cd274 (PD-L1) mRNA expression in CAF populations, while myeloid cells, especially neutrophils, highly express PD-L1 in mRNA. However, our flow cytometry data revealed that CAFs in the Pan02 tumor microenvironment highly express PD-L1 in proteins. To avoid ambiguity, we have now rephrased the sentence regarding PD-L1-expressing populations in TME.

Supplementary Figure 2 | (j) Flow cytometry gating strategy to identify indicated cellular populations. **(k)** Histogram showing PD-L1 expression on indicated populations (cancer-associated fibroblasts, cancer cells, endothelial cells, lymphoid cells, and myeloid cells) validated by flow cytometry. Histogram peaks are tinted with sky blue; FMO (Fluorescence minus one) control for PD-L1 (BUV395).

Figure 1 (for reviewer only) | Integrated scRNA-seq data of untreated (no-chemo) sample, gemcitabine mono-treated sample, and Ate-Grab+gemcitabine co-treated sample. mRNA expression of Cd274 (PD-L1) is represented with feature and dot plots.

This has been expanded upon in the Results section. (p. 7, highlighted in yellow; Supplementary Fig. 2j, k).

“Additionally, we analyzed the protein expression of PD-L1 in murine orthotopic Pan02 tumors using flow cytometry (Supplementary Fig. 2j and k), and confirmed that PD-L1-expressing CAFs are abundant in the TME, compared to other PD-L1-expressing cell types (Fig. 2o)”

5. Fig. 4f and extended Fig. 5f (myCAF) are not significant, this is counter to the overall conclusions of the authors. Does tumor size impact these results? Also how did the drug affect iCAF and apCAF populations?

We appreciate the reviewer's insightful comments. As the reviewer pointed out, the tumor size may have affected these results. Therefore, as the reviewer mentioned in comment 2, we monitored tumor growth for approximately four weeks to further clarify the results. Furthermore, based on the holistic analysis of single-cell RNA sequencing, Ate-Grab dramatically decreased CD141⁺ CAFs (Fig. 6). CD141⁺ CAF is a new CAF population that shares some iCAF properties with existing CAF classifications (Elyada et al., 2019, Cancer Discovery), while presenting unique signatures, hardly showing one-to-one correspondence (Please refer to the figure below presented only for the reviewer's consideration). Considering this and the other reviewer (reviewer 2)'s comment that myCAF is generally PDGFR α ^{low}, applying the CAF classification of myCAF, iCAF, and apCAF from Elyada et al. to evaluate the drug effect on CAF subsets originating from Pan02 tumors, seems inappropriate because the changes on our CAF subpopulations are not properly reflected by the classification. Therefore, we have now deleted the terms myCAF, apCAF, and iCAF from all flow cytometry data.

Figure 2 (for reviewer only) | Feature plots showing average expression of representative iCAF (left), myCAF (middle), and apCAF (right) signatures from Elyada et al., 2019 scored for our orthotopic Pan02 tumor-infiltrated CAF subsets.

6. In the scRNA seq data, the expression of VEGFRs should be shown.

As the reviewer commented, we have shown the expression of five VEGF-related receptor genes in a supplementary figure. Pan02 CAFs showed a significant expression of Nrp1 and Nrp2, while the Pan02 tumor microenvironment (TME) rarely expressed Flt1, Kdr, and Flt4, as shown below.

Supplementary Figure 6 | (c) Dot plot and (d) feature plot showing VEGF-related receptor gene expression in Pan02 tumor-infiltrating cells.

We have expanded on this in the Results section (p. 13, highlighted in yellow; Supplementary Fig. 6c, d).

“Of note, PIGF co-receptors, *Nrp1* and *Nrp2*, were specifically upregulated in CAFs within the Pan02 TME (Supplementary Fig. 6c, d).”

7. A no-chemo treatment control is lacking in the scRNA seq experiments. This clouds interpretation of the data, the caveats that this creates should be discussed.

As the reviewer suggested, we added scRNA-seq data of an untreated Pan02 tumor and analyzed the further heterogeneity of CAF in the Pan02 tumor microenvironment, CAF-1–CAF-6. Based on the re-clustered data with untreated samples, we have revised the manuscript as follows (p.12, highlighted in yellow; Fig. 6a-l, Supplementary Fig. 6a-g, Supplementary Fig. 8f-m).

“To investigate the changes occurring in the TME cellular composition upon treatment, we compared the single-cell transcriptomes of untreated (Control), gemcitabine-treated (Gem), and Ate-Grab+gemcitabine-treated (ATG + Gem) Pan02 tumors. Live cells from five untreated tumors, five gemcitabine-treated tumors, and four Ate-Grab + gemcitabine-treated tumors were separately pooled and subjected to single-cell RNA sequencing (scRNA-seq) analysis (Supplementary Fig. 6a). We sorted 35,697 cells based on a previously reported quality control scheme and identified nine distinct cell clusters (except for a low-quality cluster and doublet cluster) from these cells using a graph-based clustering method (Fig. 6a) ⁴⁵.”

Figure 6

Figure 6 | Ten days after Pan02 (5×10^5 cells) implantation, tumor-bearing mice were intraperitoneally treated with PBS (Con; n=5), gemcitabine (Gem; n=5) or a combination of Ate-Grab and gemcitabine (ATG + Gem; n=4). Mice were sacrificed for single-cell RNA sequencing on day 24, 2 days after the fifth drug injection. **(a)** Uniform manifold approximation and projection (UMAP) plot for orthotopic Pan02 tumor-infiltrating cells (35,697 cells). Nine cell types (except for doublets and low-quality cells) were assigned based on the expression of marker genes. **(b)** Unbiased clustering of CAF subsets revealed that combination therapy with Ate-Grab and gemcitabine (co-treatment) mostly depleted CAF-2. **(c)** Top 10 differentially expressed genes (DEGs) in six CAF subpopulations. **(d)** The number of each CAF subpopulation was quantified and compared between the three groups (Control, untreated; Gem, gemcitabine mono-treated; ATG + Gem, Ate-Grab and gemcitabine co-treated). **(e)** RNA velocity vector field for CAF differentiation indicated by streamlines. **(f)** Left: Kaplan-Meier overall survival curve of 150 pure PDAC patients grouped based on the expression of the top 30 CAF-2 DEGs (AUC value>0.7, pct.1-pct.2>0.05). Right: Kaplan-Meier overall survival curve of 73 pure PDAC patients with low tumor cellularity grouped based on the expression of the top 30 CAF-2 DEGs. HR: hazard ratio. **(g)** Feature plot of the average expression of the top 30 CAF-2 DEGs

scored in the mother plot (Pan02 tumor-infiltrated cells) to show CAF specificity of CAF-2 DEGs. (h) Dot plots of Cd141 and Cd74 expression in the CAF subpopulations. (i) Flow cytometry validated the phenotype of CAF-2 at the protein level, defined as FSC-SSC/CD45⁻/CD31⁻/EpCAM⁺/PDGFR α ⁺/MHCII⁻/CD141⁺ (left upper quadrant). (j) Bar plot showing each treatment effect on CAF-2 composition in orthotopic Pan02 tumor. Data from seven samples of control group, eight samples of gemcitabine group and nine samples of Ate-Grab+gemcitabine group were analyzed. Data are presented as the mean \pm SEM. One-way ANOVA, *P<0.05. (k) Feature plots of Cd8b1 expression in untreated (left), gemcitabine-treated (middle) and Ate-Grab + gemcitabine-treated (right) tumor-infiltrating T/NK populations. (l) Stacked bar plot of the number of lymphocyte subsets in untreated (black), gemcitabine-treated (blue), and Ate-Grab+gemcitabine-treated (yellow) Pan02 tumors.

Supplementary Figure 6 | (a) Flow cytometry gating strategy for sorting tumor-infiltrating live cells. High cell viability should be ensured for successful single-cell RNA sequencing. (b) Dot plot showing the expression of multiple marker genes for precise cell type annotation prior to downstream analysis. (c) Dot plot and (d) feature plot showing VEGF-related receptor gene expression in Pan02 tumor-infiltrating cells. (e) Unbiased clustering of CAFs revealed six different CAF subpopulations (CAF-1, CAF-2, CAF-3, CAF-4, CAF-5, and CAF-6). (f) Feature plot showing *Inhba* expression in Pan02 CAFs. (g) Enrichment pathway analysis with the top 30 DEGs of each CAF subpopulation (*Enrichr*).

Supplementary Fig. 8 | (f) Upper: Unbiased clustering of the integrated Pan02 tumor-infiltrating neutrophils (14,620 cells). Lower: Bar plot of the number of neutrophil subsets in untreated (Control), gemcitabine-treated (Gem), and Ate-Grab+gemcitabine-treated (ATG + Gem) Pan02 tumors. (g) Dot plot showing *Cd274* (PD-L1) expression in Pan02 tumor neutrophil subsets. (h) Enrichment pathway analysis with the top DEGs of each neutrophil subpopulation (*Enrichr*). (i) Unbiased clustering of the integrated Pan02 tumor-infiltrating T/NK cells (7,622 cells). (j) Dot plot of the indicated features in each T/NK subpopulation. (k) Feature plots of the indicated mRNA features in the integrated T/NK subpopulations. (l) Heatmap of the indicated mRNA expression in three CD8⁺ T subpopulations. (m) Violin plots of *Gzmb*, *Ifng*, and *Cd28* expression in gemcitabine and Ate-Grab+gemcitabine treatment groups.

8. Please compare the signatures of CAF subtypes in this study with previous published CAF scRNA seq data, to see if the CAFs found in this study are the known iCAF, myCAF apCAF or some new subtypes induced by gemcitabine treatment or just specific to the Pan02 orthotopic model.

We appreciate the reviewer's insightful advice. As the reviewer suggested, we comprehensively compared our Pan02 CAF subtypes with previously published CAF scRNA-seq data. In comparison with Elyada et al., 2019, most Pan02-infiltrated CAF subsets except for CAF-2 highly expressed MHC-II-related genes that were representative signatures of apCAF, while each subset cluster further represented unique signatures. CAF-5 shares several myCAF signatures

including *Spp1*, *Col12a1*, and *Egscr*, while CD141⁺ CAF-2 expresses some iCAF DEGs, such as *Ogn*, *Sema3c*, and *Mfap5*. However, all CAF subsets from Pan02 tumor did not show one-to-one correspondence with iCAF, myCAF, and apCAF from Elyada et al. (2019). In comparison with Dominguez et al. (2020), our CD141⁺ CAF-2 shared some markers of c0 CAF, including *Gstm1*, *Ogn*, and *Sema3c*, while several markers of myCAF-like c2 CAF were conserved in CAF-5. Notably, CAF-4 presented a high similarity with c8 CAF, showing the differential expression of many c8 CAF signatures.

Supplementary Figure 6 | (h) Unbiased clustering (UMAP embedding) of KPC tumor scRNA-seq data from Elyada et al., 2019 (fibroblast enriched dataset). (i) Left: UMAP of zoom-in clustering of CAF sorted from (h). Right: Heatmap showing marker gene expression of myCAF, iCAF, and apCAF. (j) Dot plot showing the average expression of myCAF (left), iCAF (middle), and apCAF (right) marker genes (Elyada et al., 2019) in orthotopic Pan02 CAF populations. (k) Unbiased clustering (UMAP embedding) of KPP tumor scRNA-seq data from Dominguez et al., 2020. (l) Left: UMAP of zoom-in clustering of CAF sorted from (k). Right: Heatmap showing marker gene expression of c0 CAF, c1 CAF, c2 CAF, and c8 CAF. (m) Dot plot showing the average expression of c0 CAF (left), myCAF-like c2CAF (middle), and c8 CAF (right) marker genes (Dominguez et al., 2020) in orthotopic Pan02 CAF populations.

For improved clarity, we have added these details in the revised manuscript and supplementary figures as follows. (p. 13, highlighted in yellow; Supplementary Fig. 6h-m)

“To further validate our CAF subpopulations, we compared the signatures of our Pan02 CAF subpopulations to those of previously published fibroblast-enriched datasets (Supplementary Fig. 6h–m)^{35,50}. In comparison with Elyada et al. (2019), most Pan02-infiltrated CAF subsets except for CAF-2 highly expressed MHC-II-related genes that were representative signatures of apCAF, while each clustered subset further represented its own unique signature (Supplementary Fig. 6j). CAF-5 shared several myCAF signatures including *Spp1*, *Col12a1*, and *Egscr*, while CAF-2 expressed some iCAF DEGs, such as *Ogn*, *Sema3c*, and *Mfap5* (Supplementary Fig. 6j). However, all CAF subsets from the Pan02 tumor hardly showed one-to-one correspondence with iCAF, myCAF, and apCAF³⁵. For comparison with Dominguez et al. (2020), we sorted pure CAFs from the data, and re-clustered the sorted CAF cells, so that the DEGs of each CAF subset could be calculated for pure CAF cells (not contaminated with tissue

fibroblasts or mesothelial cells) (Supplementary Fig. 6k, l). The comparison revealed that our CAF-2 shared some markers of c0 CAF, including *Gstm1*, *Ogn*, and *Sema3c*, while several markers of myCAF-like c2 CAF were conserved in CAF-5 (Supplementary Fig. 6m). Notably, CAF-4 represented a substantial similarity with c8 CAF, showing the most c8 CAF signatures (Supplementary Fig. 6m).”

9. How much of the tumor-inhibiting effects of Ate-Grab is immune related? A CD8 T cell ablation assay *in vivo* might be useful to discern the importance of the adaptive immune response.

We further evaluated whether the antitumor effect is directly mediated by the immune-related mechanism. To examine this, anti-CD8 antibody (BioXcell, BE0223; 5 mg/kg) was co-administered with Ate-Grab+gemcitabine (Fig. 7a-j; Supplementary Fig. 9a). Kindly see our answer to the comment 2.

Minor:

1. Fig. 1e, f should be validated through ELISA.

As the Reviewer suggested, we performed ELISA to measure PIGF and VEGFA protein expressions as below.

Figure 1 | (f) Relative *Pdgf* and *Vegfa* protein levels of Pan02 tumor tissue homogenates measured by ELISA. Data from seven technical repeats (n = 7) were analyzed.

We have now included the related results in Figure 1f. (p. 5, highlighted in yellow; Fig. 1f)

“To explore the significance of PIGF in pancreatic cancer, we measured its mRNA and protein levels in our orthotopic PDAC model and observed an increase after gemcitabine treatment, while the level of VEGF-A increased slightly (Fig. 1e, f).”

2. Fig. 1g is interesting. The figure legend should specify how CAFs were identified in the flow analysis. Are these CAFs *Pdgfra* or α SMA positive or was another marker used.

We apologize for the ambiguity caused by the omission of the information of how CAFs were defined. All CAFs in the paper were defined as FSC-SSC/Single cells/Live/CD45⁻/CD31⁻Cancer cell/PDGFR α ⁺ population. We have now clarified the gating strategy of CAF as below (Supplementary Fig. 1b).

Supplementary Figure 1 | (b) Gating strategy for flow cytometry analysis of cancer-associated fibroblast (CAF) and its subsets in the tumor microenvironment (TME). CAFs were gated as FSC-SSC/Live/CD45⁻/CD31⁻EPCAM⁻/PDGFR α ⁺ cells.

3. Extended Fig. 1d, the current gating may include mesenchymal cancer cells, it needs to be gated for PDGFR α or podoplanin⁺ cells.

We apologize for the ambiguity caused by the omission of the FACS gating strategy of CAFs. As mentioned above (minor comment 2), all CAFs in the paper were defined as FSC-SSC/Single cells/Live/CD45⁻/CD31⁻Cancer cell/PDGFR α ⁺ population to exclude non-CAF populations including mesenchymal cancer cells. Kindly refer to our answer for minor comment 2 above.

Reviewer #2 (Remarks to the Author): with expertise in pancreatic cancer; system oncology

Here Kim and Jung present some interesting results indicating a role for PIGF/VEGF in regulating tumour desmoplasia and immune evasion and that combined targeting of PIGF/VEGF and PD-L1 in tumour bearing animals improves vessel function and gemcitabine efficiency.

Generally, the manuscript is interesting with intriguing data of potential translational relevance. However, several conclusions are overstated and based on data which are presented with limited clarity. Thus, some of the take-away messages need additional clarifications and experimentation. Further, some conclusions that indicate causative relationships are only supported by correlative observations which should be either re-written or supported by additional data.

Specifically:

Generally, the authors have not provided sufficient information about number of replicates (biological and technical), number of animals in experimental conditions, and, in most cases, individual data points are not shown. This absolutely needs to be corrected as the validity of the results and conclusion cannot be evaluated in sufficient detail without.

We appreciate the reviewer's assessment of our work. We hope that our revisions have assuaged any remaining concerns.

Figure 1:

The authors present data demonstrating an increased level of Collagen I/tumour fibrosis following Gemcitabine. Second Harmonics generation typically capture crosslinked fibrillar collagen, and correlates with tissue rigidity. Thus, SHG is not necessarily a broad measure of desmoplasia as a general phenomenon, although I agree it correlates. However, with some textual clarification I am happy with the data included. It would be interesting to determine whether the structure of the Collagen fibres is changed fx length and thickness (this is not a critical suggestion, but mostly out of curiosity).

We appreciate the reviewer's inspection of our manuscript. For an in-depth analysis of collagen fibers, we 3D reconstructed collagen fibers utilizing IMARIS software from OXFORD Instruments. SHG images of three spots consisting of 300 sheets per tumor were taken and analyzed for the tumor tissues. The number, volume, and area of reconstructed collagen fiber were measured.

Figure 7 | (f) Schematic diagram of 3D reconstruction of collagen fibers using IMARIS software (left) and representative images of 3D reconstructed collagen fibers processed by IMARIS (right). (g) Total volume of 3D reconstructed collagen fibers in tumors compared across different treatment groups. Data from 15 randomly selected spots per group (three spots per one tumor x five tumors) were analyzed (n=15/group). Data are presented as the mean ± SEM. *P<0.05, **P<0.01; one-way ANOVA.

Supplementary Fig. 9 | (a) Total number, average volume, and average length of 3D reconstructed collagen fibers in tumors compared across different treatment groups. Data from 15 randomly selected spots per group (three spots per one tumor x five tumors) were analyzed (n=15/group). Data are presented as the mean \pm SEM. **P<0.01; one-way ANOVA.

We have now added the related data in the Methods section and Figures as shown below. (p. 25, highlighted in yellow; Fig. 7f, g, Supplementary Fig. 9a)

“SHG images were obtained from three different random spots per Pan02 tumor. Each spot consisted of 100 consecutive images in total, taken at 1 μ m intervals along the z-axis. The images were used for 3D reconstruction with Imaris 9.3 software (Bitplane Inc.), and surface functions (surface grain size = 1 μ m, automatic threshold for region growing) were applied for volume rendering. As a result, we obtained three 3D reconstructed regions, each with a 100 μ m-thick layer of collagen per tumor, and further analyzed the images with IMARIS 9.3 software by measuring the volume, area, and number. For one spot, we measured volume and length of an individual fiber, counted the number of fibers (defined as ‘Total number’), and calculated total volume of the fibers (defined as ‘Total volume’) and average volume and length of each fiber (defined as ‘Average volume’ and ‘Average length’, respectively).”

What is less clear, and should be clarified, is the number of animals included in the study and the number of areas/size of areas included in Fig 1d. Moreover, individual data points should be shown. The concentration and dosing of gemcitabine isn’t clear and should be described – this is particular important because some reports in the literature demonstrate gemcitabine sensitivity with the Pan02 cell line in vivo. Thus, is the major effect observed with Ate-Grab under gemcitabine limited conditions (which still may be interesting)

We appreciate the reviewer's helpful comment. We have clarified dosing of gemcitabine, the number of animals and the number/size of areas in the legend of Fig. 1 as illustrated below.

“On day 10 when tumor volume reached 50–100 mm³, tumor-bearing mice were intraperitoneally treated with either PBS (control, n=6) or gemcitabine (50 mg/kg every 3 days for five times, n=6).”

“Average percentages of collagen+ area out of the total area in tumors treated with PBS (control) or gemcitabine (n=25/group). For quantification, four to five SHG images of 512 μ m in width and height were obtained per tumor.”

Figure 1e/f demonstrates mRNA expression level of PIGF and VEGF, which isn't the same as protein level and should be complemented with protein measurements (fx ELISA) to solidify this critical piece of data in the manuscript.

We appreciate the reviewer's comment. As the reviewer suggested, we performed ELISA to measure the expression of PIGF and VEGFA proteins. We have now included the related results in Figure 1e, f. (p. 5, highlighted in yellow; Fig. 1f)

"To explore the significance of PIGF in pancreatic cancer, we measured its mRNA and protein levels in our orthotopic PDAC model and observed an increase after gemcitabine treatment, while the level of VEGF-A increased slightly (Fig. 1e, f)."

Figure 1 | (f) Relative *Plgf* and *Vegfa* protein levels of Pan02 tumor tissue homogenates measured by ELISA. Data from seven technical repeats (n = 7) were analyzed.

Fig 1h shows increased expression of myofibroblastic genes in primary CAFs treated with VEGF or PIGF in vitro. It is unclear how many times the experiment were repeated and whether these were individual independent experiments or technical repeats and how long were cells treated with recombinant proteins for? The authors should include additional target genes in their analysis of better defined myCAF, iCAF and apCAF markers as these are used interchangeably throughout the manuscript as CAF markers. It is therefore necessary that the authors determine whether PIGF (and VEGF) induce a specific transcriptional subtype or a more general overall expression of CAF markers across all CAF subtypes.

We appreciate the reviewer's suggestions to improve our manuscript. We have included more detailed information about the experiment, including the number of repeats and incubation time with proteins. Moreover, considering the reviewer's comment on confirming additional target genes for myCAF, iCAF, and apCAF, we selected several CAF subset markers referring to Elyada et al., and examined their expression in response to PIGF and VEGFA. We have now included the related results in the Results section and Figures as shown below. (p. 6, highlighted in yellow; Fig. 1h, Supplementary Fig. 1f)

"Recombinant mPIGF treatment induced myCAF marker expression; a ~4-fold increase in *Col1a1* and a ~5-fold increase in *Acta2* and *expression compared to controls (Fig. 1h). Treatment with recombinant mVEGF-A164 induced *Col1a1* expression, but not significantly (Fig. 1h). While PIGF upregulated the expression of several myCAF marker genes and induced a tendency to decrease in apCAF markers, no significant change in the expression of iCAF*

markers was observed (Supplementary Fig. 1f).”

Figure 1 | (h) RT-PCR detection of myCAF markers, *Col1a1*, *Acta2*, and *Fn1* transcripts, in CAFs in response to PIGF, or VEGF-A. After sequential serum deprivation for sorted CAFs, the indicated proteins (PIGF, VEGFA) and atezolizumab or Ate-Grab were added and co-incubated for 6 h. Data from three technical repeats (n = 3) were analyzed. *P<0.05, **P<0.01, ***P<0.001 vs. control; two-tailed Student's t-test.

Supplementary Figure 1 | (f) RT-PCR detection of myCAF, iCAF, and apCAF markers in response to PIGF or VEGFA. After serum starvation, PDGFR α ⁺ CAFs were treated with PIGF or VEGF for 6 h at 37°C. Data from three technical repeats were analyzed. Data are presented as the mean \pm SEM. two-tailed Student's t-test, vs. control.

Furthermore, the markers and gating strategy used for CAF isolation cannot exclude epithelial cells that have undergone EMT (low EPCAM) and only captures a minor population of CAFs as PDGFR α and α SMA expression is commonly inversely related (Elyada et al Cancer Discovery 2019, Dominguez et al Cancer Disc 2000 and Hutton et al Cancer Cell 2021). At the very least the authors should verify the absence of tumour cells in their ‘CAF’ population and also clarify in their text/legend that they are referring to subset of CAFs defined by PDGFR α expression.

We appreciate the reviewer's advice. To check the contamination of tumor cells in the PDGFR α ⁺ CAF population, we performed experiments using the Pan02-luciferase cell line by capturing anti-luciferase antibody (Abcam, EPR17789). We confirmed the expression of luciferase from the Pan02-luciferase cell line in *in vitro* settings (Figure 1a for reviewer only). By orthotopically inoculating the mouse pancreas with Pan02-luciferase cells, we were able to generate an orthotopic tumor model, which allowed a clear validation of cancer cell contamination within the PDGFR α ⁺ CAF population (Figure 1b, c for reviewer only). Cancer cell contamination to

PDGFR α ⁺ CAF was measured at approximately 8% (Figure 1d for reviewer only). We then examined the expressions of CAF subset markers including MHC-II and CD141 in the contaminated cells, which presented no difference with their expression on pure CAFs (Figure 1e for reviewer only). Therefore, we concluded that the small portion of contaminated cells did not significantly affect the results.

To address the concern regarding the use of PDGFR α for representative CAF marker, we examined correlation between the protein expression of PDGFR α and α SMA in CD45^{neg} CD31^{neg} Pan02-Luciferase^{neg} cells (Figure 1f for reviewer only). While we could not see an inverse correlation between the two molecules in our orthotopic Pan02 tumors, approximately 70% of α SMA-expressing cells also co-expressed PDGFR α in protein level (Figure 1f, g for reviewer only). However, since PDGFR α still cannot capture a portion of CAFs (approximately 30%), we decided that it was better not to define myCAF, iCAF, and apCAF in our flow cytometry results to avoid confusion. Please refer to the figures shown below presented only for the reviewer's consideration.

Figure 1 (for reviewer only) | (a) Luciferase expression of Pan02-Luciferase cells (Pan02-Luci). After Pan02-Luci was stabilized via cell culture, FACS staining was performed according to manufacturer's fixation/permeabilization procedure. **(b)** FACS gating strategy for PDGFR α ⁺ CAFs for Pan02-Luci tumor. **(c)** Assessment of cancer cell contamination in PDGFR α ⁺ CAFs. Luciferase expression, which indicates cancer cells, was examined in FSC-SSC/Single cells/Live/CD45^{neg}/CD31^{neg}/Epcam/PDGFR α ⁺ cells to evaluate cancer cell contamination in PDGFR α ⁺ CAFs. **(d)** Approximately 8% of PDGFR α ⁺ CAFs were contaminated cancer cells. **(e)** Protein expression of CAF subset markers (CD141, MHCII) over parent cells (pure CAF vs contaminated cancer cells). **(f)** Correlation between PDGFR α and α SMA expression in orthotopic Pan02 tumor. **(g)** Percentage of double positive cells (PDGFR α ⁺

α SMA⁺ cells) over either PDGFR α ⁺ or α SMA⁺ CAFs.

Finally, the data suggests that Gemcitabine driving desmoplasia through PIGF signalling in CAFs. This could be due to a change in gene expression or increased abundance of CAFs. This could be tested through IHC staining for CAFs and RT-qPCR on isolated CAFs from the tumours.

Our scRNA-seq data and flow cytometry data confirmed that gemcitabine treatment increases the CAF population in the Pan02 tumor microenvironment (TME). Moreover, the scRNA-seq data revealed that gemcitabine mono-treatment substantially increased CAF-2 and CAF-3 in Pan02 TME, and gene set enrichment analysis showed that differentially expressed genes (DEGs) of CAF-2 and CAF-3 were significantly associated with the 'TGF-beta regulation of extracellular matrix' signaling pathway, which are well known for desmoplasia induction in pancreatic cancer. We present these data here only for the reviewer's consideration. Please note that we have now added scRNA-seq data of untreated Pan02 tumor samples together with those of the gemcitabine treated group and Ate-Grab+gemcitabine treated group, considering the suggestions from other reviewers (Reviewer 1 and Reviewer 4), and six CAF subsets were identified (Fig. 6 and Supplementary Fig. 6).

Figure 2 (for reviewer only) | (a) Percentages of CAF and CAF subsets in TME, compared between untreated Pan02 sample and Gemcitabine-treated Pan02 sample. **(b)** Gene set enrichment analysis with differential features of CAF-2 and CAF-3. 'TGF-beta regulation of extracellular matrix' signaling pathway was enriched in gemcitabine-induced CAF-2 and CAF-3.

Figure 2:

Its unclear whether the patients analysed had been subjected to chemotherapy and whether patients in Sub4 and Sub6 have been subjected to similar treatment regime.

All patients in this study had not received chemotherapy prior to surgery.

Also, the authors compare patient subsets based on best and worst prognosis, however for the purpose of interrogating tumour desmoplasia it would be worthwhile to also compare the subsets with most and least amount of desmoplasia (fx as determined by tumour cellularity).

We agree with the reviewer's insightful comments. We confirmed that there is a strong positive correlation (P -value = 0.002, r value = 0.6586) between PIGF and collagen abundance (Fig. 2e). Additionally, there is a strong positive correlation (P -value=0.006, r value=0.5955) between VEGF-A and collagen abundance (Fig. 2f). Furthermore, the co-stained area with VEGF-related receptor (NRP1 or VEGFR1 or VEGFR2) and α -SMA had a positive correlation with collagen

abundance as well (P -value=0.006, r value=0.594; P -value = 0.015, r value=0.5353; P -value=0.053, r value=0.4391) (Fig. 2k, Supplementary Fig. 2f, g). We have now elucidated all of these results in the Results section (p. 7, highlighted in yellow; Fig. 2e, f, k; Supplementary Fig. 2f, g).

“Furthermore, tumor fibrosis demonstrated a significant correlation with PIGF and VEGF expression ($r = 0.6586$, $p = 0.002$; $r = 0.5955$, $p = 0.006$ respectively) (Fig. 2e, f).

Double-staining IHC analysis revealed that CAFs expressed PIGF/VEGF receptors (Fig. 2g), and NRP1⁺ CAF, VR1⁺ CAF, and VR2⁺ CAF were strongly negatively correlated with patient prognosis ($r = -0.7199$, $p < 0.001$; $r = -0.6898$, $p < 0.001$; $r = -0.7131$, $p < 0.001$, respectively) (Fig. 2h-j; Supplementary Fig. 2d, e), while NRP1⁺ CAF and VR1⁺ CAF positively correlated with tumor fibrosis measured by Masson’s trichrome⁺ area ($r = 0.594$, $p = 0.006$; $r = 0.5353$, $p = 0.015$ respectively) (Fig. 2k; Supplementary Fig. 2f, g).”

Figure 2 | Correlation analyses between tumor fibrosis (measured by Masson’s trichrome⁺ area) and (e) PIGF⁺ area% or (f) VEGFA⁺ area%. Data from 20 PDAC patient samples were analyzed (n=20). Correlation analyses between (k) tumor fibrosis and NRP1⁺α-SMA⁺ area%. Data from 20 PDAC patient samples were analyzed (n=20).

Supplementary Figure 2 | Correlation analyses between tumor fibrosis and (f) VR1⁺α-SMA⁺ area% or (g) VR2⁺α-SMA⁺ area%. Data from 20 PDAC patient tumor samples were analyzed (n=20).

The representation of the flow cytometry analysis of PD-L1 levels across tumour and stromal cells is dependent on the cell viability, where different cell types are more or less viable. The authors should show the biaxial plots in the supplemental information. It is also not clear which markers were used for cell identification, whether the analysis was done in one experiment or in different experiments and the number of repeats?

We apologize for omitting the gating strategies used to define each cell population. CAFs, myeloid cells, lymphoid cells, endothelial cells, and cancer cells are defined as follows, and PD-L1 expression was validated by a comparison with IgG control. (Supplementary Fig. 2j, k) Our flow cytometry data confirmed that CAFs in the Pan02 tumor microenvironment highly expressed PD-L1. In addition, we obtained the PD-L1 expression data of FACS from one experiment with five orthotopic pancreatic tumors (n=5). (Supplementary Fig. 2j, k)

Supplementary Figure 2 | (j) Flow cytometry gating strategy to identify indicated cellular populations. (k) Histogram showing PD-L1 expression on indicated populations (cancer-associated fibroblasts, cancer cells, endothelial cells, lymphoid cells, and myeloid cells) validated by flow cytometry. Histogram peaks are tinted with sky blue; FMO (Fluorescence minus one) control for PD-L1 (BUV395).

Figure 3:

For most experiments its unclear how many biological and technical repeats were included.

We apologize for omitting the information about statistics. Three replicates per group were included in all experiments in Fig. 3. We have now included the information in the legend of Fig. 3 as below.

“Data from three technical repeats were analyzed (n=3/group).”

Fig3j, as for Fig 1, additional markers of myCAF, iCAF and apCAF should be included and purity of CAFs should be validated.

We appreciate the Reviewer's insightful comment. As for revised Fig. 1, we examined additional markers of myCAF and apCAF in response to Atezolizumab and Ate-Grab. To note, since there was no any change of iCAF marker expression in response to PIGF (Supplementary Fig. 1), we did not examine iCAF markers for this. We have now included the related results in the Results section and Figures. (p. 9, highlighted in yellow; Fig. 3j)

“Based on previously published CAF subset markers by Elyada et al.³⁵, we examined *in vitro* marker gene expression of myCAF and apCAF in our Pan02 tumor-infiltrated CAFs in response to atezolizumab, VEGF-Grab, or Ate-Grab, and observed that Ate-Grab effectively inhibits PIGF-induced myCAF marker upregulation (*Col1a1*, *Acta2*) compared to atezolizumab, while no significant change was observed with apCAF marker expression (*Cd74*, *H2-Ab1*) between the two groups (Fig. 3j)”

Figure 3 | (j) Relative expression levels of myCAF (*Col1a1*, *Acta2*) and apCAF (*Cd74*, *H2-Ab1*) markers in CAFs from Pan02 tumors. CAFs were isolated from Pan02 tumors via fluorescence-activated cell sorting (FACS) and treated with PIGF or the indicated drug combinations with PIGF for 6 h after the sequential serum deprivation. Data from three technical repeats were analyzed (n=3/group). Data are presented as the mean ± SEM. *P<0.05, ***P<0.001 versus control (e, f, h, j). one-way ANOVA.

Regarding the purity of CAFs, cancer cell contamination of PDGFR α ⁺ CAF was measured as approximately 8%, and there was no difference in CAF subset marker expressions compared with that in pure CAFs. Therefore, we concluded that the small portion of contaminated cells did not significantly affect the results. Please refer to the answer to the comment for Fig. 1.

Its unclear why a comparison was made between atezolizumab and Ate-Grab rather than VEGF-Grab and Ate-Grab?

VEGF-Grab fused to antibody scFv was verified as a bispecific platform (Lee et al., 2018, *Biomaterials*). Since the binding affinity of Ate-Grab to target antigens PD-L1 and VEGFA/PIGF is equivalent to that of parental drugs (VEGF-Grab and atezolizumab), the concerns regarding the target specificity of drugs were considered as minimal. Kindly refer to Fig. 3c, Supplementary Fig. 3b. By comparing the effect between atezolizumab and Ate-Grab, we indicated the anti-VEGF/PIGF effect of Ate-Grab that are absent in Atezolizumab. Indeed, Ate-Grab can effectively inhibit PIGF-induced CAF activation; however, the blocked PD1/PD-L1 axis does not affect the PIGF-induced CAF activation.

Figure 4:

Its unclear at which point tumours were treated from and number of animals pr treatment group

We apologize for omitting this important information on treatments. Each drug was treated five times every 3 days from Day 10 after tumor implantation. We have illustrated the treatment schedule and number of animals per treatment group in Supplementary Fig. 4a and its legend.

Supplementary Figure 4 | (a) Experimental treatment scheme. Murine orthotopic pancreatic models were generated by implanting Pan02 cells (5×10^5 cells/mouse) into the pancreas of 7–10-week old C57BL/6 mice. When the tumor volume reached 50–100 mm³, tumor-bearing mice were intraperitoneally treated with either PBS (control; n=6) or atezolizumab (10 mg/kg, every 3 days; n=5), VEGF-Grab (10 mg/kg; n=4) and Ate-Grab (10 mg/kg; n=5).

The authors claim that Ate-Grab reduce myCAFs, however the gating used in this experiment does not support this conclusion. myCAFs are generally low for PDGFR α (as described above), thus pre-gating for PDGFR α CAFs excludes myCAFs and thus the authors have already selected for a CAF subset prior to their analysis. Furthermore, EPCAM will only exclude epithelial and not mesenchymal tumour cells from their analysis, which can cause inference in the analysis. The authors should re-do the experiment where total CAFs are then analysed by markers for myCAF/iCAF and apCAF as well as RNA expression (qPCR perfectly fine) should be used to validate this observation.

We appreciate the reviewer's insightful comment. We used PDGFR α as a total CAF marker because, unlike KPC tumors, CAFs in Pan02 tumors hardly express PDPN. We agree with the reviewer's comment that myCAFs are low for PDGFR α , and therefore our PDGFR α ^{pos} CAFs may exclude myCAF, as by Elyada et al. Therefore, we performed comparison analysis between CAF subsets from Elyada et al. and our Pan02 CAFs, and the result showed low similarity between the two tumor models. Therefore, applying the CAF classification of myCAF, iCAF, and apCAF from Elyada et al. to evaluate the drug effect on CAF subsets originating from Pan02 tumors seems inappropriate since the changes in our CAF subpopulations were not properly reflected by this classification of Elyada et al. Kindly refer to the figure below and our answers to the reviewer's comment on Figure 6.

Figure 3 (for reviewer only) | Feature Plots showing average expression of representative iCAF (left), myCAF (middle), apCAF (right) signatures from Elyada et al., 2019 scored in our orthotopic Pan02 tumor-infiltrated CAF subsets.

The change in vessel formation and function is based solely on PDGFRb overlay with CD31, which doesn't add functional insight. The authors should demonstrate increased perfusion such as by a high molecular dextran.

We appreciate the reviewer's insightful comment. Considering the reviewer's comment, we conducted NG2 overlay with CD31 and perfusion assay using 2000 kDa high molecular dextran to evaluate vessel normalization, and successfully confirmed that dextran⁺ area significantly increased in the Ate-Grab treatment group. (p. 10, highlighted in yellow; Fig. 4g-j)

“As tumor fibrosis is closely related to vessel normalization^{10,11}, we evaluated Pan02 tumor vasculature through immunofluorescence analysis of CD31 as an endothelial cell marker and NG2 or PDGFR β as a pericyte marker. The Ate-Grab-treated group, which exhibited decreased tumor fibrosis (Fig. 4d, e), had greater NG2⁺ or PDGFR β ⁺ pericyte coverage (Fig. 4g, h; Supplementary Fig. 4b, c). Furthermore, a perfusion assay using 2000 kDa dextran revealed that only Ate-Grab significantly recovered vessel perfusion (Fig. 4i, j). These results suggest that Ate-Grab treatment inhibits CAF activation by targeting VEGF/PIGF and relieves the vessel compression caused by tumor desmoplasia, promoting vessel normalization.”

Figure 4 | (g) Representative immunofluorescence (IF) images staining for NG2 (green) and CD31 (red) in Pan02 tumors treated with different agents. Yellow indicates the co-expression of NG2 and CD31. Scale bar, 100 μ m. (h) Quantifications of yellow regions (NG2⁺CD31⁺) in IF data were analyzed by ImageJ. Data from 28 randomly selected fields of view per group were analyzed (n=28). **P<0.01, One-way ANOVA. (i) Representative immunofluorescence (IF) images staining for CD31 (red) and Dextran (green) in Pan02 tumors treated with different agents. Yellow indicates the co-expression of CD31 and Dextran. Scale bar, 100 μ m. (j) Quantifications of Dextran⁺/CD31⁺ in IF data were analyzed by ImageJ. Data from 28 randomly selected fields of view per group were analyzed (n=28). *P<0.05, **P<0.01, One-way ANOVA.

Figure 5:

As noted above, FACS gating strategy should be clarified, n numbers displayed and individual data points displayed. If similar strategy for CAF analysis were used as in previous figures the authors cannot distinguish between myCAF, iCAF and apCAF. Moreover, the authors cannot exclude mesenchymal tumour cells are included in this analysis. This should be backed up by improved FACS, IHC and gene expression analysis.

We appreciate the reviewer's advice. As the reviewer suggested, we included the FACS gating strategy (Supplementary Fig. 2) and indicated the total number and individual data points in all figure legends. Regarding CAF purity, please refer to the answer to the comment for Fig. 1. For the identification of myCAF, iCAF, and apCAF, we considered that applying the CAF classification of myCAF, iCAF, and apCAF from Elyada et al. to evaluate the drug effect on CAF subsets originating from Pan02 tumors was inappropriate, and we thus decided that it was better not to measure myCAF, iCAF, and apCAF% in our flow cytometry results to avoid confusion. For more details, please kindly refer to our answers for the comment for Fig. 4 above.

The decrease in total CAF numbers is notable (Figure 5d). This is a possible point of concern as increased vascularisation and CAF depletion in other studies (Rhim and Stager Cancer Cell 2014, Ozdemir and Kalluri Cancer Cell 2014) have shown this is correlated with increased metastasis and poor overall survival. The authors should collect and analyse livers for metastatic seeding.

We appreciate the Reviewer's valuable suggestion. We examined tumor metastasis in liver and lung tissues of tumor bearing mice through H&E staining across treatment groups. Tumor metastases in the liver and lung were not observed in all the treatment groups of Pan02 tumor-bearing mice. The related results are now illustrated in Supplementary Figure (p. 11. highlighted in yellow; Supplementary Fig. 4h).

Supplementary Figure 4 | (h) Representative H&E images of liver and lung tissues. Scale bar, 100 μ m.

“Activated CAFs reportedly suppress tumor growth and metastasis⁴⁴. Hence, we performed H&E staining of lungs and livers derived from Pan02 tumor-bearing mice; however, no metastases were observed in the specimens (Supplementary Fig. 4h).”

The authors demonstrate decreased tumour volume and hypothesise this is due to increased anti-tumour immunity. However, there is no data to support this is due to decrease in tumour cell abundance and increased killing of tumour cells by immune cells. Additional data to test this hypothesis are needed.

We appreciate the reviewer's criticism. To support the claim that the decreased tumor volume was mediated by anti-tumor immunity, we validated that the *in vivo* treatment of anti-CD8 antibodies significantly inhibited both anti-tumor effects that were accelerated by Ate-Grab+gemcitabine treatment. In the Ate-Grab+gemcitabine+anti-CD8 antibody-treated group, both the volumes and weights of the tumors increased to similar levels as those in the control group, while tumor fibrosis and CAF population % were partially increased, however not significantly (Fig. 7f-g; Supplementary Fig. 9a).

Figure 7

Figure 7 | (a) Experimental treatment scheme. Murine orthotopic pancreatic models were generated by implanting Pan02-Luciferase (5×10^5 cells/mouse) into the pancreas of 7–10-week-old C57BL/6 mice. When the tumor volume reached 50–100 mm³, tumor-bearing mice were intraperitoneally treated with either PBS (control), gemcitabine (50 mg/kg, every 3.5 days), gemcitabine with Ate-Grab (10 mg/kg, every 3.5 days), or gemcitabine with Ate-Grab and anti-CD8 antibody (5 mg/kg, once a week). **(b)** Representative ultrasonographic images for measuring orthotopic Pan02-Luciferase tumor growth *in vivo*. **(c)** Tumor volumes measured by ultrasonography and compared across indicated treatment groups (Con, n=6; Gem, n=7; ATG+Gem, n=8; ATG+Gem+ α -CD8, n=6). Con, control; Gem, gemcitabine; ATG+Gem, Ate-Grab and gemcitabine co-treatment; ATG+Gem+ α -CD8, Ate-Grab, gemcitabine, and α -CD8a antibody co-treatment. $\star\star\star P < 0.001$, two-way ANOVA. **(d)** Representative images of KPC tumors from indicated treatment groups. Ruler scale, 1 mm. **(e)** Tumor weights measured and compared across indicated treatment groups (Con, n=6; Gem, n=7; ATG+Gem, n=8; ATG+Gem+ α -CD8, n=6). Data are presented as the mean \pm SEM. $\star P < 0.05$, $\star\star P < 0.01$, one-way ANOVA. **(f)** Schematic diagram of 3D reconstruction of collagen fibers using IMARIS software (left) and representative images of 3D reconstructed collagen fibers processed by IMARIS (right). **(g)** Total volume of 3D reconstructed collagen fibers in tumors compared across different treatment groups. Data from 15 randomly selected spots per group (three spots per one tumor x five tumors) were analyzed (n=15/group). Data are presented as the mean \pm SEM. $\star P < 0.05$, $\star\star P < 0.01$; one-way ANOVA. **(h, i)** Average percentages of (h) total CAFs and (i) CD141+ CAFs in Pan02 tumor microenvironment of each treatment group (Con, n=6; Gem, n=7; ATG+Gem, n=8; ATG+Gem+ α -CD8, n=6), measured by flow cytometry. Data are presented as the mean \pm SEM. $\star P < 0.05$, $\star\star P < 0.01$, one-way ANOVA. **(j)** Average percentages of CD4+ and CD8+ T cells in Pan02 tumor microenvironment of each treatment group, measured by flow cytometry (Con, n=6; Gem, n=7; ATG+Gem, n=8; ATG+Gem+ α -CD8, n=6). Data are presented as the mean \pm SEM. $\star P < 0.05$, $\star\star P < 0.01$, $\star\star\star P < 0.001$, one-way ANOVA.

ANOVA.

Supplementary Figure 9 | (a) Total number, average volume, and average length of 3D reconstructed collagen fibers in tumors compared across different treatment groups. Data from 15 randomly selected spots per group (three spots per one tumor x five tumors) were analyzed (n=15/group). Data are presented as the mean ± SEM. **P<0.01; one-way ANOVA.

We have now elucidated all of these results in the Results section. (p. 16, highlighted in yellow; Fig. 7a-j, Supplementary Fig. 9a)

“Based on the increased intratumoral CD8⁺ T cells and their cytotoxic features observed with Ate-Grab+gemcitabine combination therapy, we further determined whether the anti-tumor effect of Ate-Grab is directly mediated by immune-related mechanisms. An anti-CD8 neutralizing antibody was co-administered with Ate-Grab+gemcitabine (Fig. 7a), causing a substantial inhibition of the anti-tumor effects that were improved in response to the Ate-Grab+gemcitabine regimen (Fig. 7b-e). Specifically, in the Ate-Grab+gemcitabine+anti-CD8 antibody-treated group, tumor volume and weight increased to the levels of the control (Fig. 7b-e). Also, 3D reconstructed images of tumor collagen structure demonstrated that co-treatment of anti-CD8 antibody with Ate-Grab+gemcitabine did not significantly reverse the anti-fibrotic effect exerted by Ate-Grab, while individual collagen fiber volume and length did not change either (Fig. 7f, g; Supplementary Fig. 9a). Flow cytometry analysis also showed that co-treatment of Ate-Grab+gemcitabine with anti-CD8 antibody does not significantly affect total or CD141⁺ CAF population, while dramatically depleting CD8⁺ T cells and also reducing the number of CD4⁺ T cells *in vivo* (Fig. 7h-j). These results demonstrate that the tumor-inhibiting effect of Ate-Grab is mediated by CD8⁺ T cells in the TME. ”

Figure 6:

The scRNAseq leaves a number of critical questions open:

It's curious why so few endothelial cells are picked up when the authors in previous figures demonstrate an 'normalisation' of the vasculature which would be expected to improve therapeutic delivery. At the very least I would anticipate some CD31 endothelial cells would have been identified.

We appreciate the helpful comments from the reviewer for all scRNA-seq data. Regarding endothelial cells, CD31 (Pecam1)-expressing endothelial cells were not identified in our Pan02 scRNA-seq data, for unknown reasons. Meanwhile, we recently discovered another scRNA-seq

study on murine Pan02 orthotopic tumors (Zhou et al., 2022, *Translational Oncology*). In this study, endothelial cells were also not detected with scRNA-seq. We show the related figures below only for the reviewer's consideration.

Figure 5 (for reviewer only) | a, Feature plot showing Pecam1 expression in our Pan02 tumor microenvironment (Kim et al.). **b**, Dimplot showing all cell types in Pan02 tumor scRNA-seq data illustrated by Zhou et al. (2022), *Translational Oncology*.

From the methods and text, it appears that the authors use only one gene to identify individual cell types and subsets? This can be misleading and additional genes should be used to verify cell identity to avoid mis-annotation.

As the reviewer suggested, we have confirmed the expressions of additional marker genes to clearly verify cell identity, and the data are shown in a supplementary figure as follows. (p. 12, highlighted in yellow; Supplementary Fig. 6b)

“The nine clusters were named based on the predominant expression of cellular markers including those related to CAFs, T/NK cells, B cells, monocytes/macrophages, neutrophils, DCs, plasmacytoid DCs, mast cells and cancer cells (Supplementary Fig. 6b).”

Supplementary Figure 6 | (b) Dot plot showing the expression of multiple marker genes for precise cell type annotation prior to downstream analysis.

As CAFs, endothelial cells (vascular and lymphatic) as well as mesenchymal tumour cells can easily

appear similar by scRNAseq, the authors should assign these population with much greater care. Its also curious that only few tumour cells are identified in the scRNA analysis – which is a possible concern as ‘CAFs’ could originate from epithelial cells and thus be a mis-classification. Alternatively, most tumour cells are dead at the time of analysis and therefore raise a question to the relevance of looking at this particular timepoint?

We thank the reviewer for her/his comment. As answered above, multiple cell type marker genes were examined in our scRNA-seq data to validate the cell types without misclassification. Meanwhile, when we processed scRNA-seq data for quality control, we filtered cells with the threshold of nFeature_RNA (min = 500, max = 7500) value and percentage of mitochondrial genes (<15) to exclude dead cells. To examine whether dead cells were filtered out for downstream analysis, we adjusted threshold values to include low quality and dead cells. However, there was no significant increase in cancer cells in our scRNA-seq data. As the reviewer pointed out, there is a possibility of sample bias at the time point of the experiment and sample processing to make single cell suspension or to sort live cells by FACS. We decided the particular time point of the experiment considering the treatment schedule to examine the treatment effect in the tumor microenvironment. Despite the possibility, we considered that it would not significantly affect the biological phenomenon observed in CAF and immune populations.

It is not clear how CAF1-3 were assigned and whether these correspond to previously identified CAF subset (see Elyada et al Cancer Discovery 2019, Dominguez et al Cancer Discovery 2000). Also, the authors should verify that these CAF populations can be identified in other models of PDA such as by re-analysing the afore mentioned papers. This is important to confirm the general observation of the CAF subset identified.

We appreciate the reviewer’s insightful suggestion. As the reviewer suggested, we comprehensively compared our Pan02 CAF subtypes with previously published CAF scRNA-seq data. In comparison with Elyada et al. (2019), most Pan02-infiltrated CAF subsets, except for CAF-2, highly express MHC-II-related genes that are representative signatures of apCAF, while each subset clustered further represented its own unique signature. CAF-5 shared several myCAF signatures including *Spp1*, *Col12a1*, and *Ecscr*, while CD141⁺ CAF-2 expressed iCAF DEGs such as *Ogn*, *Sema3c*, and *Mfap5*. However, all CAF subsets from Pan02 tumor did not show one-to-one correspondence with iCAF, myCAF, and apCAF from Elyada et al. (2019). In comparison with Dominguez et al. (2020), our CD141⁺ CAF-2 shared some markers of c0 CAF, including *Gstm1*, *Ogn*, and *Sema3c*, while several markers of myCAF-like c2 CAF were conserved in CAF-5. Notably, CAF-4 represents a high similarity with c8 CAF, showing the differential expression of most c8 CAF signatures.

Supplementary Figure 6 | (h) Unbiased clustering (UMAP embedding) of KPC tumor scRNA-seq data from Elyada et al., 2019 (fibroblast enriched dataset). (i) Left: UMAP of zoom-in clustering of CAF sorted from (h). Right: Heatmap showing marker gene expression of myCAF, iCAF, and apCAF. (j) Dot plot showing the average expression of myCAF (left), iCAF (middle), and apCAF (right) marker genes (Elyada et al., 2019) in orthotopic Pan02 CAF populations. (k) Unbiased clustering (UMAP embedding) of KPP tumor scRNA-seq data from Dominguez et al., 2020. (l) Left: UMAP of zoom-in clustering of CAF sorted from (k). Right: Heatmap showing marker gene expression of c0 CAF, c1 CAF, c2 CAF, and c8 CAF. (m) Dot plot showing the average expression of c0 CAF (left), myCAF-like c2CAF (middle), and c8 CAF (right) marker genes (Dominguez et al., 2020) in orthotopic Pan02 CAF populations.

For improved clarity, we have added these details in the revised manuscript and supplementary figures as follows. (p. 13, highlighted in yellow; Supplementary Fig. 6h-m)

“To further validate our CAF subpopulations, we compared the signatures of our Pan02 CAF subpopulations to those of previously published fibroblast-enriched datasets (Supplementary Fig. 6h–m)^{35,50}. In comparison with Elyada et al. (2019), most Pan02-infiltrated CAF subsets except for CAF-2 highly expressed MHC-II-related genes that were representative signatures of apCAF, while each clustered subset further represented its own unique signature (Supplementary Fig. 6j). CAF-5 shared several myCAF signatures including *Spp1*, *Col12a1*, and *Ectscr*, while CAF-2 expressed some iCAF DEGs, such as *Ogn*, *Sema3c*, and *Mfap5* (Supplementary Fig. 6j). However, all CAF subsets from the Pan02 tumor hardly showed one-to-one correspondence with iCAF, myCAF, and apCAF³⁵. For comparison with Dominguez et al. (2020), we sorted pure CAFs from the data, and re-clustered the sorted CAF cells, so that the DEGs of each CAF subset could be calculated for pure CAF cells (not contaminated with tissue fibroblasts or mesothelial cells) (Supplementary Fig. 6k, l). The comparison revealed that our CAF-2 shared some markers of c0 CAF, including *Gstm1*, *Ogn*, and *Sema3c*, while several markers of myCAF-like c2 CAF were conserved in CAF-5 (Supplementary Fig. 6m). Notably, CAF-4 represented a substantial similarity with c8 CAF, showing the most c8 CAF signatures (Supplementary Fig. 6m).”

The survival analysis with CAF2 signature is potentially confounded. If individual genes from the CAF2 signature are also expressed in other cell types how can the authors conclude that it is the CAF2 subset that is correlated with patient survival and not changes in other cellular sources? This analysis should

also take tumour cellularity into account as the authors may otherwise compare tumour samples with different tumour cellularity. The authors should also annotate the subtype of the tumours to test whether then CAF signature is related to different tumour transcriptional subtypes (eg Collison et al Nat Med 2011, Bailey et al Nature 2016, Moffit et al Nat Genet 2015). For comparison the authors should include a comparison with CAF1 and 3 transcriptional signatures on patient survival.

We appreciate the insightful comment from the reviewer. Please note that we have excluded the following samples from the TCGA-PAAD dataset (Neuroendocrine/Acinar cell, carcinoma/Intraductal papillary, mucinous neoplasm, Undifferentiated/Systemic treatment given to the prior/other malignancy, Samples without tumor cellularity information) considering reviewer 4's suggestion and performed downstream analysis with the pure PDAC patients. We then confirmed that the top 30 DEGs of CAF-2 (AUC value > 0.7, pct.1-pct.2 > 0.05) significantly affected the overall survival of 150 pure PDAC patients, while genes specific to CAF were validated using a mother plot including the whole cell types in the Pan02 tumor microenvironment. Moreover, considering the Reviewer's suggestion, we grouped patients by tumor cellularity and verified that the top DEGs of CAF-2 showed a higher hazard ratio (HR) in PDAC with low tumor cellularity (< median value 18.333), while the top DEGs of other CAF subsets were not significant for overall survival. The related results are now depicted in the Results section and Figures. Again, please note that we have now added scRNA-seq data of untreated Pan02 tumor samples in response to the suggestions from reviewers 1 and 4, and the six CAF subsets are analyzed (no change on CD141⁺ CAF-2 subset). (p. 14, highlighted in yellow; Fig. 6f-g; Supplementary Fig. 7a, b)

“To determine the clinical value of CAF-2, we performed Kaplan–Meier overall survival analysis using publicly available PDAC patient data ⁵¹ (Fig. 6f; Supplementary Fig. 7a, b). The CAF-specific DEGs of CAF-2, showing higher expression in pancreatic tumors compared to normal tissues (Supplementary Fig. 8a), were negatively correlated with patient prognosis (HR=1.65), and showed a higher hazard ratio (HR=1.97) in PDAC patients with lower tumor cellularity (Fig. 6f, g), highlighting CAF-2 as a promising therapeutic target, especially for desmoplastic cancer.”

Figure 6 | (f) Left: Kaplan-Meier overall survival curve of 150 pure PDAC patients grouped based on the expression of the top 30 CAF-2 DEGs (AUC value>0.7, pct.1-pct.2>0.05). Right: Kaplan-Meier overall survival curve of 73 pure PDAC patients with low tumor cellularity grouped based on the expression of the top 30 CAF-2 DEGs. HR: hazard ratio. (g) Feature plot of the average expression of the top 30 CAF-2 DEGs scored in the mother plot (Pan02 tumor-infiltrated cells) to show CAF specificity of CAF-2 DEGs.

Supplementary Figure 7 | (a) Kaplan-Meier overall survival curves of 150 PAAD patients grouped based on the expression of the top 30 DEGs of CAF-1, CAF-3, CAF-4, CAF-5, and CAF-6 were analyzed respectively via the KM-plotter. HR: hazard ratio.

Supplementary Figure 7 | (b) Kaplan-Meier overall survival curves of 77 PAAD patients with high tumor cellularity (\geq median value of 18.333) and 73 PAAD patients with low tumor cellularity ($<$ median value of 18.333) grouped based on the expression of the top 30 DEGs of CAF-1~CAF-6 were analyzed respectively via the KM-plotter. HR:

hazard ratio.

The FACS gating in 6f is not clear and should be shown in supplemental data

We apologize for omitting the FACS gating strategy. We divided the CAF subsets using CD141 and MHCII as shown in Figure 6i, with PDGFR α ⁺ CAF gated as in Supplementary Fig. 2j.

Supplementary Figure 2 | (j) Flow cytometry gating strategy to identify indicated cellular populations.

General edits:

The authors state that “the desmoplastic stroma consists mainly of CAFs...” (line 47) Although CAFs certainly are very abundant, cell quantification by tissue disaggregation is depending on equal liberation of all cell types (which I don’t think has been demonstrated to be the case). This also doesn’t seem to be the observation in the authors scRNAseq and in other reports (Elyada et al and Dominguez et al). In situ studies, such as IHC, IF etc, are more limited in the number of cells that can be enumerated and cellular composition is likely to change throughout a tumour. Also, I am not aware of studies that have enumerated stromal cells across several tumours using consecutive slides. As such, unless the authors can provide a more convincing reference I think they should change their statement, perhaps just referencing that CAFs are an abundant cell type in the microenvironment...

We appreciate the reviewer's insightful suggestion. As the reviewer mentioned, as the paper did not elucidate stromal cell composition in detail, we have now rephrased our descriptions as below (p. 3, highlighted in yellow).

"Cancer-associated fibroblasts (CAFs) are one of the abundant cell types in the desmoplastic stroma, which is the major source of extracellular matrix (ECM) within the tumor microenvironment (TME) ¹²."

L28/29: The authors write “Ate-Grab... target PD-L1 expressing CAFs”. There is no evidence in the manuscript that this is a case. While it’s a possible explanation this should be part of the discussion and should be presented as one of several possible explanations.

We appreciate the reviewer's insightful suggestion. We have now presented possible explanations regarding this in the Discussion section (p. 21, highlighted in yellow).

" This study has some limitations. For example, the concrete mechanisms of Ate-Grab *in vivo* were not clarified. Further, although we successfully confirmed that Ate-Grab was efficiently delivered to tumor tissues, the specific cell types directly targeted by Ate-Grab were unclear. Regarding this issue of the specific cell types, we have the following explanations: first, since Ate-Grab has a binding affinity for PD-L1, Ate-Grab can activate T cells and boost their infiltration into the TME. The activated T cells would secrete inflammatory cytokines and modulate the TME as observed in our study with murine PDAC by affecting CAFs and myeloid cells, major components of the TME. Second, bi-specific molecules reportedly exhibit inherent functions beyond a simple dual-targeting effect. As the various cell types in the TME express PD-L1, there is a possibility that Ate-Grab blocks angiogenic molecules around PD-L1-expressing cells. To clearly address this question, further studies, including inhibition assays of specific cell types with genetic models, are required."

L32/33: The authors write: “the CD141+ CAF population, as responsible for the therapeutic effect of Ate-Grab”. Another overstatement in the manuscript which there is no evidence for. The authors can show that the level of CD141+ CAFs are decreased in Ate-Grab + Gem treated tumours in comparison with Gem treated tumours, but there is no evidence to support the presented conclusion. This should be removed or supported by additional evidence.

We appreciate the reviewer's valuable suggestions. Because of the lack of direct evidence for the effect of CD141+ CAFs on the therapeutic effect of Ate-Grab as mentioned by the reviewer, we have revised the sentence in the Abstract as below. (p. 2, highlighted in yellow).

"Single-cell RNA sequencing identified that the CD141⁺ CAF population was reduced upon Ate-Grab and gemcitabine combination treatment."

Reviewer #3 (Remarks to the Author): with expertise in pancreatic cancer; cancer associated fibroblasts

In this manuscript by Kim, Jeong and colleagues, the authors investigate associations among chemotherapy treatment, growth factor signaling, collagen deposition, and response to immune checkpoint blockade in pancreatic cancer. In light of the very poor prognosis for pancreatic cancer patients and the broad ineffectiveness of chemotherapy and ICB within this patient population, mechanistic studies highlighting signaling nodes for therapeutic intervention to foster ICB efficacy in this disease setting is highly significant and represents an important and timely goal for the field. Using pancreatic cancer mouse models and clinical specimens, the authors here test the overarching hypothesis that gemcitabine treatment leads to induction of growth factors including placental growth factor and VEGF-A, which subsequently act on CAFs to enhance collagen deposition. Evidence in support of these connections in the manuscript are rather tenuous; for example, the impact of gemcitabine on collagen I deposition in vivo appears very modest and is of questionable relevance to the therapeutic preclinical studies later in the manuscript. The authors go on to develop and test an antibody called Ate-Grab, building on a prior and somewhat similar molecule but here with the ability to inhibit PD-1/PD-L1 signaling as well as VEGF-A and PlGF signaling. They show efficacy of this antibody combined with gemcitabine in reducing tumor growth in vivo. This is an interesting and novel therapeutic approach, with additional novelty stemming from the focus on PlGF, which has been subject to rather little study in pancreatic cancer. However, effects of VEGF/PlGF inhibition by Ate-Grab in vivo are modest and only shown in a single mouse model which bears questionable relevance to human pancreatic cancer. Addressing the comments below would help to strengthen the authors central claims.

We appreciate the reviewer's positive assessment of our findings. We hope that our revisions have assuaged any remaining concerns.

Specific comments:

1. In vivo studies here are limited to transplant experiments using the Pan02 cell line, which is unusual in that it harbors wild-type KRAS while pancreatic tumors are almost always KRAS-mutant. In light of this and the increasingly appreciated connection between oncogenic KRAS signaling and regulation of the immune microenvironment, key experiments should be repeated in an independent model harboring mutant KRAS.

We appreciate the reviewer's insightful suggestion to improve our manuscript. To address this concern, we performed additional experiments using the KPC001 cell line (Pdx1-Cre, lox-stop-lox-KrasG12D/+, lox-stop-lox-tp53R172H/+). The murine pancreatic orthotopic model was produced using the KPC001 cells and mice were administered gemcitabine or Ate-Grab+gemcitabine (Supplementary Fig. 9b). We then confirmed both the anti-tumor and anti-fibrotic effects of Ate-Grab, comparable to those of KRAS-wild type Pan02 tumors, as shown below.

Figure 8

Figure 8 | (a) Representative ultrasonographic images for measuring orthotopic KPC tumor growth *in vivo*. (b) Tumor volumes measured by ultrasonography and compared across indicated treatment groups (Con, n=6; Gem, n=5; ATG+Gem, n=7). Con, control; Gem, gemcitabine; ATG+Gem, Ate-Grab and gemcitabine cotreatment. ***P<0.001; two-way ANOVA. (c) Representative images of KPC tumors from indicated treatment groups. Ruler scale, 1 mm. (d) Tumor weights measured and compared across indicated treatment groups (Con, n=6; Gem, n=5; ATG+Gem, n=7). Data are presented as the mean ± SEM. *P<0.05, ***P<0.001, one-way ANOVA. (e) Representative SHG images of KPC tumors from each treatment group. Scale bars, 100 μm. (f) Average percentages of collagen+ area out of total area in tumors from each treatment group. Data from randomly selected fields of view were analyzed (Con, n=48; Gem, n=40; ATG+Gem, n=56). Data are presented as the mean ± SEM. ***P<0.001, one-way ANOVA. (g) Flow cytometry gating strategy to evaluate VEGF-related receptor expression in KPC tumor-infiltrated CAFs. Cells gated in red box were considered positive for the indicated receptors. (h) Average percentages of NRP1-, VEGFR1-, or VEGFR2-expressing cells and triple negative cells for these receptors over total KPC tumor-infiltrated CAFs. (i, j) Average percentages of (i) total CAFs and (j) CD141⁺ MHC II⁻ CAFs in the KPC tumor microenvironment of each treatment group, measured by flow cytometry. Data from six mice from the control group, five mice from the gemcitabine group, and seven mice from the Ate-Grab+gemcitabine group were analyzed. (k, l) Average percentages of (k) CD3⁺ T cells, (l) CD4⁺ T cells, CD8⁺ T cells and Tregs in the KPC tumor microenvironment of each treatment group, measured by flow cytometry (Con, n=6; Gem, n=5; ATG+Gem, n=6). Data are presented as the mean ± SEM. *P<0.05, **P<0.01, ***P<0.001, one-way ANOVA.

Supplementary Figure 9 | (b) Experimental treatment scheme. Murine orthotopic pancreatic models were generated by implanting KPC001 cells (5×10^5 cells/mouse) into the pancreas of 7–10-week-old C57BL/6 mice. When the tumor volume reached 50–100 mm³, tumor-bearing mice were intraperitoneally treated with either PBS (control), gemcitabine (50 mg/kg, every 3.5 days), or gemcitabine with Ate-Grab (10 mg/kg, every 3.5 days). **(c)** Flow cytometry gating strategy to evaluate PDPN⁺ CAF in KPC tumor and CAF subsets defined by MHC-II and CD141 expression. **(d)** Average percentages of CD141⁺MHC-II⁻ cells in total CAFs were measured and compared across indicated treatment groups (Con, n=6; Gem, n=5; ATG+Gem, n=7). Data are presented as the mean \pm SEM. **P<0.01; one-way ANOVA. **(e)** Average percentages of CAF subsets in total tumor-infiltrating live cells were measured and compared across indicated treatment groups (Con, n=6; Gem, n=5; ATG+Gem, n=7). Data are presented as the mean \pm SEM; one-way ANOVA. **(f)** Average percentages of B cells and NK cells in total tumor-infiltrating live cells were measured and compared across indicated treatment groups (Con, n=6; Gem, n=5; ATG+Gem, n=6). Data are presented as the mean \pm SEM; one-way ANOVA.

We have further detailed all related results in the Results section as below. (p. 17, highlighted in yellow; Fig. 8a-I; Supplementary Fig. 9b-f)

“KRAS mutation accounts for more than 90% of all human PDAC cases⁵⁶. As Pan02 tumor is a KRAS-wild type⁵⁷, we verified the therapeutic efficacy of Ate-Grab in KRAS-mutated murine PDAC. Orthotopic murine pancreatic cancer models were generated using KPC001 cell line derived from GEMM (Pdx1-Cre, lox-stop-lox-KrasG12D/+, lox-stop-lox-tp53R172H/+)⁵⁸, and the mice were divided into three experimental groups based on the different treatment regimens (Supplementary Fig. 9b). Similar to the Pan02 tumor, KPC001 tumor volumes and weights were significantly lower in the Ate-Grab+gemcitabine treatment group than in the gemcitabine and untreated (control) groups (Fig. 8a-d). In addition, we validated that gemcitabine induced tumor fibrosis, which was effectively suppressed by co-treatment with Ate-Grab (Fig. 8e, f). With the

expression of VEGF-related receptors validated in KPC001 tumor-infiltrated CAFs (Fig. 8g, h; Supplementary Fig. 9c), flow cytometry analysis confirmed that gemcitabine treatment increased total CAF% in the TME and co-treatment with Ate-Grab significantly decreased CAFs (Fig. 8i). Notably, further analysis with representative markers of CAF subsets revealed that CD141⁺ MHCII⁻ CAFs, defined as CAF-2, were dramatically diminished in response to Ate-Grab co-treatment, while other CAF subsets showed no statistically significant difference between treatments (Fig. 8j; Supplementary Fig. 9c-e). Furthermore, higher infiltration of CD3⁺ T cells, especially CD8⁺ T cells, was observed in response to Ate-Grab and gemcitabine co-treatment, while no significant difference was identified in B and NK cell populations (Fig. 8k, l; Supplementary Fig. 9f). Overall, validation analysis with KPC001 orthotopic murine PDAC indicated that Ate-Grab has a comparable therapeutic effect on KRAS-mutated PDAC as that on KRAS-wild type PDAC.”

2. The results shown in ED Figure 1a-c are nice, but a change in CAF phenotype should be assessed (i.e., analysis of pSMAD and pSTAT3 within the CAF compartment) beyond just analyzing their abundance.

We appreciate the reviewer's valuable suggestion. To assess a change in CAF phenotypes, we examined the pSMAD2,3 and pSTAT3 using flow cytometry as the reviewer kindly recommended. After surface staining single cell suspensions acquired from the tumor, we performed pSMAD2 (Bioss, bs-3420R), pSMAD3 (Bioss, bs-3425R), or pSTAT3 (Biolegend, 651010) staining according to the manufacturer's fixation/permeabilization procedure. The results indicated that pSTAT3 significantly increased in the gemcitabine mono-treatment group, which decreased in response to Ate-Grab co-treatment, while pSMAD2 and pSMAD3 showed no significant difference. We have now presented these results in manuscript as follows. (p. 11, highlighted in yellow; Supplementary Fig. 4e-g)

“We examined whether gemcitabine or Ate-Grab treatment affected signaling pathways related to CAF activation. Gemcitabine increased the expression of p-STAT3, and co-treatment with Ate-Grab significantly reversed p-STAT3 expression in CAFs (Supplementary Fig. 4e, f). Ate-Grab also showed a tendency to reduce p-SMAD2 and p-SMAD3 expression, albeit not significant (Supplementary Fig. 4g).”

Supplementary Figure 4 | (e) Flow cytometry gating strategy to evaluate PDGFR α ⁺ CAF and representative plots of pSTAT3 expression in PDGFR α ⁺ CAF. (f, g) Average MFI (mean fluorescence intensity) values of (f) p-STAT3, (g) p-SMAD2 and p-SMAD3 compared across different treatment groups (Con, n=6; Gem, n=7; ATG+Gem, n=8). Data are presented as the mean \pm SEM. *P<0.05, **P<0.01; one-way ANOVA.

3. The clinical data presented in Figure 2 are very nice but could be further developed to better support the authors' model. Is there a correlation between collagen abundance and PIGF levels? Is prior gemcitabine treatment status known for these patients, and if so, do this treatment influence collagen and/or PIGF abundance? These analyses would help tie these clinical results to the preclinical data in Figure 1.

We agree with the reviewer's insightful comments. To address the comments, we utilized the same patient sample data shown in Supplementary Fig. 2 (Fig. 2 in initial submitted version). We confirmed that there is a strong positive correlation (P -value=0.002, r value=0.6586) between PIGF and collagen abundance in human PDAC tissues. However, unfortunately, all analyzed patients were those who had not received any chemotherapy such as gemcitabine. The result is now depicted in main figure as follows. (p. 7, highlighted in yellow; Fig. 2e)

“Furthermore, tumor fibrosis demonstrated a significant correlation with PIGF and VEGF expression ($r = 0.6586$, $p = 0.002$; $r = 0.5955$, $p = 0.006$ respectively) (Fig. 2e, f).”

Fig.2 | Correlation analyses between tumor fibrosis and (e) PIGF+ area%. Data from 20 PDAC patient samples were analyzed (n=20).

4. Figure 3 nicely demonstrates that Ate-Grab inhibits VEGF-A, PIGF, and PD-1/PD-L1 signaling, but what about specificity? Genetic inhibition experiments would be helpful to address this question.

We appreciate the reviewer's careful inspection. As the reviewer mentioned, target specificity of drugs can be clearly verified by genetic inhibition experiments. However, in the case of our platform, VEGF-Grab which is a backbone of Ate-Grab, has already been validated through a previous study (Lee et al., 2018, Biomaterials). Furthermore, we successfully evaluated binding affinity to target antigens of Ate-Grab, which is equivalent to template drugs (VEGF-Grab, Atezolizumab). Therefore, the concerns regarding target specificity of drugs were considered minimal. Kindly refer to Fig. 3c, Supplementary Fig. 3b.

5. CAF phenotypes in response to Ate-Grab should be analyzed to accompany the results presented in Figure 4 (p -STAT3, p -SMAD), which should be doable using tumor tissues already obtained from prior experiments.

We appreciate the Reviewer's valuable suggestion. We assessed pSMAD2, pSMAD3, and pSTAT3 expression in response to Ate-Grab. Kindly refer to our answers for comment 2.

6. In Figure 4g,h and ED Figure 4b,c, PDGFR β is insufficient to define pericytes, and the authors should co-stain for a more specific pericyte marker such as RGS5 or NG2.

We appreciate the reviewer's valuable comment. In addition to PDGFR β , we overlaid NG2 with CD31 to define pericytes specifically around vessels. (p. 10, highlighted in yellow; Fig. 4g-j)

“As tumor fibrosis is closely related to vessel normalization^{10,11}, we evaluated Pan02 tumor vasculature through immunofluorescence analysis of CD31 as an endothelial cell marker and NG2 or PDGFR β as a pericyte marker. The Ate-Grab-treated group, which exhibited decreased tumor fibrosis (Fig. 4d, e), had greater NG2⁺ or PDGFR β ⁺ pericyte coverage (Fig. 4g, h; Supplementary Fig. 4b, c).”

Figure 4 | (g) Representative immunofluorescence (IF) images staining for NG2 (green) and CD31 (red) in Pan02 tumors treated with different agents. Yellow indicates the co-expression of NG2 and CD31. Scale bar, 100 μ m. **(h)** Quantifications of yellow regions (NG2⁺CD31⁺) in IF data were analyzed by ImageJ. Data from 28 randomly selected fields of view per group were analyzed (n=28). **P<0.01, One-way ANOVA.

7. The conclusion on lines 199-201 is not supported by the data. If the authors wish to make claims about vessel normalization in response to Ate-Grab treatment, functional perfusion experiments are needed.

We appreciate the reviewer's insightful comment. Considering the reviewer's comment, we performed a perfusion assay using 2000 kDa high molecular dextran to evaluate vessel normalization, and successfully confirmed that the dextran⁺ area significantly increased in the Ate-Grab treatment group. (p. 10, highlighted in yellow; Fig. 4i, j)

“Furthermore, a perfusion assay using 2000 kDa dextran revealed that only Ate-Grab significantly recovered vessel perfusion (Fig. 4i, j).”

Figure 4 | (i) Representative immunofluorescence (IF) images staining for CD31 (red) and Dextran (green) in Pan02 tumors treated with different agents. Yellow indicates the co-expression of CD31 and Dextran. Scale bar, 100 μ m. **(j)** Quantifications of Dextran⁺/CD31⁺ in IF data were analyzed by ImageJ. Data from 28 randomly selected

fields of view per group were analyzed (n=28). *P<0.05, **P<0.01, One-way ANOVA.

8. In Figure 4f, how were myCAFs identified/defined?

We referred to the gating strategy of CAFs proposed by Elyada et al. (2019) (Cancer Discovery). We have shown the gating strategy plots of CAF and its three subtypes in Supplementary Figures. All CAFs in the paper were defined as the FSC-SSC/Single cells/Live/CD45⁻/CD31⁻ Cancer cell/PDGFR α ⁺ population. After gating the CAF population, iCAF, myCAF, and apCAF were identified with marker expressions of Ly6C and I-A/I-E (MHC-II). We have now presented the gating strategy of CAF as below (for the reviewer's consideration only). However, we note here that we have removed the CAF subset data (myCAF, iCAF, and apCAF) from our flow cytometry results, considering reviewer 2's comment that myCAF is generally low for PDGFR α , which was used as a marker for total CAFs in Pan02 orthotopic tumor.

Figure 1 (for reviewer only) | Gating strategy for flow cytometry analysis of cancer-associated fibroblast (CAF) and its subsets in the tumor microenvironment (TME). CAFs were gated as FSC-SSC/Live/CD45⁻/CD31⁻EPCAM⁺/PDGFR α ⁺ cells. myCAF, iCAF and apCAF were defined with differential expression of I-A/I-E and Ly6C.

9. It is important to further assess the CAF-2 population displayed in Figure 6 to confirm that these are CAFs as opposed to PDAC cells with a mesenchymal phenotype. Does this cell population express other, well-established CAF markers like Acta2, Fap, and Pdpn?

In the Pan02 tumor, our scRNA-seq data identified that cancer-associated fibroblasts did express Acta2, but not Fap and Pdpn, as shown below (for the reviewer's consideration only). We also tested the protein expression of Pdpn with flow cytometry and confirmed that Pan02 tumor-infiltrated CAFs hardly expressed Pdpn in proteins. Several previous studies on CAF have reported that some CAF subsets rarely express Fap or Pdpn (Nurmik et al., 2019, *International Journal of Cancer*). Please note that we have now added scRNA-seq data of an untreated Pan02 tumor sample, considering the suggestions of reviewers 1 and 4.

Figure 2 (for reviewer only) | Feature plot showing the expression of representative CAF marker genes in our orthotopic Pan02 tumor microenvironment.

10. The report of CD141 expression on CAFs is quite surprising and should be further validated. Figure 6g makes it seem that nearly all PDGFR α ⁺ CAFs express CD141. Can this be demonstrated in tumor tissues by co-staining for CD141 and a CAF marker by IHC?

We apologize for the ambiguity caused by the inappropriate choice of figure. The expression of CD141 over PDGFR α ⁺ CAFs varies greatly for each individual tumor (40–85%). We presented the related data here only for the reviewer’s consideration. In addition, we replaced the original FACS plot with a more representative plot.

Figure 3 (for reviewer only) | CD141 expression over PDGFR α ⁺ CAFs.

Figure 6 | (i) Flow cytometry validated the phenotype of CAF-2 at the protein level, defined as FSC-SSC/CD45⁻/CD31⁻/EpCAM⁻/PDGFR α ⁺/MHCII⁻/CD141⁺ (left upper quadrant).

11. (Minor) Are the results in Figure 1e,f from tumor homogenates? Or from PDAC cells in vitro? This should be stated in the legend and/or results section.

We apologize for missing the information. Figures 1e and f are from Pan02 tumor tissue homogenates. We have now stated this in the legend of Figure 1. In addition, we validated the protein levels of PIGF and VEGFA using ELISA, considering other reviewers’ comments. We have now added the ELISA results in the revised manuscript. (p. 5, highlighted in yellow; Fig. 1e, f)

“To explore the significance of PIGF in pancreatic cancer, we measured its mRNA and protein levels in our orthotopic PDAC model and observed an increase after gemcitabine treatment,

while the level of VEGF-A increased slightly (Fig. 1e, f).”

Figure 1 | (e) Relative *Plgf* and *Vegfa* mRNA expression levels of Pan02 tumor tissue homogenates measured by qRT-PCR. Data from four technical repeats for control group and five technical repeats for gemcitabine group were analyzed. (f) Relative *Plgf* and *Vegfa* protein expression levels of Pan02 tumor tissue homogenates measured by ELISA. Data from seven technical repeats (n = 7) were analyzed.

Reviewer #4 (Remarks to the Author): with expertise in scRNAseq, cancer associated fibroblasts.

Kim et al. describe a PlGF/VEGF - PlGF/VEGF receptor axis that induces collagen deposition in pancreatic cancer via cancer-associated fibroblasts. They develop a multi-paratopic VEGF decoy receptor (Ate-Grab) to target PlGF/VEGF within the PD-L1-enriched TME and investigate its impact on the cancer-associated fibroblast landscape in the orthotopic Pan02 tumor model. The authors observe anti-tumor/anti-fibrotic effects and identify a CD141+ CAF population from single-cell RNA sequencing data, which is proposed to be responsible for the therapeutic effect of the Ate-Grab. The manuscript is overall well written and the majority of conclusions are supported by the data. The manuscript is of interest to the CAF research field, but would benefit from additional experiments/analyses/clarifications I am outlining below.

We appreciate the reviewer's overall positive assessment of our paper and hope that our revisions have assuaged any remaining concerns.

Major

1) It is quite important to understand the baseline expression phenotypes in the single-cell experiment. How does CAF heterogeneity look like in untreated animals with orthotopically implanted tumors?

We appreciate the reviewer's comments to improve our scRNA-seq analysis throughout the manuscript. As the reviewer suggested, we added scRNA-seq data of untreated Pan02 tumors and analyzed the further heterogeneity of CAF in the Pan02 tumor microenvironment (CAF-1–CAF-6). Based on the re-clustered data with untreated samples, we have revised the manuscript as follows (p.12, highlighted in yellow; Fig. 6a-l, Supplementary Fig. 6a-g, Supplementary Fig. 8f-m).

“To investigate the changes occurring in the TME cellular composition upon treatment, we compared the single-cell transcriptomes of untreated (Control), gemcitabine-treated (Gem), and Ate-Grab+gemcitabine-treated (ATG + Gem) Pan02 tumors. Live cells from five untreated tumors, five gemcitabine-treated tumors, and four Ate-Grab + gemcitabine-treated tumors were separately pooled and subjected to single-cell RNA sequencing (scRNA-seq) analysis (Supplementary Fig. 6a). We sorted 35,697 cells based on a previously reported quality control scheme and identified nine distinct cell clusters (except for a low-quality cluster and doublet cluster) from these cells using a graph-based clustering method (Fig. 6a) ⁴⁵.”

Figure 6

Figure 6 | Ten days after Pan02 (5×10^5 cells) implantation, tumor-bearing mice were intraperitoneally treated with PBS (Con; n=5), gemcitabine (Gem; n=5) or a combination of Ate-Grab and gemcitabine (ATG + Gem; n=4). Mice were sacrificed for single-cell RNA sequencing on day 24, 2 days after the fifth drug injection. **(a)** Uniform manifold approximation and projection (UMAP) plot for orthotopic Pan02 tumor-infiltrating cells (35,697 cells). Nine cell types (except for doublets and low-quality cells) were assigned based on the expression of marker genes. **(b)** Unbiased clustering of CAF subsets revealed that combination therapy with Ate-Grab and gemcitabine (co-treatment) mostly depleted CAF-2. **(c)** Top 10 differentially expressed genes (DEGs) in six CAF subpopulations. **(d)** The number of each CAF subpopulation was quantified and compared between the three groups (Control, untreated; Gem, gemcitabine mono-treated; ATG + Gem, Ate-Grab and gemcitabine co-treated). **(e)** RNA velocity vector field for CAF differentiation indicated by streamlines. **(f)** Left: Kaplan-Meier overall survival curve of 150 pure PDAC patients grouped based on the expression of the top 30 CAF-2 DEGs (AUC value>0.7, pct.1-pct.2>0.05). Right: Kaplan-Meier overall survival curve of 73 pure PDAC patients with low tumor cellularity grouped based on the expression of the top 30 CAF-2 DEGs. HR: hazard ratio. **(g)** Feature plot of the average expression of the top 30 CAF-2 DEGs

scored in the mother plot (Pan02 tumor-infiltrated cells) to show CAF specificity of CAF-2 DEGs. (h) Dot plots of Cd141 and Cd74 expression in the CAF subpopulations. (i) Flow cytometry validated the phenotype of CAF-2 at the protein level, defined as FSC-SSC/CD45⁻/CD31⁻/EpCAM⁻/PDGFR α ⁺/MHCII⁻/CD141⁺ (left upper quadrant). (j) Bar plot showing each treatment effect on CAF-2 composition in orthotopic Pan02 tumor. Data from seven samples of control group, eight samples of gemcitabine group and nine samples of Ate-Grab+gemcitabine group were analyzed. Data are presented as the mean \pm SEM. One-way ANOVA, *P<0.05. (k) Feature plots of Cd8b1 expression in untreated (left), gemcitabine-treated (middle) and Ate-Grab + gemcitabine-treated (right) tumor-infiltrating T/NK populations. (l) Stacked bar plot of the number of lymphocyte subsets in untreated (black), gemcitabine-treated (blue), and Ate-Grab+gemcitabine-treated (yellow) Pan02 tumors.

Supplementary Figure 6 | (a) Flow cytometry gating strategy for sorting tumor-infiltrating live cells. High cell viability should be ensured for successful single-cell RNA sequencing. (b) Dot plot showing the expression of multiple marker genes for precise cell type annotation prior to downstream analysis. (c) Dot plot and (d) feature plot showing VEGF-related receptor gene expression in Pan02 tumor-infiltrating cells. (e) Unbiased clustering of CAFs revealed six different CAF subpopulations (CAF-1, CAF-2, CAF-3, CAF-4, CAF-5, and CAF-6). (f) Feature plot showing *Inhba* expression in Pan02 CAFs. (g) Enrichment pathway analysis with the top 30 DEGs of each CAF subpopulation (*Enrichr*).

Supplementary Fig. 8 | (f) Upper: Unbiased clustering of the integrated Pan02 tumor-infiltrating neutrophils (14,620 cells). Lower: Bar plot of the number of neutrophil subsets in untreated (Control), gemcitabine-treated (Gem), and Ate-Grab+gemcitabine-treated (ATG + Gem) Pan02 tumors. (g) Dot plot showing *Cd274* (PD-L1) expression in Pan02 tumor neutrophil subsets. (h) Enrichment pathway analysis with the top DEGs of each neutrophil subpopulation (*Enrichr*). (i) Unbiased clustering of the integrated Pan02 tumor-infiltrating T/NK cells (7,622 cells). (j) Dot plot of the indicated features in each T/NK subpopulation. (k) Feature plots of the indicated mRNA features in the integrated T/NK subpopulations. (l) Heatmap of the indicated mRNA expression in three CD8⁺ T subpopulations. (m) Violin plots of *Gzmb*, *Ifng*, and *Cd28* expression in gemcitabine and Ate-Grab+gemcitabine treatment groups.

2) I'm confused by the statement in the discussion: "Of note, the characteristics of CD141⁺ CAFs reported herein are generally different from those previously described for myCAF_s. That is, CD141⁺ CAF_s secrete various factors, including INHBA, which can activate other CAF populations to further induce desmoplasia." *Inhba* is expressed by myCAF_s in human patients and in murine PDAC models (Elyada et al., 2019, CD). In order to properly align the subsets identified in this manuscript with the CAF subtypes described in the literature, I would like to ask the authors to perform a more rigorous comparison between their subtypes and the subtypes identified in other studies. It will be important to

understand the CAF heterogeneity in the orthotopic model used by the authors compared to heterogeneity in GEMM models without any treatment, and then relate these findings to the changes observed with Gem alone and ATG + Gem.

We appreciate the reviewer’s insightful suggestion to improve our manuscript. As the reviewer suggested, we comprehensively compared our Pan02 CAF subtypes with previously published CAF scRNA-seq data. In comparison with Elyada et al. (2019), most Pan02-infiltrated CAF subsets (except for CAF-2) highly expressed MHC-II-related genes, which are representative signatures of apCAF, while each subset clustered further represented its own unique signature. CAF-5 shares several myCAF signatures, including *Spp1*, *Col12a1*, and *Ecscr*, while CD141⁺ CAF-2 expresses some iCAF DEGs, such as *Ogn*, *Sema3c*, and *Mfap5*. However, all CAF subsets from Pan02 tumor did not show one-to-one correspondence with iCAF, myCAF, and apCAF from Elyada et al. (2019). In comparison with Dominguez et al. (2020), our CD141⁺ CAF-2 shared some markers of c0 CAF, including *Gstm1*, *Ogn* and *Sema3c*, while several markers of myCAF-like c2 CAF were conserved in CAF-5. Notably, CAF-4 presented a high similarity with c8 CAF, showing differential expression of most c8 CAF signatures.

Supplementary Figure 6 | (h) Unbiased clustering (UMAP embedding) of KPC tumor scRNA-seq data from Elyada et al., 2019 (fibroblast enriched dataset). (i) Left: UMAP of zoom-in clustering of CAF sorted from (h). Right: Heatmap showing marker gene expression of myCAF, iCAF, and apCAF. (j) Dot plot showing the average expression of myCAF (left), iCAF (middle), and apCAF (right) marker genes (Elyada et al., 2019) in orthotopic Pan02 CAF populations. (k) Unbiased clustering (UMAP embedding) of KPP tumor scRNA-seq data from Dominguez et al., 2020. (l) Left: UMAP of zoom-in clustering of CAF sorted from (k). Right: Heatmap showing marker gene expression of c0 CAF, c1 CAF, c2 CAF, and c8 CAF. (m) Dot plot showing the average expression of c0 CAF (left), myCAF-like c2CAF (middle), and c8 CAF (right) marker genes (Dominguez et al., 2020) in orthotopic Pan02 CAF populations.

For improved clarity, we have detailed this in the revised manuscript and supplementary figures as follows. (p. 13, highlighted in yellow; Supplementary Fig. 6h-m)

“To further validate our CAF subpopulations, we compared the signatures of our Pan02 CAF subpopulations to those of previously published fibroblast-enriched datasets (Supplementary Fig. 6h–m)^{35,50}. In comparison with Elyada et al. (2019), most Pan02-infiltrated CAF subsets except for CAF-2 highly expressed MHC-II-related genes that were representative signatures of apCAF, while each clustered subset further represented its own unique signature

(Supplementary Fig. 6j). CAF-5 shared several myCAF signatures including *Spp1*, *Col12a1*, and *Ecscr*, while CAF-2 expressed some iCAF DEGs, such as *Ogn*, *Sema3c*, and *Mfap5* (Supplementary Fig. 6j). However, all CAF subsets from the Pan02 tumor hardly showed one-to-one correspondence with iCAF, myCAF, and apCAF35. For comparison with Dominguez et al. (2020), we sorted pure CAFs from the data, and re-clustered the sorted CAF cells, so that the DEGs of each CAF subset could be calculated for pure CAF cells (not contaminated with tissue fibroblasts or mesothelial cells) (Supplementary Fig. 6k, l). The comparison revealed that our CAF-2 shared some markers of c0 CAF, including *Gstm1*, *Ogn*, and *Sema3c*, while several markers of myCAF-like c2 CAF were conserved in CAF-5 (Supplementary Fig. 6m). Notably, CAF-4 represented a substantial similarity with c8 CAF, showing the most c8 CAF signatures (Supplementary Fig. 6m).”

3) CAF-1 and CAF-2 look like they exist on a spectrum rather representing two clear distinct subsets of cells. This suggests they differ in their activation programs and can transition between states. The increase in CAF-1 in ATG + Gem seems mostly driven by increased INF-signaling into CAFs, which can induce expression of *Gbps*, antigen presentation machinery, etc. That makes interpretation slightly more complicated, as the results could be mostly a readout of increased infiltration with cytotoxic T cells that’s described by the authors (thus shifting CAFs from CAF2 into CAF1 phenotypes). To understand the relationships between CAF-1 and CAF-2 better, I would like to suggest the authors perform RNA velocity (or) non-velocity-based pseudo time reconstruction analysis to understand program activity associated with transitions between activation states. How does treatment affect the transitions?

We appreciate the helpful comments to improve the manuscript. We assessed RNA velocity on Pan02-infiltrated CAF, revealing that CAF-1 differentiates into CAF-3, and then into CAF-2. Gemcitabine treatment facilitates the transition towards CD141⁺ CAF-2, and Ate-Grab+Gemcitabine co-treatment attenuates the tumor-promoting transition. The RNA velocity data has been added in the Result section and main figure as follows. (p. 13, highlighted in yellow; Fig. 6e)

“To gain insight into cellular transition statuses of CAFs, we performed RNA velocity analysis on our Pan02 CAF subsets and found that CAF-1 differentiates toward CAF-2. These results indicate that Ate-Grab inhibits the cellular transition toward CAF-2 (Fig. 6e).”

Figure 6 | (e) RNA velocity vector field for CAF differentiation indicated by streamlines.

4) Given that PD-L1 is not only expressed by CAFs in the TME, as shown in figure 2I, I would suggest the authors at least use their single-cell dataset to investigate a potential direct (or indirect) effect of Ate-Grab on myeloid cells.

In response to the reviewer’s valuable suggestion, we analyzed the indirect effect of Ate-Grab on myeloid cells. We identified that some neutrophil subsets express PD-L1 (Cd274), however Ate-Grab did not further decrease PD-L1-expressing Neu-2 and Neu-5 subpopulations of neutrophils, which featured type I interferon-sensitized pathways. Meanwhile, Ate-Grab treatment highly increased the Neu-4 subpopulation that is involved in inflammatory responses, such as TNF-alpha signaling via NF-κB. We have presented the related data in a Supplementary Figure. (p. 15, highlighted in yellow; Supplementary Fig. 8f-h)

“Finally, we explored immune populations indirectly affected by Ate-Grab treatment. In the orthotopic Pan02 TME, PD-L1 was expressed not only in CAFs, but also in immune cells, especially myeloid cells (Supplementary Fig. 2k). Since the scRNA-seq data indicated Cd274 (PD-L1) mRNA expression in some neutrophils, we investigated neutrophil subclusters (Supplementary Fig. 8f). Ate-Grab did not further decrease PD-L1-expressing Neu-2 and Neu-5 subpopulations of neutrophils, which are characterized by type I interferon-sensitized pathways (Supplementary Fig. 8f–h). Meanwhile, it is notable that Ate-Grab treatment highly increased the Neu-4 subpopulation that could be involved in inflammatory responses, such as TNF-alpha signaling via NF-κB (Supplementary Fig. 8f–h).”

Supplementary Figure 8 | (f) Upper: Unbiased clustering of the integrated Pan02 tumor-infiltrating neutrophils (14,620 cells). Lower: Bar plot of the number of neutrophil subsets in untreated (Control), gemcitabine-treated (Gem), and Ate-Grab+gemcitabine-treated (ATG + Gem) Pan02 tumors. (g) Dot plot showing Cd274 (PD-L1) expression in Pan02 tumor neutrophil subsets. (h) Enrichment pathway analysis with the top DEGs of each neutrophil subpopulation (*Enrichr*).

5) The authors focus strongly on the PIGF/VEGF - PIGF/VEGF axis when speaking about tumor fibrosis in PDAC. I would like to suggest to discuss this axis in the context of TGFβ signaling at least in the discussion section, given the prominent role of TGFβ signaling in pancreatic cancer, tumor fibrosis, and cancer immunotherapy response.

We appreciate the reviewer's insightful suggestion. As recommended by the reviewer, we have elucidated the TGF-β signaling of pancreatic cancer in the Discussion section as follows. (p. 20, highlighted in yellow).

“TGF-β reportedly activates CAFs and promotes tumor fibrosis in pancreatic cancer^{59,60}. It also

promotes epithelial-mesenchymal transition (EMT) and exacerbates tumor invasiveness and metastasis⁶¹. Furthermore, TGF- β suppresses adoptive immune response by altering T cell differentiation toward Th2 rather than Th1 and affects innate immune response by facilitating M2 transition of macrophages⁶²⁻⁶⁴. These are considered as important reasons why PDAC rarely responds to the current immunotherapies. Although the relationship between the TGF- β signaling and PIGF signaling pathways has not been comprehensively examined, TGF- β reportedly increases PIGF expression, and PIGF activates the TGF- β signaling pathway in different cell types, but not CAFs^{65,66}. Herein, we revealed that gemcitabine increased the number of CD141⁺MHCII⁻ CAF-2, which exhibits enhanced TGF- β signaling pathway, and Ate-Grab dramatically reduced CAF transition toward CAF-2 by blocking the PIGF effect. Our flow cytometry data illustrated that compared to gemcitabine treatment alone, Ate-Grab co-treatment decreased the expression of pSTAT3 in CAFs, which has reported essential for TGF- β -induced transcription⁶⁷⁻⁶⁹, and also showed a tendency to decrease pSMAD2 and pSMAD3, which are representative indicators of TGF- β signaling activation^{70,71}.”

6) The authors cite their unpublished work in reference 37 to establish six PDAC subtypes. Given that the cited manuscript has not been formally reviewed, relying on the proposed subtypes (with limited ways to understand the characteristics of those proposed subtypes for the reader of this manuscript) for survival associations seems not the most desirable strategy to me. In order avoid relying on the subtypes, I suggest that the authors directly analyze the impact of the expression/%area of PIGF/VEGF-A ligands as well as their receptors on patient survival instead of using the subtypes as an intermediate tool.

We appreciate the reviewer's helpful suggestions. As the reviewer suggested, we calculated the direct correlation between patient survivals and the % area of PIGF+/VEGF-A+ or α -SMA+ VEGF, expression, not restricted to PDAC subtype. The result showed that PIGF expression is associated with decreased overall survival (OS) and disease-free survival (DFS), and VEGF-A expression is associated with decreased OS. Furthermore, we also verified that α -SMA+ area expressing NRP1 or VEGFR1 showed negative correlation with OS and DFS of pancreatic cancer patients. These new analyses are illustrated in the Results section (p. 7, highlighted in yellow; Fig. 2b-d, 2h-j and Supplementary Fig. 2c-e)

“Tumor fibrosis was negatively correlated with patient prognosis ($r = -0.6499$, $p = 0.002$) as was PLGF and VEGFA expression (Fig. 2a-d; Supplementary Fig. 2b, c). Furthermore, tumor fibrosis demonstrated a significant correlation with PIGF and VEGF expression ($r = 0.6586$, $p = 0.002$; $r = 0.5955$, $p = 0.006$ respectively) (Fig. 2e, f).

Double-staining IHC analysis revealed that CAFs expressed PIGF/VEGF receptors (Fig. 2g), and NRP1⁺ CAF, VR1⁺ CAF, and VR2⁺ CAF were strongly negatively correlated with patient prognosis ($r = -0.7199$, $p < 0.001$; $r = -0.6898$, $p < 0.001$; $r = -0.7131$, $p < 0.001$, respectively) (Fig. 2h-j; Supplementary Fig. 2d, e), while NRP1⁺ CAF and VR1⁺ CAF positively correlated with tumor fibrosis measured by Masson's trichrome⁺ area ($r = 0.594$, $p = 0.006$; $r = 0.5353$, $p = 0.015$ respectively) (Fig. 2k; Supplementary Fig. 2f, g).”

Figure 2 | Correlation analyses between PIGF⁺ area% (detected by IHC staining) and (b) overall survivals or (c) disease free survivals of PDAC patients, between (d) VEGFA⁺ area% (detected by IHC staining) and overall survivals of PDAC patients. Correlation analyses between overall survivals of PDAC patients and (h) NRP1⁺α-SMA⁺ area%, (i) VR1⁺α-SMA⁺ area%, or (j) VR2⁺α-SMA⁺ area%. Data from 20 PDAC patient samples were analyzed (n=20)

Supplementary Figure 2 | Correlation analyses between disease free survivals of human PDAC patients and (c) VEGFA⁺ area%, (d) NRP1⁺α-SMA⁺ area% or (e) VR1⁺α-SMA⁺ area%. Data from 20 PDAC patient tumor samples were analyzed (n=20).

7) There is no direct evidence for CD141⁺ CAFs being “responsible for the therapeutic effects of Ate-Grab”, only an association (of these CAFs being reduced upon ATG + Gem). I would like to suggest rewording these statements to reflect the lack of direct evidence (as in: Does depletion of these CAFs have the same therapeutic effect?).

We appreciate the reviewer's valuable suggestion. Because of the lack of direct evidence to prove the effect of CD141⁺ CAFs in response to Ate-Grab treatment as mentioned by the reviewer, we have revised the sentence in the Abstract as below. (p. 2, highlighted in yellow).

"Single-cell RNA sequencing identified that the CD141⁺ CAF population was reduced upon Ate-Grab and gemcitabine combination treatment"

Minor

It is unclear to me how myCAFs were defined in figure 4f and 5d (legend does not explain figure for 5d).

We referred to the gating strategy of CAFs proposed by Elyada et al. (2019) (Cancer Discovery). We have shown the gating strategy plots of CAF and its three subtypes in a Supplementary Figure. All CAFs in the paper were defined as FSC-SSC/Single cells/Live/CD45⁻/CD31⁻Cancer cell⁻/PDGFR α ⁺ populations. After gating the CAF population, iCAF, myCAF, and apCAF were identified with marker expressions of Ly6C and I-A/I-E (MHC-II). We have now shown the gating strategy of CAF as below (for the reviewer’s consideration only). However, we note here that we removed the CAF subset data (myCAF, iCAF, and apCAF) from our flow cytometry results, considering reviewer 2’s comment that myCAF is generally low for PDGFR α , which we used as a marker of total CAFs in Pan02 orthotopic tumor.

Figure 1 (for reviewer only) | Gating strategy for flow cytometry analysis of cancer-associated fibroblast (CAF) and its subsets in the tumor microenvironment (TME). CAFs were gated as FSC-SSC/Live/CD45⁻/CD31⁻EPCAM⁻/PDGFR α ⁺ cells. myCAF, iCAF, and apCAF were defined with differential expression of I-A/I-E and Ly6C.

and batch effects were corrected using the “FindIntegrationAnchors” function.” The authors should define what the batches were and what effect they aimed to correct for. How was the experiment designed?

As three samples (Control, GEM, ATG+Gem) were sequenced separately, there was a batch effect between samples owing to technical variance. We performed batch correction with canonical correction analysis (CCA) for an initial dimension reduction, using the ‘FindIntegrationAnchors’ function of Seurat v3. Considering the reviewer’s comments, we have amended the Methods section with further details as follows. (p. 33, highlighted in yellow)

“To integrate the three samples sequenced separately, individual Seurat objects were merged and normalized with the “NormalizeData” function, and technical batch effects were corrected by canonical correction analysis (CCA) for an initial dimension reduction, using the “FindIntegrationAnchors” function of Seurat v3.”

“The top DEGs were selected via the vst method, and principal component analysis (PCA) was then performed on 2000 DEGs. We used the “FindClusters” function on 10–30 PCs with a resolution of 0.4–1.5 to cluster cells on the UMAP plot. “How did the authors evaluate which of the results generated should be used for downstream analysis? Why were a certain number of PCs chosen? Or a particular clustering resolution?”

First, we included the scRNA-seq data of the Control (untreated) sample and performed integration as the reviewers suggested. Therefore, we regenerated the UMAP plots with different resolutions and a different number of PCs. We have illustrated the altered information in the Methods section as shown below (p. 33, highlighted in yellow). To generate the UMAP plots, we referred to Seurat::ElbowPlot, which indicates the amount of variance exhibited by each principal component. We then determined the final number of PCs and resolution by considering the biological meaning suggested by the differentially expressed genes (DEGs) of each cluster with a significant *p*-value. Considering the reviewer’s comments, we have amended the Methods section with further details as follows. (p. 33, highlighted in yellow)

“The “FindClusters” function on 20–50 PCs with a resolution of 0.4–1.5 was used to cluster cells on the UMAP plot. To determine the number of PCs, Seurat:ElbowPlot served as a reference to confirm the amount of variance represented by each PC, and the biological interpretation of differentially expressed genes (DEGs) of each cluster was considered.”

“Gene set enrichment analysis revealed that epithelial-mesenchymal transition (EMT), which is involved in pathological fibrosis 53, was enriched in the CAF-2 subset (Extended Data Fig. 6f).” Given these are mesenchymal cells, the observation of EMT being upregulated does not seem surprising.

We thank the reviewer for the valuable comment. In addition to the EMT-related pathway, CD141+ CAF-2 was featured with the ‘TGF-beta regulation of extracellular matrix’ pathway and ‘Wnt signaling’ pathway. Therefore, we have now deleted the description related to the observation of EMT from the result section and amended as follows. (p. 13, highlighted in yellow)

“Gene set enrichment analysis revealed that the signaling pathways of ‘TGF-beta regulation of extracellular matrix’ and ‘Wnt signaling pathway’, which are activated in pathological fibrosis, were enriched in the CAF-2 subset (Supplementary Fig. 6g), while fibrosis-promoting secretory molecules including *Ogn*, *Prelp*, *Omd*, *Inhba*, and *Ecr4* were confirmed among the top 30 differentially expressed genes (DEGs) in CAF-2 (Fig. 6c, Supplementary Fig. 6f)”

Supplementary Figure 6 | (g) Enrichment pathway analysis with the top 30 DEGs of each CAF subpopulation (*Enrichr*).

“Human pancreatic cancer samples from GEO, EGA, and TCGA were used to evaluate the prognostic value of DEGs from CAF subclusters. Kaplan-Meier Plotter was used for survival analysis based on gene signature expression.” Please provide much more detail regarding the data analysis and sources, such as normalization, how signature scores were calculated, if certain patients’ samples were excluded,

...

As the reviewer suggested, we have now included more information regarding survival analysis in the Methods section as below. (p. 34, highlighted in yellow)

“Kaplan–Meier Plotter (www.kmplot.com) was utilized for survival analysis based on the gene signatures of populations of interest. With Kaplan–Meier Plotter ⁷⁰, we performed overall survival analysis with 150 pure pancreatic ductal adenocarcinoma patient samples acquired from the TCGA repository (TCGA-PAAD), not restricted to gender, race, stage, grade, and mutation burden. DEGs used for survival analysis were endowed with the same weight, applied with MAS-5 algorithm-based normalization and second scaling normalization, and the mean expression of the genes was calculated.”

In their TCGA survival analysis, the authors decide to use the top 40 upregulated genes from their mouse model and perform survival analysis in the TCGA PAAD dataset (how was a score for each penitent calculated?). I would strongly suggest to exclude the PNET samples in the PAAD TCGA cohort from this analysis. Furthermore, I would like to ask the authors to make a statement about the specificity of their markers. Markers were identified comparing expression within CAFs, but not to other cell types in the TME. So how specific are the genes they have chosen?

We appreciate the insightful comment from the reviewer. We have now excluded the following samples from TCGA-PAAD dataset (did not arise from pancreas/Neuroendocrine/Acinar cell carcinoma/Intraductal papillary mucinous neoplasm/Undifferentiated/Systemic treatment given to the prior/other malignancy/Samples without tumor cellularity information) and performed downstream analysis with pure PDAC patients as the reviewer suggested. We then confirmed that top 30 DEGs of CAF-2 (AUC value > 0.7, pct.1-pct.2 > 0.05) significantly affect the overall survival of 150 pure PDAC patients, while these genes are specific to CAF as validated in mother plots including the whole cell types in the Pan02 tumor microenvironment. To score values for each patient, the mean expression of the input genes was calculated. These results are depicted in the Methods and Results section as follows. (p. 34, 14 highlighted in yellow; Fig. 6f, g)

“To sort pure PDAC patients, we excluded the following samples from the TCGA-PAAD dataset; did not arise from pancreas/Neuroendocrine/Acinar cell carcinoma/Intraductal papillary mucinous neoplasm/Undifferentiated/Systemic treatment given to the prior/other malignancy/Samples without tumor cellularity information.”

“To determine the clinical value of CAF-2, we performed Kaplan–Meier overall survival analysis using publicly available PDAC patient data ⁵¹ (Fig. 6f; Supplementary Fig. 7a, b). The CAF-specific DEGs of CAF-2, showing higher expression in pancreatic tumors compared to normal tissues (Supplementary Fig. 8a), were negatively correlated with patient prognosis (HR=1.65), and showed a higher hazard ratio (HR=1.97) in PDAC patients with lower tumor cellularity (Fig. 6f, g), highlighting CAF-2 as a promising therapeutic target, especially for desmoplastic cancer.”

Figure 6 | (f) Left: Kaplan-Meier overall survival curve of 150 pure PDAC patients grouped based on the expression of the top 30 CAF-2 DEGs (AUC value>0.7, pct.1-pct.2>0.05). Right: Kaplan-Meier overall survival curve of 73 pure PDAC patients with low tumor cellularity grouped based on the expression of the top 30 CAF-2 DEGs. HR: hazard ratio. (g) Feature plot of the average expression of the top 30 CAF-2 DEGs scored in the mother plot (Pan02 tumor-infiltrated cells) to show CAF specificity of CAF-2 DEGs.

“These results suggested that excessive ECM production by CAFs confers chemotherapy resistance in PDAC.” The authors describe an association, but claim causality. I would recommend to rephrase this, unless this statement can be supported by experimental evidence, such as depletion of (a subset) of CAFs.

Considering the reviewer’s comment, we have rephrased the descriptions as follows. (p. 5, highlighted in yellow)

"These results suggested that gemcitabine treatment increases CAF populations in the PDAC microenvironment and facilitates tumor fibrosis."

REVIEWERS' COMMENTS

Reviewer #1 (Remarks to the Author):

The authors have done a nice job addressing the concerns of the prior critique. The addition of the KPC cell line orthotopic experiments, the T cell depletion, the longer term in vivo studies and the expanded analysis have improved the impact of the study. The in vivo data are consistent. It is an interesting and well done study.

Reviewer #2 (Remarks to the Author):

The authors have addressed my concerns. Congratulations on a very interesting manuscript.

I only have one minor edit to suggest. IN the abstract the authors write: "Patients with poor prognosis had high PIGF/VEGF expression and an increased number of PIGF/VEGF receptor-expressing CAFs, leading to enhanced collagen deposition". As this is data from patients is correlative not causative and I suggests the authors changes the text to "Patients with poor prognosis had high PIGF/VEGF expression and an increased number of PIGF/VEGF receptor-expressing CAFs, associated with enhanced collagen deposition"

Reviewer #3 (Remarks to the Author):

The authors have meaningfully and thoroughly responded to my comments from the first submission, and have substantially strengthened the manuscript in the process.

Reviewer #4 (Remarks to the Author):

I would like to thank the authors for providing lots of additional data and analyses. They have successfully addressed my key concerns.

Point-by-Point Response

We are grateful to the reviewers for their careful evaluation of our manuscript. In the text below, the reviewers' comments are in italics and our responses and descriptions of the changes made in the manuscript are in bold blue typeface.

Reviewer #1 (Remarks to the Author):

The authors have done a nice job addressing the concerns of the prior critique.

The addition of the KPC cell line orthotopic experiments, the T cell depletion, the longer term in vivo studies and the expanded analysis have improved the impact of the study. The in vivo data are consistent. It is an interesting and well done study.

We appreciate these favorable and supportive comments.

Reviewer #2 (Remarks to the Author):

The authors have addressed my concerns. Congratulations on a very interesting manuscript.

I only have one minor edit to suggest. IN the abstract the authors write: "Patients with poor prognosis had high PIGF/VEGF expression and an increased number of PIGF/VEGF receptor-expressing CAFs, leading to enhanced collagen deposition". As this is data from patients is correlative not causative and I suggests the authors changes the text to "Patients with poor prognosis had high PIGF/VEGF expression and an increased number of PIGF/VEGF receptor-expressing CAFs, associated with enhanced collagen deposition"

We, again, thank the reviewer for her/his comment. We have revised the sentence in the Abstract as the reviewer suggested (p. 2, highlighted as blue character):

"Patients with poor prognosis have high PIGF/VEGF expression and an increased number of PIGF/VEGF receptor-expressing CAFs, associated with enhanced collagen deposition."

Reviewer #3 (Remarks to the Author):

The authors have meaningfully and thoroughly responded to my comments from the first submission, and have substantially strengthened the manuscript in the process.

We appreciate the reviewer's endorsement of our work.

Reviewer #4 (Remarks to the Author):

I would like to thank the authors for providing lots of additional data and analyses. They have successfully addressed my key concerns.

We appreciate the reviewer's assessment of our work.